# Finite-Time Analysis of Fully Decentralized Single-Timescale Actor-Critic

## Abstract

Decentralized Actor-Critic (AC) algorithms have been widely utilized for multi-agent reinforcement learning (MARL) and have achieved remarkable success. Apart from its empirical success, the theoretical convergence property of decentralized AC algorithms is largely unexplored. The existing finite-time convergence results are derived based on either double-loop update or two-timescale step sizes rule, which is not often adopted in real implementation. In this work, we introduce a fully decentralized AC algorithm, where actor, critic, and global reward estimator are updated in an alternating manner with step sizes being of the same order, namely, we adopt the *single-timescale* update. Theoretically, using linear approximation for value and reward estimation, we show that our algorithm has sample complexity of $\tilde{\mathcal{O}}(\epsilon^{-2})$ under Markovian sampling, which matches the optimal complexity with double-loop implementation (here, $\tilde{\mathcal{O}}$ hides a log term). The sample complexity can be improved to $\mathcal{O}(\epsilon^{-2})$ under the i.i.d. sampling scheme. The central to establishing our complexity results is *the hidden smoothness of the optimal critic variable* we revealed. We also provide a local action privacy-preserving version of our algorithm and its analysis. Finally, we conduct experiments to show the superiority of our algorithm over the existing decentralized AC algorithms.

## 1 Introduction

Multi-agent reinforcement learning (MARL) [16, 30] has been very successful in various models of multi-agent systems, such as robotics [14], autonomous driving [37], Go [25], etc. MARL has been extensively explored in the past decades; see, e.g., [18, 20, 41, 26, 8, 22]. These works either focus on the setting where an central controller is available, or assuming a common reward function for all agents. Among the many cooperative MARL settings, the work [42] proposes the fully decentralized MARL with networked agents. In this setting, each agent maintains a private heterogeneous reward function, and agents can only access local/neighboring information through communicating with its neighboring agents on the network. Then, the objective of all agents is to jointly maximize the average long-term reward through interacting with environment modeled by multi-agent Markov decision process (MDP). They proposed the decentralized Actor-Critic (AC) algorithm to solve this MARL problem, and showed its impressive performance. However, the theoretical convergence properties of such class of decentralized AC algorithms are largely unexplored; see [41] for a comprehensive survey. In this work, our goal is to establish the strong finite-time convergence results under this fully decentralized MARL setting. We first review some recent progresses on this line of research below.

**Related works and motivations.** The first fully decentralized AC algorithm with provable convergence guarantee was proposed by [42], and they achieved asymptotic convergence results under two-time scale step sizes, which requires actor's step sizes to diminish in a faster scale than the critic's step sizes. The sample complexities of decentralized AC were established recently. In particular, [6]

and [11] independently propose two communication efficient decentralized AC algorithms with optimal sample complexity of $\mathcal{O}(\varepsilon^{-2}\log(\varepsilon^{-1}))$ under Markovian sampling scheme. Their analysis are based on *double-loop* implementation, where each policy optimization step follows a nearly accurate critic optimization step (a.k.a. policy evaluation), i.e., solving the critic optimization subproblem to $\varepsilon$-accuracy. Such a double-loop scheme requires careful tuning of two additional hyper-parameters, which are the batch size and inner loop size. In particular, the batch size and inner loop size need to be of order $\mathcal{O}(\varepsilon^{-1})$ and $\mathcal{O}(\log(\varepsilon^{-1}))$ in order to achieve their sample complexity results, respectively. In practice, single-loop algorithmic framework is often utilized, where one updates the actor and critic in an alternating manner by performing only one algorithmic iteration for both of the two subproblems; see, e.g., [23, 18, 15, 39]. The work [38] proposes a new decentralized AC algorithm based on such a single-loop alternative update. Nevertheless, they have to adopt *two-timescale* step sizes rule to ensure convergence, which requires actor's step sizes to diminish in a faster scale than the critic's step sizes. Due to the separation of the step sizes, the critic optimization sub-problem is solved exactly when the number of iterations tends to $\infty$. Such a restriction on the step size will slow down the convergence speed of the algorithm. As a consequence, they only obtain sub-optimal sample complexity of $\mathcal{O}(\varepsilon^{-\frac{5}{2}})$. In practice, most algorithms are implemented with *single-timescale* step size rule, where the step sizes for actor and critic updates are of the same order. Though there are some theoretical achievements for single-timescale update in other areas such as TDC [31] and bi-level optimization [4], similar theoretical understanding under AC setting is largely unexplored.

Indeed, even when reducing to single-agent setting, the convergence property of single-timescale AC algorithm is not well established. The works [9, 10] establish the finite-time convergence result under a special single-timescale implementation, where they attain the sample complexity of $\mathcal{O}(\varepsilon^{-2})$. However, their analysis is based on an algorithm where the critic optimization step is formulated as a least-square temporal difference (LSTD) at each iteration, where they need to sample the transition tuples for $\tilde{\mathcal{O}}(\varepsilon^{-1})$ times to form the data matrix in the LSTD problem. Then, they solve the LSTD problem in a closed-form fashion, which requires to invert a matrix of large size. Later, [4] obtains the same sample complexity using TD(0) update for critic variables under i.i.d. sampling. Nonetheless, their analysis highly relies on the assumption that the Jacobian of the stationary distribution is Lipschitz continuous, which is not justified in their work.

The above observations motivate us to ask the following question:

*Can we establish finite-time convergence result for decentralized AC algorithm with single-timescale step sizes rule?*[1]

**Main contributions.** By answering this question positively, we have the following contributions:

- We design a fully decentralized AC algorithm, which employs a *single-timescale* step sizes rule and adopts Markovian sampling scheme. The proposed algorithm allows communication between agents for every $K_c$ iterations with $K_c$ being any integer lies in $[1, \mathcal{O}(\varepsilon^{-\frac{1}{2}})]$, rather than communicating at each iteration as adopted by previous single-loop decentralized AC algorithms [38, 42].

- Using linear approximation for value and reward estimation, we establish the *finite-time* convergence result for such an algorithm under the standard assumptions. In particular, we show that the algorithm has the sample complexity of $\tilde{\mathcal{O}}(\varepsilon^{-2})$, which matches the optimal complexity up to a logarithmic term. In addition, we show that the logarithmic term can be removed under the i.i.d. sampling scheme. Note that these convergence results are valid for all the above mentioned choices for $K_c$.

- To preserve the privacy of local actions, we propose a variant of our algorithm which utilizes noisy local rewards for estimating global rewards. We show that such an algorithm will maintain the optimal sample complexity at the expense of communicating at each iteration.

The underlying principle for obtaining the above convergence results is that we reveal *the hidden smoothness of the optimal critic variable*, so that we can derive an approximate descent on the averaged critic's optimal gap at each iteration. Consequently, we can resort to the classic convergence analysis for alternating optimization algorithms to establish the approximate ascent property of the overall optimization process, which leads to the final sample complexity results.

---

[1]As convention [9], when we use "single-timescale", it means we utilize a single-loop algorithmic framework with single-timescale step sizes rule.

89  Another technical highlight is the Lyapunov function we construct for measuring the progress of our
90  algorithm. Such a construction is motivated by [4], which analyzes bi-level optimization algorithm.
91  However, our Lyapunov function is different from theirs as it involves the additional optimal gap of
92  averaged critic and reward estimator, which is necessary for dealing with the decentralized setting.

93  We finish this section by remarking that our convergence results are even new for single agent AC
94  algorithms under the setting of single-timescale step sizes rule.

## 2  Preliminary

96  In this section, we introduce the problem formulation and the policy gradient theorem, which serves
97  as the preliminary for the analyzed decentralzed AC algorithm.

98  Suppose there are multiple agents aiming to independently optimize a common global objective, and
99  each agent can communicate with its neighbors through a network. To model the topology, we define
100 the graph as $\mathcal{G} = (\mathcal{N}, \mathcal{E})$, where $\mathcal{N}$ is the set of nodes with $|\mathcal{N}| = N$ and $\mathcal{E}$ is the set of edges with
101 $|\mathcal{E}| = E$. In the graph, each node represents an agent, and each edge represents a communication
102 link. The interaction between agents follows the networked multi-agent MDP.

### 2.1  Markov decision process

104 A networked multi-agent MDP is defined by a tuple $(\mathcal{G}, \mathcal{S}, \{\mathcal{A}^i\}_{i\in\mathcal{N}}, \mathcal{P}, \{r^i\}_{i\in[N]}, \gamma)$. $\mathcal{G}$ denotes the
105 communication topology (the graph), $\mathcal{S}$ is the finite state space observed by all agents, $\mathcal{A}^i$ represents
106 the finite action space of agent $i$. Let $\mathcal{A} := \mathcal{A}^1 \times \cdots \times \mathcal{A}^N$ denote the joint action space and
107 $\mathcal{P}(s'|s, a) : \mathcal{S} \times \mathcal{A} \times \mathcal{S} \to [0, 1]$ denote the transition probability from any state $s \in \mathcal{S}$ to any state
108 $s' \in \mathcal{S}$ for any joint action $a \in \mathcal{A}$. $r^i : \mathcal{S} \times \mathcal{A} \to \mathbb{R}$ is the local reward function that determines the
109 reward received by agent $i$ given transition $(s, a)$; $\gamma \in [0, 1]$ is the discount factor.

110 For simplicity, we will use $a := [a^1, \cdots, a^N]$ to denote the joint action, and $\theta := [\theta^1, \cdots, \theta^N] \in$
111 $\mathbb{R}^{d_\theta \times N}$ to denote joint parameters of all actors, with $\theta^i \in \mathbb{R}^{d_\theta}$. Note that different actors may have
112 different number of parameters, which is assumed to be the same for our paper without loss of
113 generality. The MDP goes as follows: For a given state $s$, each agent make its decision $a^i$ based
114 on its policy $a^i \sim \pi_{\theta^i}(\cdot|s)$. The state transits to the next state $s'$ based on the joint action of all the
115 agents: $s' \sim \mathcal{P}(\cdot|s, a)$. Then, each agent will receive its own reward $r^i(s, a)$. For the notation brevity,
116 we assume that the reward function mapping is deterministic and does not depend on the next state
117 without loss of generality. The stationary distribution induced by the policy $\pi_\theta$ and the transition
118 kernel is denoted by $\mu_{\pi_\theta}(s)$.

119 Our objective is to find a set of policies that maximize the accumulated discounted mean reward
120 received by agents

$$\theta^* = \arg\max_\theta J(\theta) := \mathbb{E}\left[\sum_{k=0}^{\infty} \gamma^k \bar{r}(s_k, a_k)\right]. \tag{1}$$

121 Here, $k$ represents the time step. $\bar{r}(s_k, a_k) := \frac{1}{N}\sum_{i=1}^{N} r^i(s_k, a_k)$ is the mean reward among agents
122 at time step $k$. The randomness of the expectation comes from the initial state distribution $\mu_0(s)$, the
123 transition kernel $\mathcal{P}$, and the stochastic policy $\pi_{\theta^i}(\cdot|s)$.

### 2.2  Policy gradient Theorem

125 Under the discounted reward setting, the global state-value function, action-value function, and
126 advantage function for policy set $\theta$, state $s$, and action $a$, are defined as

$$V_{\pi_\theta}(s) := \mathbb{E}\left[\sum_{k=0}^{\infty} \gamma^k \bar{r}(s_k, a_k)|s_0 = s\right] \tag{2}$$

$$Q_{\pi_\theta}(s, a) := \mathbb{E}\left[\sum_{k=0}^{\infty} \gamma^k \bar{r}(s_k, a_k)|s_0 = s, a_0 = a\right]$$

$$A_{\pi_\theta}(s, a) := Q_{\pi_\theta}(s, a) - V_{\pi_\theta}(s).$$

To maximize the objective function defined in (1), the policy gradient [28] can be computed as follow

$$\nabla_\theta J(\theta) = \mathbb{E}_{s \sim d_{\pi_\theta}, a \sim \pi_\theta} \left[ \frac{1}{1-\gamma} A_{\pi_\theta}(s, a) \psi_{\pi_\theta}(s, a) \right],$$

where $d_{\pi_\theta}(s) := (1 - \gamma) \sum_{k=0}^{\infty} \gamma^k \mathbb{P}(s_k = s)$ is the discounted state visitation distribution under policy $\pi_\theta$, and $\psi_{\pi_\theta}(s, a) := \nabla \log \pi_\theta(s, a)$ is the score function.

Following the derivation of [42], the policy gradient for each agent under discounted reward setting can be expressed as

$$\nabla_{\theta^i} J(\theta) = \mathbb{E}_{s \sim d_{\pi_\theta}, a \sim \pi_\theta} \left[ \frac{1}{1-\gamma} A_{\pi_\theta}(s, a) \psi_{\pi_{\theta^i}}(s, a^i) \right]. \tag{3}$$

# 3 Decentralized single-timescale actor-critic

---

**Algorithm 1:** Decentralized single-timescale AC (reward estimator version)

---

1: **Initialize:** Actor parameter $\theta_0$, critic parameter $\omega_0$, reward estimator parameter $\lambda_0$, initial state $s_0$.
2: **for** $k = 0, \cdots, K - 1$ **do**
3:     **Option 1: i.i.d. sampling:**
4:     $s_k \sim \mu_{\theta_k}(\cdot), a_k \sim \pi_{\theta_k}(\cdot|s_k), s_{k+1} \sim \mathcal{P}(\cdot|s_k, a_k)$.
5:     **Option 2: Markovian sampling:**
6:     $a_k \sim \pi_{\theta_k}(\cdot|s_k), s_{k+1} \sim \mathcal{P}(\cdot|s_k, a_k)$.
7:
8:     **Periodical consensus:** Compute $\tilde{\omega}_k^i$ and $\tilde{\lambda}_k^i$ by (4) and (7).
9:
10:     **for** $i = 0, \cdots, N$ **in parallel do**
11:         **Reward estimator update:** Update $\lambda_{k+1}^i$ by (8).
12:         **Critic update:** Update $\omega_{k+1}^i$ by (5).
13:         **Actor update:** Update $\theta_{k+1}^i$ by (6).
14:     **end for**
15: **end for**

---

We introduce the decentralized single-timescale AC algorithm; see Algorithm 1. In the remaining parts of this section, we will explain the updates in the algorithm in details.

In fully-decentralized MARL, each agent can only observe its local reward and action, while trying to maximize the global reward (mean reward) defined in (1). The decentralized AC algorithm solves the problem by performing online updates in an alternative fashion. Specifically, we have $N$ pairs of actor and critic. In order to maximize $J(\theta)$, each critic tries to estimate the *global* state-value function $V_{\pi_\theta}(s)$ defined in (2), and each actor then updates its policy parameter based on approximated policy gradient. We now provide more details about the algorithm.

**Critics' update.** We will use $\omega^i \in \mathbb{R}^{d_\omega}$ to denote the $i_{th}$ critic's parameter and $\bar{\omega} := \frac{1}{N} \sum_{i=1}^{N} \omega^i$ to represent the averaged parameter of critic. The $i_{th}$ critic approximates the global value function as $V_{\pi_\theta}(s) \approx \hat{V}_{\omega^i}(s)$.

As we will see, the critic's approximation error can be categorized into two parts, namely, the consensus error $\frac{1}{N} \sum_{i=1}^{N} \|\omega^i - \bar{\omega}\|$, which measures how close the critics' parameters are; and the approximation error $\|\bar{\omega} - \omega^*(\theta)\|$, which measures the approximation quality of averaged critic.

In order for critics to reach consensus, we perform the following update for all critics

$$\tilde{\omega}_k^i = \begin{cases} \sum_{j=1}^{N} W^{ij} \omega_k^j & \text{if } k \bmod K_c = 0 \\ \omega_k^i & \text{otherwise.} \end{cases} \tag{4}$$

where $W \in \mathbb{R}^{n \times n}$ is a weight matrix for communication among agents, whose property will be specified in Assumption 5; $K_c$ denotes the consensus frequency.

To reduce the approximation error, we will perform the local TD(0) update [29] as

$$\omega_{k+1}^i = \prod_{R_\omega} (\tilde{\omega}_k^i + \beta_k g_c^i(\xi_k, \omega_k^i)), \tag{5}$$

where $\xi := (s, a, s')$ represents a transition tuple, $g_c^i(\xi, \omega) := \delta^i(\xi, \omega)\nabla \hat{V}_\omega(s)$ is the update direction, $\delta^i(\xi, \omega) := r^i(s, a) + \gamma \hat{V}_\omega(s') - \hat{V}_\omega(s)$ is the local temporal difference error (TD-error). $\beta_k$ is the step size for critic at iteration $k$. $\prod_{R_\omega}$ projects the parameter into a ball of radius of $R_\omega$ containing the optimal solution, which will be explained when discussing Assumption 1 and 2.

**Actors' update.** We will use stochastic gradient ascent to update the policy's parameter, and the stochastic gradient is calculated based on policy gradient theorem in (3). The advantage function $A_{\pi_\theta}(s, a)$ can be estimated by

$$\delta(\xi, \theta) := \bar{r}(s, a) + \gamma V(s') - V(s),$$

with $a$ sampled from $\pi_\theta(\cdot|s)$. However, to preserve the privacy of each agents, the local reward cannot be shared to other agents under the fully decentralized setting. Thus, the averaged reward $\bar{r}(s_k, a_k)$ is not directly attainable. Consequently, we need a strategy to approximate the averaged reward. In this paper, we will adopt the strategy proposed in [42]. In particular, each agent $i$ will have a local reward estimator with parameter $\lambda^i \in \mathbb{R}^{d_\lambda}$, which estimates the global averaged reward as $\bar{r}(s_k, a_k) \approx \hat{r}_{\lambda^i}(s_k, a_k)$.

Thus, the update of the $i_{th}$ actor is given by

$$\theta_{k+1}^i = \theta_k^i + \alpha_k \hat{\delta}(\xi_k, \omega_{k+1}^i, \lambda_{k+1}^i)\psi_{\pi_{\theta_k^i}}(s_k, a_k^i), \tag{6}$$

where $\hat{\delta}(\xi, \omega, \lambda) := \hat{r}_\lambda(s, a) + \gamma \hat{V}_\omega(s') - \hat{V}_\omega(s)$ is the approximated advantage function. $\alpha_k$ is the step size for actor's update at iteration $k$.

**Reward estimators' update.** Similar to critic, each reward estimator's approximation error can be decomposed into consensus error and the approximation error.

For each local reward estimator, we perform the consensus step to minimize the consensus error as

$$\tilde{\lambda}_k^i = \begin{cases} \sum_{j=1}^N W^{ij}\lambda_k^j & \text{if } k \bmod K_c = 0 \\ \lambda_k^i & \text{otherwise.} \end{cases} \tag{7}$$

To reduce the approximation error, we perform a local update of stochastic gradient descent.

$$\lambda_{k+1}^i = \prod_{R_\lambda}(\tilde{\lambda}_k^i + \eta_k g_r^i(\xi_k, \lambda_k^i)), \tag{8}$$

where $g_r^i(\xi, \lambda) := (r^i(s, a) - \hat{r}_\lambda(s, a))\nabla \hat{r}_\lambda(s, a)$ is the update direction. $\eta_k$ is the step size for reward estimator at iteration $k$. Note the calculation of $g_r^i(\xi, \lambda)$ does not require the knowledge of $s'$; we use $\xi$ in (8) just for notation brevity. Similar to critic's update, $\prod_{R_\lambda}$ projects the parameter into a ball of radius of $R_\lambda$ containing the optimal solution.

In our Algorithm 1, we will use the same order for $\alpha_k$, $\beta_k$, and $\eta_k$ and hence, our algorithm is in *single-timescale*.

**Linear approximation for analysis.** In our analysis, we will use linear approximation for both critic and reward estimator variables, i.e. $\hat{V}_\omega(s) := \phi(s)^T\omega; \hat{r}_\lambda(s, a) := \varphi(s, a)^T\lambda$, where $\phi(s) : \mathcal{S} \to \mathbb{R}^{d_\omega}$ and $\varphi(s, a) : \mathcal{S} \times \mathcal{A} \to \mathbb{R}^{d_\lambda}$ are two feature mappings, whose property will be specified in the discussion of Assumption 1.

**Algorithm for preserving the local action.** Note that in Algorithm 1, the reward estimators need the knowledge of joint actions in order to estimate the global rewards. To preserve the privacy of local actions, we further propose a variant of Algorithm 1, which estimates the global rewards by communicating noisy local rewards; see [6] for the original idea. However, to maintain the optimal sample complexity, such an approach requires $\mathcal{O}(\log(\varepsilon^{-1}))$ communication rounds for each iteration. We postpone the detailed design and analysis of such an algorithm scheme into Appendix B.

**Remarks on sampling scheme.** The unbiased update for critic and actor variables requires sampling from $\mu_{\pi_\theta}$ and $d_{\pi_\theta}$, respectively. However, in practical implementations, states are usually collected from an online trajectory (Markovian sampling), whose distribution is generally different for $\mu_{\pi_\theta}$ and $d_{\pi_\theta}$. Such a distribution mismatch will inevitably cause biases during the update of critic and actor variables. One has to bound the corresponding error terms when analyzing the algorithm. In this work, we will provide the analysis for both sampling schemes.

## 4  Main Results

In this section, we first introduce the technical assumptions used for our analysis, which are standard in the literature. Then, we present the convergence results for both actor and critic variables under i.i.d. sampling and Markovian sampling.

### 4.1  Assumptions

**Assumption 1** (bounded rewards and feature vectors). *All the local rewards are uniformly bounded, i.e., there exists a positive constants $r_{\max}$ such that $|r^i(s,a)| \le r_{\max}$, for all feasible $(s,a)$ and $i \in [N]$. The norm of feature vectors are bounded such that for all $s \in \mathcal{S}$, $a \in \mathcal{A}$, $\|\phi(s)\| \le 1, \|\varphi(s,a)\| \le 1$.*

Assumption 1 is standard and commonly adopted; see, e.g., [3, 35, 38, 24, 21]. This assumption can be achieved via normalizing the feature vectors.

**Assumption 2** (negative definiteness of $A_{\theta,\phi}$ and $A_{\theta,\varphi}$). *There exists two positive constants $\lambda_\phi, \lambda_\varphi$ such that for all policy $\theta$, the following two matrices are negative definite*

$$A_{\theta,\phi} := \mathbb{E}_{s \sim \mu_\theta(s)}[\phi(s)(\gamma\phi(s')^T - \phi(s)^T)]$$

$$A_{\theta,\varphi} := \mathbb{E}_{s \sim \mu_\theta(s), a \sim \pi_\theta(\cdot|s)}[-\varphi(s,a)\varphi(s,a)^T],$$

*with $\lambda_{\max}(A_{\theta,\phi}) \le \lambda_\phi, \lambda_{\max}(A_{\theta,\varphi}) \le \lambda_\varphi$, where $\lambda_{\max}(\cdot)$ represents the largest eigenvalue.*

Assumption 2 can be achieved when the matrices $\Phi_\phi := [\phi(s_1), \cdots, \phi(s_{|\mathcal{S}|})]$ and $\Phi_\varphi := [\varphi(s_1,a_1), \cdots, \varphi(s_{|\mathcal{S}|}, a_{|\mathcal{A}|})]$ have full row rank, which ensures that the optimal critic and reward estimator are unique; see also [24, 34]. Together with Assumption 1, we can show that the norm of $\omega^*(\theta)$ and $\lambda^*(\theta)$ are bounded by some positive constant, which justifies the projection steps.

**Assumption 3** (Lipschitz properties of policy). *There exists constants $C_\psi, L_\psi, L_\pi$ such that for all $\theta, \theta', s \in \mathcal{S}$ and $a \in \mathcal{A}$, we have (1). $|\pi_\theta(a|s) - \pi_{\theta'}(a|s)| \le L_\pi \|\theta - \theta'\|$; (2). $\|\psi_\theta(s,a) - \psi_{\theta'}(s,a)\| \le L_\psi \|\theta - \theta'\|$; (3). $\|\psi_\theta(s,a)\| \le C_\psi$.*

Assumption 3 is common for analyzing policy-based algorithms; see, e.g., [33, 32, 11]. The assumption ensures the smoothness of objective function $J(\theta)$. It holds for a large range of policy classes such as tabular softmax policy [1], Gaussian policy [7], and Boltzman policy [13].

**Assumption 4** (irreducible and aperiodic Markov chain). *The Markov chain under $\pi_\theta$ and transition kernel $\mathcal{P}(\cdot|s,a)$ is irreducible and aperiodic for any $\theta$.*

Assumption 4 is a standard assumption, which holds for any uniformly ergodic Markov chains and any time-homogeneous Markov chains with finite-state space. It ensures that there exists constants $\kappa > 0$ and $\rho \in (0,1)$ such that

$$\sup_{s \in \mathcal{S}} d_{TV}(\mathbb{P}(s_k \in \cdot|s_0 = s, \pi_\theta), \mu_\theta) \le \kappa\rho^k, \ \forall k.$$

**Assumption 5** (doubly stochastic weight matrix). *The communication matrix $W$ is doubly stochastic, i.e. each column/row sum up to 1. Moreover, the second largest singular value $\nu$ is smaller than 1.*

Assumption 5 is a common assumption in decentralized optimization and multi-agent reinforcement learning; see, e.g., [27, 5, 6]. It ensures the convergence of consensus error for critic and reward estimator variables.

### 4.2  Sample complexity under i.i.d. sampling

**Theorem 1** (sample complexity under i.i.d. sampling). *Suppose Assumptions 1-5 hold. Consider the update of Algorithm 1 under i.i.d. sampling. Let $\alpha_k = \frac{\bar{\alpha}}{\sqrt{K}}$ for some positive constant $\bar{\alpha}$, $\beta_k = \frac{C_9}{2\lambda_\phi}\alpha_k$, and $\eta_k = \frac{C_{10}}{2\lambda_\varphi}\alpha_k$, $K_c \le \mathcal{O}(\alpha_k^{-\frac{1}{2}})$, where $K$ denotes the total number of iterations. Then, we have*

$$\frac{1}{K}\sum_{k=1}^{K}\sum_{i=1}^{N}\mathbb{E}\left[\|\omega_k^i - \omega^*(\theta_k)\|^2\right] \le \mathcal{O}\left(\frac{1}{\sqrt{K}}\right)$$

$$\frac{1}{K}\sum_{k=1}^{K}\sum_{i=1}^{N}\mathbb{E}\left[\|\nabla_{\theta^i}F(\theta_k)\|^2\right] \le \mathcal{O}\left(\frac{1}{\sqrt{K}}\right) + \mathcal{O}(\varepsilon_{app} + \varepsilon_{sp}), \tag{9}$$

where $C_9, C_{10}$ are positive constants defined in the proof.

The proof of Theorem 1 can found in Appendix E.1. It establishes the iteration complexity of $\mathcal{O}(1/\sqrt{K})$, or equivalently, sample complexity of $\mathcal{O}(\varepsilon^{-2})$ for Algorithm 1. Note that actors, critics, and reward estimators use the step sizes of the same order. The sample complexity matches the optimal rate of SGD for general non-convex optimization problem. To explain the errors in (9), let us define the approximation error as the following:

$$\varepsilon_{app} := \max_{\theta,a} \sqrt{\mathbb{E}_{s\sim\mu_\theta}\left[|V_{\pi_\theta}(s) - \hat{V}_{\omega^*(\theta)}(s)|^2 + |\bar{r}(s,a) - \hat{r}_{\lambda^*(\theta)}(s,a)|^2\right]}.$$

The error $\varepsilon_{app}$ captures the approximation power of critic and reward estimator. Similar terms also appear in the literature (see e.g., [35, 1, 21]). Such an approximation error becomes zero in tabular case. The error $\varepsilon_{sp}$ is inevitably caused by the mismatch between discounted state visitation distribution $d_{\pi_\theta}$ and stationary distribution $\mu_{\pi_\theta}$; see, e.g., [38, 24]. It is defined as

$$\varepsilon_{sp} := 2C_\theta(\log_\rho \kappa^{-1} + \frac{1}{\rho})(1-\gamma).$$

When $\gamma$ is close to 1, the error becomes small. This is because $d_{\pi_\theta}$ approaches to $\mu_{\pi_\theta}$ when $\gamma$ goes to 1. In the literature, some works assume that sampling from $d_{\pi_\theta}$ is permitted, thus eliminate this error; see, e.g., [4].

## 4.3 Sample complexity under markovian sampling

**Theorem 2** (sample complexity under Markovian sampling). *Suppose Assumptions 1-5 hold. Consider the update of Algorithm 1 under Markovian sampling. Let $\alpha_k = \frac{\bar{\alpha}}{\sqrt{K}}$ for some positive constant $\bar{\alpha}$, $\beta_k = \frac{C_9}{2\lambda_\phi}\alpha_k$, and $\eta_k = \frac{C_{10}}{2\lambda_\varphi}\alpha_k$, $K_c \leq \mathcal{O}(\alpha_k^{-\frac{1}{2}})$, where $K$ is the total number of iterations. Then, we have*

$$\frac{1}{K}\sum_{k=1}^{K}\sum_{i=1}^{N}\mathbb{E}\left[\|\omega_k^i - \omega^*(\theta_k)\|^2\right] \leq \mathcal{O}\left(\frac{\log^2 K}{\sqrt{K}}\right)$$

$$\frac{1}{K}\sum_{k=1}^{K}\sum_{i=1}^{N}\mathbb{E}\left[\|\nabla_{\theta^i}F(\theta_k)\|^2\right] \leq \mathcal{O}\left(\frac{\log^2 K}{\sqrt{K}}\right) + \mathcal{O}(\varepsilon_{app} + \varepsilon_{sp}), \tag{10}$$

*where $C_9, C_{10}$ are positive constants defined in proof.*

We put the proof of Theorem 2 in Appendix E.2. In Markovian sampling, the updates are biased for critics, actors, and reward estimators. The error will decrease as the Markov chain mixes, and the logarithmic term is due to the cost for mixing.

Theorem 2 establishes the iteration complexity of $\mathcal{O}(\log^2 K/\sqrt{K})$, or equivalently, sample complexity of $\widetilde{\mathcal{O}}(\varepsilon^{-2})$ for Algorithm 1. It matches the state-of-the-art sample complexity of decentralized AC algorithms, which are implemented in double-loop fashion [11, 6].

## 4.4 Proof sketch

We present the main elements for the proof of Theorem 2, which helps in understanding the difference between classical two-timescale/double-loop analysis and our single-timescale analysis. The proof of Theorem 1 follows the same framework with simpler sampling scheme.

Under Markovian sampling, it is possible to show the following inequality, which characterizes the ascent of the objective.

$$\begin{aligned}
\mathbb{E}[J(\theta_{k+1})] - J(\theta_k) \geq \sum_{i=1}^{N}\Bigg[&\frac{\alpha_k}{2}\mathbb{E}\|\nabla_{\theta^i}J(\theta_k)\|^2 + \frac{\alpha_k}{2}\mathbb{E}\|g_a^i(\xi_k, \omega_{k+1}^i, \lambda_{k+1}^i)\|^2 \\
&- 8C_\psi^2\alpha_k\mathbb{E}\|\omega^*(\theta_k) - \omega_{k+1}^i\|^2 - 4C_\psi^2\alpha_k\mathbb{E}\|\lambda^*(\theta_k) - \lambda_{k+1}^i\|^2\Big] \\
&- \mathcal{O}(\log^2(K)\alpha_k^2) - \mathcal{O}((\varepsilon_{app} + \varepsilon_{sp})\alpha_k).
\end{aligned} \tag{11}$$

To analyze the errors of critic $\|\omega^*(\theta_k) - \omega_{k+1}^i\|^2$ and reward estimator $\|\lambda^*(\theta_k) - \lambda_{k+1}^i\|^2$, the two-timescale analysis requires $\mathcal{O}(\alpha_k) < \min\{\mathcal{O}(\beta_k), \mathcal{O}(\eta_k)\}$ in order for these two errors to converge. The double-loop approach runs lower-level update for $\mathcal{O}(\log(\varepsilon^{-1}))$ times with batch size $\mathcal{O}(\varepsilon^{-1})$ to drive these errors below $\varepsilon$ and hence, they cannot allow inner loop size and bath size to be $\mathcal{O}(1)$ simultaneously. To obtain the convergence result for *single-timescale* update, the idea is to further upper bound these two lower-level errors by the quantity $\mathcal{O}(\alpha_k \mathbb{E}\|g_a^i(\xi_k, \omega_{k+1}^i, \lambda_{k+1}^i)\|^2)$ (through a series of derivations), and then eliminate these errors by the ascent term $\frac{\alpha_k}{2}\mathbb{E}\|g_a^i(\xi_k, \omega_{k+1}^i, \lambda_{k+1}^i)\|^2$.

We mainly focus on the analysis of critic's error through the proof sketch. The analysis for reward estimator's error follows similar procedure. We start by decomposing the error of critic as

$$\sum_{i=1}^N \|\omega_{k+1}^i - \omega^*(\theta_k)\|^2 = \sum_{i=1}^N (\|\omega_{k+1}^i - \bar{\omega}_{k+1}\|^2 + \|\bar{\omega}_{k+1} - \omega^*(\theta_k)\|^2). \tag{12}$$

The first term represents the consensus error, which can be bounded by the next lemma.

**Lemma 1.** *Suppose Assumptions 1 and 5 hold. Consider the sequence $\{\omega_k^i\}$ generated by Algorithm 1, then the following holds*

$$\|Q\boldsymbol{\omega}_{k+1}\| \leq \nu^{\frac{k'}{K_c}}\|\boldsymbol{\omega}_0\| + 4\sum_{t=0}^k \nu^{\lceil \frac{k'-1-t}{K_c}\rceil}\beta_t\sqrt{N}C_\delta,$$

*where $\boldsymbol{\omega}_0 := [\omega^1, \cdots, \omega^N]^T, Q := I - \frac{1}{N}\mathbf{1}\mathbf{1}^T, k' := \lfloor\frac{k}{K_c}\rfloor * K_c$. The constant $\nu \in (0,1)$ is the second largest singular value of $W$.*

Based on Lemma 1 and follow the step size rule of Theorem 2, it is possible to show $\|Q\boldsymbol{\omega}_{k+1}\|_F^2 = \sum_{i=1}^N \|\omega_{k+1}^i - \bar{\omega}_{k+1}\|^2 = \mathcal{O}(K_c^2\beta_k^2)$. Let $K_c = \mathcal{O}(\beta_k^{-\frac{1}{2}})$, we have $\|Q\boldsymbol{\omega}_{k+1}\|_F^2 = \mathcal{O}(\beta_k)$, which maintains the optimal rate.

To analyze the second term in (12), we first construct the following Lyapunov function

$$\mathbb{V}_k := -J(\theta_k) + \|\bar{\omega}_k - \omega^*(\theta_k)\|^2 + \|\bar{\lambda}_k - \lambda^*(\theta_k)\|^2. \tag{13}$$

Then, it remains to derive an approximate descent property of the term $\|\bar{\omega}_k - \omega^*(\theta_k)\|^2$ in (13). Towards that end, our key step lies in establishing the *smoothness of the optimal critic variables* shown in the next lemma.

**Lemma 2** (smoothness of optimal critic). *Suppose Assumptions 1-3 hold, under the update of Algorithm 1, there exists a positive constant $L_{\mu,1}$ such that for all $\theta, \theta'$, it holds that*

$$\|\nabla\omega^*(\theta) - \nabla\omega^*(\theta')\| \leq L_{\mu,1}\|\theta - \theta'\|,$$

*where $\nabla\omega^*(\theta)$ denotes the Jacobian of $\omega^*(\theta)$ with respect to $\theta$.*

This smoothness property is essential for achieving our $\tilde{\mathcal{O}}(1/\sqrt{K})$ convergence rate.

To the best of our knowledge, the smoothness of $\omega^*(\theta)$ has not been justified in the literature. Equipped with Lemma 2, we are able to establish the following lemma.

**Lemma 3** (Error of critic). *Under Assumptions 1-5, consider the update of Algorithm 1. Then, it holds that*

$$\mathbb{E}[\|\bar{\omega}_{k+1} - \omega^*(\theta_{k+1})\|^2] \leq (1 + C_9\alpha_k)\|\bar{\omega}_{k+1} - \omega^*(\theta_k)\|^2$$
$$+ \frac{\alpha_k}{4}\sum_{i=1}^N \|\mathbb{E}[g_a^i(\xi_k, \omega_{k+1}^i, \lambda_{k+1}^i)]\|^2 + \mathcal{O}(\alpha_k^2). \tag{14}$$
$$\mathbb{E}[\|\bar{\omega}_{k+1} - \omega^*(\theta_k)\|^2] \leq (1 - 2\lambda_\phi\beta_k)\|\bar{\omega}_k - \omega^*(\theta_k)\|^2$$
$$+ C_{K_1}\beta_k\beta_{k-Z_K} + C_{K_2}\alpha_{k-Z_K}\beta_k. \tag{15}$$

*Here, $Z_K := \min\{z \in \mathbb{N}^+ | \kappa\rho^{z-1} \leq \min\{\alpha_k, \beta_k, \eta_k\}\}$, $C_9, \lambda_\phi$ are constants specified in appendix, and $C_{K_1}$ and $C_{K_2}$ are of order $\mathcal{O}(\log(K))$ and $\mathcal{O}(\log^2(K))d$ respectively.*

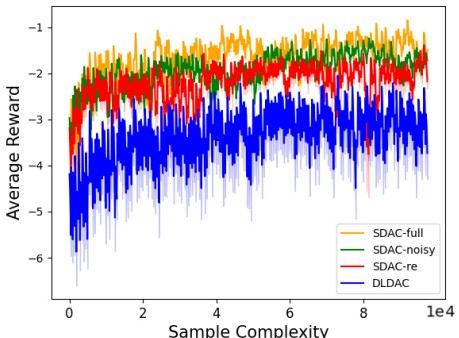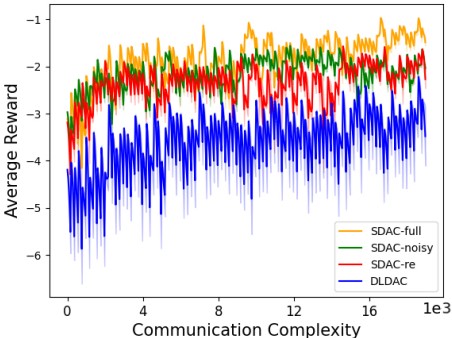

Figure 1: Averaged reward versus sample complexity and communication complexity. The vertical axis is the averaged reward over all the agents.

Plug (15) into (14), we can establish the approximate descent property of $\|\bar{\omega}_k - \omega^*(\theta_k)\|^2$ in (13):

$$
\begin{aligned}
\mathbb{E}[\|\bar{\omega}_{k+1} - \omega^*(\theta_{k+1})\|^2] \leq{} & (1 + C_9\alpha_k)(1 - 2\lambda_\phi\beta_k)\|\bar{\omega}_k - \omega^*(\theta_k)\|^2 \\
& + \frac{\alpha_k}{4}\sum_{i=1}^{N}\|\mathbb{E}[g_a^i(\xi_k, \omega_{k+1}^i, \lambda_{k+1}^i)]\|^2 \\
& + \mathcal{O}(C_{K_1}\beta_k\beta_{k-Z_K} + C_{K_2}\alpha_{k-Z_K}\beta_k).
\end{aligned}
\tag{16}
$$

Finally, plugging (11), (14), and (16) into (13) gives the ascent of the Lyapunov function, which leads to our convergence result through steps of standard arguments.

## 5   Numerical results

In this section, our objective is to illustrate the empirical sample complexity and communication complexity of the proposed algorithms. We also implement the algorithm in [6] to serve as a baseline, which employs double-loop algorithmic framework. Our simulation is based on the grounded communication environment proposed in [19]; see Appendix A for detailed set up. Through the discussion, we refer the algorithm in [6] as "DLDAC", the Algorithm 1 as "SDAC-re", the Algorithm 2 as "SDAC-noisy" (see Appendix B). We also provide the result which assumes full reward is available to serve as baseline, which we refer as "SDAC-full". We set $K_r = 5$ for "SDAC-noisy"; $K_c = 1$ for "SDAC-re", "SDAC-noisy", and "SDAC-full". We choose $T_c = 5$ (loop size), $T_c' = 1$ (critic consensus number every iteration), $T' = 5$ (reward consensus number every iteration) for "DLDAC".

The sample complexity and communication complexity are shown in Figure 1. The results are averaged over 10 Monte Carlo runs. As we can see, the proposed two algorithms achieve significantly higher reward than "DLDAC" in terms of both sample complexity and communication complexity. Moreover, their performances approach the baseline "SDAC-full", where the global reward is assumed to be available, indicating that the reward approximation is nearly accurate. Due to space limit, we will put additional experiments on the comparison with existing decentralized AC algorithms and the ablation study of hyper-parameters to Appendix A.

## 6   Conclusion and future direction

In this paper, we studied the convergence of fully decentralized AC algorithm under practical single-timescale update for the first time. We designed such an algorithm which maintains the optimal sample complexity of $\widetilde{\mathcal{O}}(\varepsilon^{-2})$ under less communications. We also proposed a variant to preserve the privacy of local actions by communicating noisy rewards. Extensive simulation results demonstrate the superiority of our algorithms' empirical performance over existing decentralized AC algorithms. One limitation of our work is that we only study the convergence to stationary point. Thus, we leave the research on the avoidance of saddle points and convergence to global optimum as promising future directions.

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
