experimental setting. Then, we present more experiments on the comparison between the proposed algorithms and existing decentralized AC algorithms. Additionally, we conduct ablation study on different consensus frequencies of the proposed algorithm.

**Experiment setting.** We adopt the grounded communication environment proposed in [19]. Our task consists of $N$ agents and the corresponding $N$ landmarks inhabited in a two-dimension world, where each agent can observe the relative position of other agents and landmarks. For every discrete time step, agents take actions to move along certain directions, and receive their rewards. Agents are rewarded based on the distance to their own landmark, and penalized if they collide with other agents. The objective is to maximize the long-term averaged reward over all agents. Since we focus on decentralized setting, each agent shall not know the target landmark of others, i.e., the reward function of others. To exchange information, each agent is allowed to send their local information via a fixed communication link. Through all the experiments, the agent number $N$ is set to be 5, and the discount factor $\gamma$ is set to be 0.95.

**Comparison to double-loop decentralized AC under mini-batch update.** Since the algorithm in [6] uses mini-batch update to reduce the variance during the update, we will compare the proposed algorithms with [6] under different choices of actor's batch sizes, critic's batch sizes, and inner loop sizes, respectively. Since their algorithm communicates noisy reward to achieve consensus, we will use "SDAC-noi" to serve as baseline.

1. **Actor's batch size.** We fix $T_c = 50$, $T_c' = 10$, $N_c = 10$, [2] which is adopted by [6]. We examine values of $N$ in $\{10, 50, 100\}$. The results are in Figure 2a. We observe that the best choice of actor's batch size $N$ is 50, and the proposed "SDAC-noi" converges faster than it in terms of sample complexity.

2. **Critic's batch size.** We fix $T_c = 50$, $T_c' = 10$, $N = 100$, which is adopted by [6]. We examine values of $N_c$ in $\{2, 10, 50\}$. The results are shown in Figure 2b. As we can see, "DLDAC" with smaller critic's batch sizes can achieve better sample complexity, indicating that the variance of critic's update is relatively small and the mini-batch update is not needed for this task. Our proposed "SDAC-noi" achieves better convergence compared with the double-loop decentralized AC under different choices of $N_c$.

3. **Inner loop size.** We fix $T_c' = 10$, $N = 100$, $N_c = 10$, which is adopted by [6]. We examine values of $T_c$ in $\{5, 20\}$. The results are shown in Figure 3. We can see that the proposed "SDAC-noi" enjoys a better convergence in terms of sample complexity.

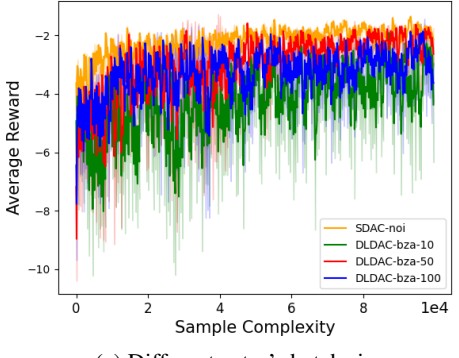
(a) Different actor's batch sizes.

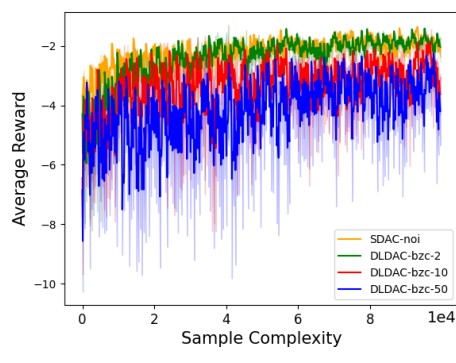
(b) Different critic's batch sizes.

Figure 2: Comparison between the proposed algorithms and the double-loop decentralized AC algorithm that uses mini-batch update. The results are averaged over 10 Monte Carlo runs.

---

[2]Note that we adopt the notations in [6]. Here, $T_c$ is the inner loop size, $T_c'$ is the communication number for each outer loop, $N$ is the batch size for actor's update, and $N_c$ is the batch size for critic's update.

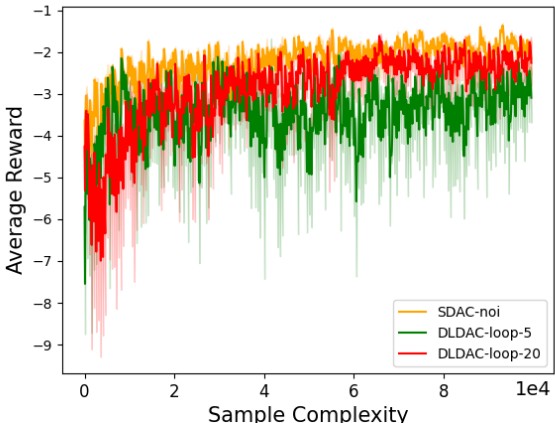

Figure 3: Comparison between the proposed algorithm and the double-loop decentralized AC algorithm under different inner loop sizes. The results are averaged over 10 Monte Carlo runs.

**Comparison to two-timescale decentralized AC.** Next, we compare the empirical performance between single-timescale and two-timescale implementations. The baseline we compare here is the existing decentralized two-timescale AC algorithm [38].

We use "TDAC-re" to denote the algorithm proposed in [38]. To compare with our proposed Algorithm 2, we also implement a noisy reward version of "TDAC-re" and denote it by "TDAC-noi". We fix $K_c = 1$, $K_r = 5$ for this experiment. We set $\alpha_k = 0.01(k+1)^{-0.5}$, $\beta_k = 0.1(k+1)^{-0.5}$, and $\eta_k = 0.1(k+1)^{-0.5}$ for "SDAC-re" and "SDAC-noi"; we set $\alpha_k = 0.01(k+1)^{-0.6}$, $\beta_k = 0.1(k+1)^{-0.4}$, and $\eta_k = 0.1(k+1)^{-0.4}$ for "TDAC-re" and "TDAC-noi". The sample complexity complexity is presented in Figure 4. We can see that the convergence speed of "TDAC-noi" is comparable to its single-timescale counterpart "SDAC-noi". However, when using reward estimator for the global reward estimation, we observe that "SDAC-re" has much more stable convergence behavior than "TDAC-re", and achieves significantly higher rewards.

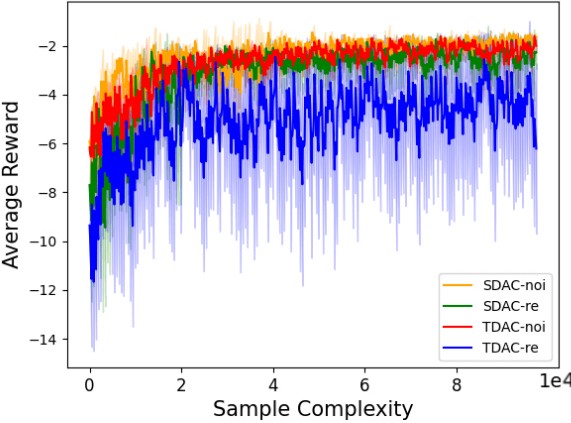

Figure 4: Comparison between the proposed algorithms and two-timescale decentralized AC algorithms [38]. The results are averaged over 10 Monte Carlo runs.

**Ablation on different consensus periods.** We compare the performance of "SDAC-noi" under different choices of consensus periods $K_c$. In particular, we let $\alpha_k = 0.01(k+1)^{-0.5}$, $\beta_k = 0.1(k+1)^{-0.5}$, $K_r = 1$ and examine the consensus periods $K_c$ of 1, 5, 10, and 20, respectively.

557 The corresponding sample complexities and are summarized in Figure 5. Evidently, as the consensus
558 period $K_c$ increases, the convergence becomes slower and become relatively unstable. Therefore,
559 when the communication cost is low, choosing a small $K_c$ will yield a better performance. For
560 this task, the consensus period $K_c$ should be kept within 5 rounds in order to ensure a reasonable
561 convergence. In Figure 5, we plot the communication complexity under the consensus periods of
562 1 and 5. We can see that the communication complexity of "cons-5" surpasses "cons-1" during the
563 training, indicating that it requires less rounds of communications to achieve better performance. Thus,
564 when the communication complexity is high, we may use large $K_c$ to achieve better communication
565 complexity. When extending the model to different tasks, we may try different values of $K_c$ to
566 balance the sample complexity and communication complexity.

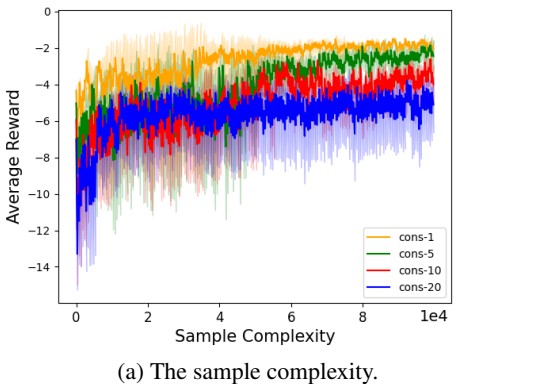

(a) The sample complexity.

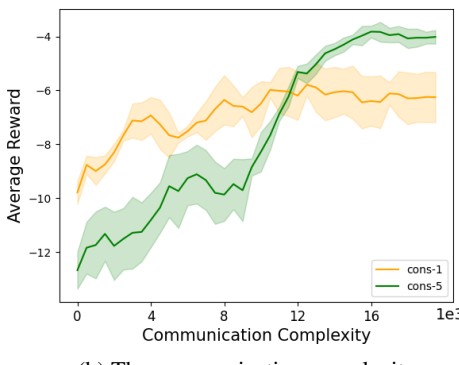

(b) The communication complexity.

Figure 5: Ablation study on the consensus periods. The results are averaged over 10 Monte Carlo runs.

# B Algorithm without local action

568 In this section, we introduce the variant of Algorithm 1 for preserving the privacy of local actions.
569 The main difference is that instead of using a reward estimator to approximate the global reward,
570 we now communicate the noisy local rewards for estimating the global rewards. Let $r_k^i$ represents
571 $r_k^i(s_k, a_k)$ for brevity. The reward estimation process goes as follow: for each agent $i$, we first
572 produce a noisy local reward $\tilde{r}_k^i = r_k^i(1 + z)$, with $z \sim \mathcal{N}(0, \sigma^2)$. Thus, the noise level is controlled
573 by the variance $\sigma^2$, which is chosen artificially. To estimate the global reward, each agent $i$ first
574 initialize the estimation as $\tilde{r}_{t,0}^i = \tilde{r}_t^i$. Then, each agent $i$ perform the following consensus step for $K_r$
575 times, i.e.

$$\tilde{r}_{t,l+1}^i = \sum_{j=1}^{N} W^{ij} \tilde{r}_{t,l}^i, \quad l = 0, 1, \cdots, K_r - 1. \tag{17}$$

576 The reward $\tilde{r}_{k,K_r}^i$ will be used for estimating global reward for agent $i$. The error for the reward
577 estimation, i.e. $|\bar{r}_k - \tilde{r}_{k,K_r}^i|$ will converge to 0 linearly. Therefore, to reduce the error to $\varepsilon$, we need
578 $K_r = \mathcal{O}(\log(\varepsilon^{-1}))$ rounds of communications.

579 The following theorem establishes the sample complexity of Algorithm 2 under Markovian sampling.
580 **Theorem 3.** *Suppose Assumptions 1-5 hold. Consider the update of Algorithm 2 under Markovian*
581 *sampling. Let $\alpha_k = \frac{\bar{\alpha}}{\sqrt{K}}$ for some positive constant $\bar{\alpha}$, $\beta_k = \frac{C_9}{2\lambda_\phi}\alpha_k$, $K_c = \mathcal{O}(\log(K^{1/4}))$,*
582 *$K_r = \log(K^{1/2})$. Then, we have*

$$\frac{1}{K}\sum_{k=1}^{K}\sum_{i=1}^{N} \mathbb{E}\left[\|\omega_k^i - \omega^*(\theta_k)\|^2\right] \leq \mathcal{O}\left(\frac{\log^2 K}{\sqrt{K}}\right)$$

$$\frac{1}{K}\sum_{k=1}^{K}\sum_{i=1}^{N} \mathbb{E}\left[\|\nabla_{\theta^i} F(\theta_k)\|^2\right] \leq \mathcal{O}\left(\frac{\log^2 K}{\sqrt{K}}\right) + \mathcal{O}(\varepsilon_{app} + \varepsilon_{sp}), \tag{18}$$

**Algorithm 2:** Decentralized single-timescale AC (noisy reward version)

---

1: **Initialize:** Actor parameter $\theta_0$, critic parameter $\omega_0$, initial state $s_0$.
2: **for** $k = 0, \cdots, K - 1$ **do**
3:     **Option 1: i.i.d. sampling:**
4:     $s_k \sim \mu_{\theta_k}(\cdot), a_k \sim \pi_{\theta_k}(\cdot|s_k), s_{k+1} \sim \mathcal{P}(\cdot|s_k, a_k)$.
5:     **Option 2: Markovian sampling:**
6:     $a_k \sim \pi_{\theta_k}(\cdot|s_k), s_{k+1} \sim \mathcal{P}(\cdot|s_k, a_k)$.
7:
8:     **Periodical consensus:** Compute $\tilde{\omega}_k^i$ by (4).
9:
10:     **for** $i = 0, \cdots, N$ **in parallel do**
11:         **Global reward estimation:** Estimate $\bar{r}_k(s_k, a_k)$ by (17).
12:         **Critic update:** Update $\omega_{k+1}^i$ by (5).
13:         **Actor update:** Update $\theta_{k+1}^i$ by (6).
14:     **end for**
15: **end for**

---

583   *where $C_9$ and $C_{10}$ are positive constants defined in proof.*

584   The Theorem 3 shows that Algorithm 2 has the same sample complexity as Algorithm 1; see
585   Appendix E.3 for the proof. Algorithm 2 enjoys the advantage of preserving local actions and requiring
586   less parameters since no reward estimator is needed. The cost is that we need to communicate
587   $\mathcal{O}(\log(\varepsilon^{-1}))$ times for each iteration.

## C   Auxiliary lemmas

589   In this section, we provide some auxiliary lemmas, which serves as the preliminary for the proof of
590   main theorems and lemmas.

591   **Lemma 4** ([40], Lemma 3.2)**.** *Suppose Assumption 3 holds, then there exists a positive constant $L$*
592   *such that for all $\theta, \theta' \in \mathbb{R}^{d_\theta}$, we have $\|\nabla J(\theta) - \nabla J(\theta')\| \leq L\|\theta - \theta'\|$.*

593   **Lemma 5** ([24], Lemma 1)**.** *Suppose Assumptions 4 holds, then there exists $\kappa > 0, \rho \in [0, 1]$ such*
594   *that for any $\theta \in \mathbb{R}^{Nd_\theta}$ we have*

$$\sup_{s_0 \in \mathcal{S}} d_{TV}(\mathbb{P}((s_k, a_k, s_{k+1}) \in \cdot|s_0, \pi_\theta), \mu_\theta \otimes \pi_\theta, \mathcal{P}) \leq \kappa\rho^k,$$

595   *where $\mu_\theta$ is the stationary distribution induced by $\pi_\theta$ and transition kernel $\mathcal{P}(\cdot|s, a)$.*

596   **Lemma 6** ([24], Lemma 2)**.** *Suppose Assumption 4 holds, then for any $\theta \in \mathbb{R}_\theta^d$, we have*

$$d_{TV}(d_\theta, \mu_\theta) \leq 2(\log_\rho \kappa^{-1} + \frac{1}{1-\rho})(1-\gamma).$$

597   **Lemma 7** ([24], Lemma 4)**.** *Suppose Assumption 3 holds, for any $\theta_1, \theta_2 \in \mathbb{R}^{d_\theta}$ and $s \in \mathcal{S}$, there*
598   *exits a positive constant $L_V$ such that*

$$\|\nabla V_{\pi_{\theta_1}}(s)\| \leq L_V$$
$$|V_{\pi_{\theta_1}}(s) - V_{\pi_{\theta_2}}(s)| \leq L_V\|\theta_1 - \theta_2\|.$$

599   **Lemma 8** ([32], Lemma A.1)**.** *For any policy $\theta_1$ and $\theta_2$, it holds that*

$$d_{TV}(\mu_{\theta_1}, \mu_{\theta_2}) \leq |\mathcal{A}|L_\pi(\log_\rho \kappa^{-1} + (1-\rho)^{-1})\|\theta_1 - \theta_2\|$$
$$d_{TV}(\mu_{\theta_1} \otimes \pi_{\theta_1}, \mu_{\theta_2} \otimes \pi_{\theta_2}) \leq |\mathcal{A}|L_\pi(1 + \log_\rho \kappa^{-1} + (1-\rho)^{-1})\|\theta_1 - \theta_2\|$$
$$d_{TV}(\mu_{\theta_1} \otimes \pi_{\theta_1} \otimes \mathcal{P}, \mu_{\theta_2} \otimes \pi_{\theta_2} \otimes \mathcal{P}) \leq |\mathcal{A}|L_\pi(1 + \log_\rho \kappa^{-1} + (1-\rho)^{-1})\|\theta_1 - \theta_2\|.$$

600   *We will define $L_\mu := |\mathcal{A}|L_\pi(\log_\rho \kappa^{-1} + (1-\rho)^{-1})$ for the proof of main theorems and lemmas.*

601   **Lemma 9** ([5], Lemma F.3)**.** *For a doubly stochastic matrix $W \in \mathbb{R}^{N \times N}$ and the difference matrix*
602   $Q := I - \frac{1}{N}\mathbf{1}\mathbf{1}^T$, *it holds that for any matrix $H \in \mathbb{R}^{N \times N}$, $\|W^k H\|_F \leq \nu^k \|QH\|_F$, where $\nu$ is the*
603   *second largest singular value of $W$.*

**Lemma 10** (descent lemma in high dimension). *Consider the mapping $F : \mathbb{R}^n \to \mathbb{R}^m$. If there exists a positive constant $L$ such that*

$$\|\nabla F(x) - \nabla F(y)\|_F \leq L\|x - y\|, \ \forall x, y \in dom(F), \tag{19}$$

*then the following holds*

$$\|F(y) - F(x) - \nabla F(x)(y - x)\| \leq \frac{L_1}{2}\sqrt{m}\|y - x\|^2.$$

*Proof.* Observe that (19) directly implies the smoothness of each entry $F_i$:

$$\|\nabla F_i(x) - \nabla F_i(y)\| \leq \|\nabla F(x) - \nabla F(y)\|_F \leq L_1\|x - y\|.$$

Define

$$z_i(x, y) := F_i(y) - F_i(x) - \nabla F_i(x)^T(y - x).$$

We have

$$
\begin{aligned}
\|F(y) - F(x) - \nabla F(x)(y - x)\| &= \sqrt{\sum_{i=1}^{m} z_i(x, y)^2} \\
&\leq \sqrt{m(\frac{L_1}{2}\|y - x\|^2)^2} \\
&= \frac{L_1}{2}\sqrt{m}\|y - x\|^2,
\end{aligned}
$$

where the inequality follows the descent lemma. $\square$

**Lemma 11** (Lipschitz property of multiplication). *Suppose $f(x)$ and $g(x)$ are two functions bounded by $C_f$ and $C_g$, and are $L_f$- and $L_g$-Lipschitz continuous, then $f(x)g(x)$ is $C_f L_g + C_g L_f$-Lipschitz continuous.*

*Proof.*

$$
\begin{aligned}
\|f(x_1)g(x_1) - f(x_2)g(x_2)\| &= \|f(x_1)g(x_1) - f(x_1)g(x_2) + f(x_1)g(x_2) - f(x_2)g(x_2)\| \\
&\leq \|f(x_1)\|\|g(x_1) - g(x_2)\| + \|f(x_1) - f(x_2)\|\|g(x_2)\| \\
&\leq (C_f L_g + C_g L_f)\|x_1 - x_2\|.
\end{aligned}
$$

$\square$

**Lemma 12** (invertible property of matrix). *If a square matrix $A$ satisfying $\lim_{t \to \infty} A^t = 0$, or equivalently, $|\lambda(A)| < 1$, then $I - A$ is invertible.*

*Proof.*

$$
\begin{aligned}
(I - A)\lim_{t \to \infty}\sum_{i=0}^{t} A^t &= \lim_{t \to \infty}[\sum_{i=0}^{t} A^t - \sum_{i=1}^{t+1} A^t] \\
&= I - \lim_{t \to \infty} A^{t+1} \\
&= I.
\end{aligned}
$$

Since $I$ is invertible, by the rank inequality $\text{rank}(AB) \leq \min(\text{rank}(A), \text{rank}(B))$, $I - A$ and $\lim_{t \to \infty}\sum_{i=0}^{t} A^t$ will be invertible. $\square$

**Lemma 13** (mismatch between Markovian sampling and stationary distribution). *Consider the Markov chain:*

$$s_{k-z} \xrightarrow{\theta_{k-z}} a_{k-z} \xrightarrow{\mathcal{P}} s_{k-z+1} \xrightarrow{\theta_{k-z+1}} a_{k-z+1} \cdots \xrightarrow{\theta_{k-1}} a_{k-1} \xrightarrow{\mathcal{P}} s_k \xrightarrow{\theta_k} a_k \xrightarrow{\mathcal{P}} s_{k+1}.$$

*Also consider the auxiliary Markov chain with fixed policy:*

$$s_{k-z} \xrightarrow{\theta_{k-z}} a_{k-z} \xrightarrow{\mathcal{P}} s_{k-z+1} \xrightarrow{\theta_{k-z}} \tilde{a}_{k-z+1} \cdots \xrightarrow{\theta_{k-z}} \tilde{a}_{k-1} \xrightarrow{\mathcal{P}} \tilde{s}_k \xrightarrow{\theta_{k-z}} \tilde{a}_k \xrightarrow{\mathcal{P}} \tilde{s}_{k+1}.$$

Let $\xi_k := (s_k, a_k, s_{k+1})$ be sampled from chain 1, and $\tilde{\xi}_k := (s_k, a_k, s_{k+1})$ be sampled from chain 2. Then we have

$$d_{TV}(\mathbb{P}(\xi_k \in \cdot | \theta_{k-z}, s_{k-z+1}), \mathbb{P}(\tilde{\xi}_k \in \cdot | \theta_{k-z}, s_{k-z+1})) \leq \frac{1}{2} \sum_{m=0}^{z-1} |\mathcal{A}| L_\pi \|\theta_{k-m} - \theta_{k-z}\|.$$

*Proof.*

$$d_{TV}(\mathbb{P}(\xi_k \in \cdot), \mathbb{P}(\tilde{\xi}_k \in \cdot))$$

$$= \frac{1}{2} \int_{s \in \mathcal{S}} \int_{s' \in \mathcal{S}} \sum_{a \in \mathcal{A}} |\mathbb{P}(s_k = ds, a_k = a, s_{k+1} = ds') - \mathbb{P}(\tilde{s}_k = ds, \tilde{a}_k = a, \tilde{s}_{k+1} = ds')|$$

$$= \frac{1}{2} \int_{s \in \mathcal{S}} \sum_{a \in \mathcal{A}} |\mathbb{P}(s_k = ds, a_k = a) - \mathbb{P}(\tilde{s}_k = ds, \tilde{a}_k = a)| \int_{s' \in \mathcal{S}} \mathbb{P}(s_{k+1} = ds' | s_k = ds, a_k = a)$$

$$= \frac{1}{2} \int_{s \in \mathcal{S}} \sum_{a \in \mathcal{A}} |\mathbb{P}(s_k = ds, a_k = a) - \mathbb{P}(\tilde{s}_k = ds, \tilde{a}_k = a)|$$

$$= \frac{1}{2} \int_{s \in \mathcal{S}} \sum_{a \in \mathcal{A}} |\mathbb{P}(s_k = ds)\pi_{\theta_k}(a|ds) - \mathbb{P}(\tilde{s}_k = ds)\pi_{\theta_{k-z}}(a|ds)|$$

$$\leq \frac{1}{2} \int_{s \in \mathcal{S}} \sum_{a \in \mathcal{A}} |\mathbb{P}(s_k = ds)\pi_{\theta_k}(a|ds) - \mathbb{P}(s_k = ds)\pi_{\theta_{k-z}}(a|ds)|$$

$$+ \frac{1}{2} \int_{s \in \mathcal{S}} \sum_{a \in \mathcal{A}} |\mathbb{P}(s_k = ds)\pi_{\theta_{k-z}}(a|ds) - \mathbb{P}(\tilde{s}_k = ds)\pi_{\theta_{k-z}}(a|ds)|$$

$$\leq \frac{1}{2} \int_{s \in \mathcal{S}} |\mathcal{A}| L_\pi \|\theta_k - \theta_{k-z}\| \mathbb{P}(s_k = ds)$$

$$+ \frac{1}{2} \int_{s \in \mathcal{S}} |\mathbb{P}(s_k = ds) - \mathbb{P}(\tilde{s}_k = ds)| \sum_{a \in \mathcal{A}} \pi_{\theta_{k-z}}(a|ds)$$

$$= \frac{1}{2} |\mathcal{A}| L_\pi \|\theta_k - \theta_{k-z}\| + d_{TV}(\mathbb{P}(s_k \in \cdot), \mathbb{P}(\tilde{s}_k \in \cdot)). \tag{20}$$

The second term can be bounded as

$$d_{TV}(\mathbb{P}(s_k \in \cdot), \mathbb{P}(\tilde{s}_k \in \cdot))$$

$$= \frac{1}{2} \int_{s' \in \mathcal{S}} |\mathbb{P}(s_k = ds) - \mathbb{P}(\tilde{s}_k = ds)|$$

$$= \frac{1}{2} \int_{s' \in \mathcal{S}} |\sum_{a \in \mathcal{A}} \int_{s \in \mathcal{S}} \mathbb{P}(s_{k-1} = ds, a_{k-1} = a, s_k = ds') - \mathbb{P}(\tilde{s}_{k-1} = ds, \tilde{a}_{k-1} = a, \tilde{s}_k = ds')|$$

$$\leq \frac{1}{2} \int_{s' \in \mathcal{S}} \sum_{a \in \mathcal{A}} \int_{s \in \mathcal{S}} |\mathbb{P}(s_{k-1} = ds, a_{k-1} = a, s_k = ds') - \mathbb{P}(\tilde{s}_{k-1} = ds, \tilde{a}_{k-1} = a, \tilde{s}_k = ds')|$$

$$= d_{TV}(\mathbb{P}(\xi_{k-1} \in \cdot), \mathbb{P}(\tilde{\xi}_{k-1} \in \cdot)). \tag{21}$$

Combined (20) and (21), we obtain

$$d_{TV}(\mathbb{P}(\xi_k \in \cdot), \mathbb{P}(\tilde{\xi}_k \in \cdot)) \leq d_{TV}(\mathbb{P}(\xi_{k-1} \in \cdot), \mathbb{P}(\tilde{\xi}_{k-1} \in \cdot)) + \frac{1}{2} |\mathcal{A}| L_\pi \|\theta_k - \theta_{k-z}\|.$$

Sum over $z - 1$ steps, we obtain

$$d_{TV}(\mathbb{P}(\xi_k \in \cdot), \mathbb{P}(\tilde{\xi}_k \in \cdot)) \leq d_{TV}(\mathbb{P}(\xi_{k-z} \in \cdot), \mathbb{P}(\tilde{\xi}_{k-z} \in \cdot)) + \frac{1}{2} \sum_{m=0}^{z-1} |\mathcal{A}| L_\pi \|\theta_{k-m} - \theta_{k-z}\|$$

$$= \frac{1}{2} \sum_{m=0}^{z-1} |\mathcal{A}| L_\pi \|\theta_{k-m} - \theta_{k-z}\|.$$

$\square$

 # D  Supporting lemmas

Before proceeding to the analysis of critic variables, we firstly justify the uniqueness of optimal solution for critic and reward estimator variables. Define the following notations

$$A_{\theta,\phi} := \mathbb{E}[\phi(s)(\gamma\phi(s')^T - \phi(s)^T)] \tag{22}$$
$$A_{\theta,\varphi} := \mathbb{E}[\varphi(s,a)\varphi(s,a)^T]$$
$$b_{\theta,\phi} := \mathbb{E}[\phi(s)\bar{r}(s,a)]$$
$$b_{\theta,\varphi} := \mathbb{E}[\varphi(s,a)\bar{r}(s,a)],$$

with expectation taken from $s \sim \mu_\theta(s), a \sim \pi_\theta, s' \sim \mathcal{P}$. The optimal critic and reward estimator variables given policy $\theta$ will satisfy $A_{\theta,\phi}\omega^*(\theta) + b_{\theta,\phi} = 0$; $A_{\theta,\varphi}\lambda^*(\theta) + b_{\theta,\varphi} = 0$. By Assumption 2, $A_{\theta,\phi}$ and $A_{\theta,\varphi}$ are negative definite with largest eigenvalue $\lambda_\phi$ and $\lambda_\varphi$, which ensures the unique solution $\omega^*(\theta) = -A_{\theta,\phi}^{-1}b_{\theta,\phi}$; $\lambda^*(\theta) = -A_{\theta,\phi}^{-1}b_{\theta,\phi}$. Let $R_\omega := \frac{r_{\max}}{\lambda_\phi}, R_\lambda := \frac{r_{\max}}{\lambda_\varphi}$. Then the norm of optimal solutions will be bounded as $\|\omega^*(\theta)\| \leq R_\omega, \|\lambda^*(\theta)\| \leq R_\lambda$, which justifies the projection step of the Algorithm 1.

To study the error of critic, we introduce the following notations

$$\delta^i(\xi,\theta) := r^i(s,a) + \gamma V_\theta(s') - V_\theta(s)$$
$$\delta(\xi,\theta) := \bar{r}(s,a) + \gamma V_\theta(s') - V_\theta(s)$$
$$\tilde{\delta}(\xi,\omega) := \bar{r}(s,a) + \gamma\phi(s')^T\omega - \phi(s)^T\omega$$
$$\hat{\delta}(\xi,\omega,\lambda) := \varphi(s,a)^T\lambda + \gamma\phi(s')^T\omega - \phi(s)^T\omega, \tag{23}$$

where we overwrite $V_{\pi_\theta}$ as $V_\theta$ for simplicity.

For the ease of expression, we further define

$$g_a^i(\xi,\omega,\lambda) := \hat{\delta}(\xi,\omega,\lambda)\psi_{\theta^i}(s,a^i)$$
$$g_c^i(\xi,\omega) := \delta^i(\xi,\omega)\phi(s)$$
$$\bar{g}_c(\xi,\omega) := \tilde{\delta}(\xi,\omega)\phi(s)$$
$$g_c(\theta,\omega) := \mathbb{E}_{\xi\sim\mu_\theta}[\bar{g}_c(\xi,\omega)]. \tag{24}$$

We will start with the error of averaged critic parameter first. The following lemma characterizes the descent of averaged critic variables under i.i.d. sampling.

## D.1  Error of critic

We first present several useful lemmas and propositions, which serves as the preliminary for establishing the approximate descent property of the critic variables' optimal gap.

**Proposition 1** (Lipschitz continuity of $\omega^*(\theta)$ [32]). *Suppose Assumptions 2 and 4 hold, then there exists a positive constant $L_\omega$ such that for any $\theta_1, \theta_2 \in \mathbb{R}^{Nd_\theta}$, we have*

$$\|\omega^*(\theta_1) - \omega^*(\theta_2)\| \leq L_\omega\|\theta_1 - \theta_2\|.$$

**Lemma 14** (smoothness of stationary distribution). *Suppose Assumptions 1, 3, and 4 hold, then for any $\theta, \theta' \in \mathbb{R}^d$, there exists a positive constant $L_{\mu,1}$ such that*

$$\|\nabla\mu_\theta(s) - \nabla\mu_{\theta'}(s)\| \leq L_{\mu,1}\|\theta - \theta'\|.$$

The proof of this Lemma consists of two main steps: 1) Derive the expression of the gradient and 2) establish that the gradient is Lipschitz continuous. For the first part, we follow the main idea in [2].

*Proof.* For a given policy $\pi_\theta$, we define the transition probability $P_\theta(s|s') := \sum_a \pi_\theta(a|s')P(s|s',a)$. By the Assumption 4, there exists a stationary distribution $\mu_\theta(s)$ which satisfies for all state $s$

$$\mu_\theta(s) = \sum_{s'\in\mathcal{S}} \mu_\theta(s')P_\theta(s|s') \tag{25}$$

653  Define the following notations

$$\mu_\theta := [\mu_\theta(s_1), \mu_\theta(s_2), \cdots, \mu_\theta(s_n)]^T \qquad\qquad \mathbb{R}^{|\mathcal{S}| \times 1}$$

$$P_\theta(s) := [P_\theta(s|s_1), P_\theta(s|s_2), \cdots, P_\theta(s|s_n)]^T \qquad \mathbb{R}^{|\mathcal{S}| \times 1}$$

$$P(\theta) := [P_\theta(s_1), P_\theta(s_2), \cdots, P_\theta(s_n)] \qquad\qquad \mathbb{R}^{|\mathcal{S}| \times |\mathcal{S}|}$$

$$\nabla \mu_\theta := [\nabla \mu_\theta(s_1), \nabla \mu_\theta(s_2), \cdots, \nabla \mu_\theta(s_n)] \qquad\quad \mathbb{R}^{d_\theta \times |\mathcal{S}|}$$

$$\nabla P_\theta(s) := [\nabla P_\theta(s|s_1), \nabla P_\theta(s|s_2), \cdots, \nabla P_\theta(s|s_n)] \qquad \mathbb{R}^{d_\theta \times |\mathcal{S}|}$$

654  Upon taking derivative with respect to $\theta$ on both sides of (25), we have

$$\nabla \mu_\theta(s) = \sum_{s' \in \mathcal{S}} \nabla \mu_\theta(s') P_\theta(s|s') + \mu_\theta(s') \nabla_\theta P_\theta(s|s')$$

$$= \nabla \mu_\theta P_\theta(s) + \nabla P_\theta(s) \mu_\theta \qquad\qquad\qquad (26)$$

655  (26) can be written in compact form as

$$\nabla \mu_\theta = \nabla \mu_\theta P(\theta) + [\nabla P_\theta(s_1)\mu_\theta, \cdots, \nabla P_\theta(s_n)\mu_\theta] \qquad (27)$$

656  Therefore, we have

$$[\nabla P_\theta(s_1)\mu_\theta, \cdots, \nabla P_\theta(s_n)\mu_\theta] = \nabla \mu_\theta(I - P(\theta))$$

$$= \nabla \mu_\theta(I - (P(\theta) - e\mu_\theta^T)),$$

657  where the second inequality is due to $\nabla \mu_\theta e = \nabla(\mu_\theta e) = \nabla 1 = 0$.

658  We now show that $I - (P(\theta) - e\mu_\theta^T)$ is invertible. The first step is to show $\lim_{t \to \infty}(P(\theta) - e\mu_\theta^T)^t = 0$.
659  Let $P, \mu$ represent $P(\theta), \mu_\theta$ for simplicity, we first show $(P - e\mu^T)^t = P^t - P^{t-1}e\mu^T$ by induction.
660  Observe that when $t = 1$, this is trivially satisfied. Suppose the equality holds for $t = k$, then

$$(P - e\mu^T)^{k+1} = (P^k - P^{k-1}e\mu^T)P - (P^k - P^{k-1}e\mu^T)e\mu^T$$

$$= P^{k+1} - P^{k-1}e\mu^T - P^k e\mu^T + P^{k-1}(e\mu^T)^2$$

$$= P^{k+1} - P^k e\mu^T,$$

661  where the second equality is due to (25) such that $e\mu^T P = e\mu^T$ and the last equality is due to
662  $\mu^T e = 1$.

663  Therefore, we have

$$\lim_{t \to \infty}(P(\theta) - e\mu_\theta^T)^t = \lim_{t \to \infty}(P(\theta)^t - P(\theta)^{t-1}e\mu_\theta^T) = e\mu_\theta^T - e\mu_\theta^T = 0,$$

664  which together with Lemma 12 justifies that $I - (P(\theta) - e\mu_\theta^T)$ is invertible. Thus, we have

$$\nabla \mu_\theta = (I - (P(\theta) - e\mu_\theta^T))^{-1}[\nabla P_\theta(s_1)\mu_\theta, \cdots, \nabla P_\theta(s_n)\mu_\theta]. \qquad (28)$$

665  We will utilize Lemma 11 to prove the Lipschitz property of $\nabla \mu_\theta$. We first show the Lipschitz
666  continuous of the first term. Let $A(\theta)$ to represent $I - (P(\theta) - e\mu_\theta^T)$, then we have

$$\|A(\theta_1) - A(\theta_2)\| = \|P(\theta_1) - P(\theta_2) + e(\mu_{\theta_2} - \mu_{\theta_1})^T\|$$

$$\leq \|P(\theta_1) - P(\theta_2)\| + \|e(\mu_{\theta_2} - \mu_{\theta_1})^T\|$$

$$= \sqrt{\sum_{s,s' \in \mathcal{S}} |\sum_{a \in \mathcal{A}}(\pi_{\theta_1}(a|s') - \pi_{\theta_2}(a|s'))P(s|s',a)|^2} + \sqrt{|\mathcal{S}|}\|\mu_{\theta_2} - \mu_{\theta_1}\|$$

$$\leq \sqrt{\sum_{s,s' \in \mathcal{S}} (\sum_{a \in \mathcal{A}} |(\pi_{\theta_1}(a|s') - \pi_{\theta_2}(a|s'))P(s|s',a)|)^2} + \sqrt{|\mathcal{S}|}\|\mu_{\theta_2} - \mu_{\theta_1}\|$$

$$\leq \sqrt{\sum_{s' \in \mathcal{S}} |\mathcal{A}|^2 L_\pi^2 \|\theta_1 - \theta_2\|^2 \sum_{s \in \mathcal{S}} P(s|s',a)^2} + \sqrt{|\mathcal{S}|}L_\mu\|\theta_1 - \theta_2\|$$

$$= \sqrt{|\mathcal{S}|}(|\mathcal{A}|L_\pi + L_\mu)\|\theta_1 - \theta_2\|.$$

where the second inequality uses triangle inequality. The last inequality is due to Lipschitz continuous of the policy specified in Assumption 3, and Lipschitz continuous of $\mu_\theta$ implied by Lemma 7.

To see that $A^{-1}(\theta)$ is Lipschitz continuous and bounded, observe that

$$
\begin{aligned}
\|A^{-1}(\theta_1) - A^{-1}(\theta_2)\| &= \|A^{-1}(\theta_2)(A(\theta_2) - A(\theta_1))A^{-1}(\theta_1)\| \\
&\leq \|A^{-1}(\theta_2)\|\|A^{-1}(\theta_1)\|\|A(\theta_2) - A(\theta_1)\| \\
&\leq \sqrt{|\mathcal{S}|}(|\mathcal{A}|L_\pi + L_\mu)\|A^{-1}(\theta_2)\|\|A^{-1}(\theta_1)\|\|\theta_2 - \theta_1\|, \quad (29)
\end{aligned}
$$

where the first inequality uses Cauchy-Schwartz inequality, and the last inequality uses the Lipschitz continuous of $A(\theta)$ in (29). Since $\|A(\theta)\|$ is bounded, $\|A^{-1}(\theta)\|$ is also bounded (due to invertibility), which justifies that the first term in (28) is Lipschitz continuous and bounded.

We now consider the second term in (28). For any state $s$

$$
\begin{aligned}
\|\nabla P_{\theta_1}(s)\mu_{\theta_1} - \nabla P_{\theta_2}(s)\mu_{\theta_2}\| &= \|\nabla P_{\theta_1}(s)(\mu_{\theta_1} - \mu_{\theta_2}) + (\nabla P_{\theta_1}(s) - \nabla P_{\theta_2}(s))\mu_{\theta_2}\| \\
&\leq \|\nabla P_{\theta_1}(s)(\mu_{\theta_1} - \mu_{\theta_2})\| + \|(\nabla P_{\theta_1}(s) - \nabla P_{\theta_2}(s))\mu_{\theta_2}\| \\
&\leq \|\nabla P_{\theta_1}(s)\|\|\mu_{\theta_1} - \mu_{\theta_2}\| + \|\nabla P_{\theta_1}(s) - \nabla P_{\theta_2}(s)\|\|\mu_{\theta_2}\| \\
&\leq \sum_{s'\in\mathcal{S}}\sum_{a\in\mathcal{A}}\|\nabla\pi_{\theta_1}(a|s')P(s|s',a)\|L_\mu\|\theta_1 - \theta_2\| \\
&\quad + \sum_{s'\in\mathcal{S}}\sum_{a\in\mathcal{A}}\|(\nabla\pi_{\theta_1}(a|s') - \nabla\pi_{\theta_2}(a|s'))P(s|s',a)\| \\
&\leq |\mathcal{S}||\mathcal{A}|(C_\pi L_\mu + L_\pi)\|\theta_1 - \theta_2\|,
\end{aligned}
$$

which justifies the Lipschitz continuous of $\nabla P_\theta(s)\mu_\theta$. Define $B(\theta) := [\nabla P_\theta(s_1)\mu_\theta, \cdots, \nabla P_\theta(s_n)\mu_\theta]$, we have

$$
\|B(\theta_1) - B(\theta_2)\| \leq |\mathcal{S}|^{3/2}|\mathcal{A}|(C_\pi L_\mu + L_\pi)\|\theta_1 - \theta_2\|.
$$

Since $\nabla\mu_\theta = A^{-1}(\theta)B(\theta)$, with $A^{-1}(\theta)$ and $B(\theta)$ being Lipschitz continuous and bounded. Therefore, according to Lemma 11, there exists a positive constant $L_{\mu,1}$ which satisfies

$$
\|\nabla\mu_{\theta_1} - \nabla\mu_{\theta_2}\| \leq L_{\mu,1}\|\theta_1 - \theta_2\|.
$$

$\square$

**Proposition 2** (Lipschitz continuity of $\nabla_\theta\omega^*(\theta)$ [4]). *Suppose Assumptions 1-4 hold, then there exists a positive constant $L_{\omega,2}$ such that*

$$
\|\nabla_\theta\omega^*(\theta_1) - \nabla_\theta\omega^*(\theta_2)\|_F \leq L_{\omega,2}\|\theta_1 - \theta_2\|.
$$

*Proof.* The proof follows the derivation of Proposition 8 of [4]. However, they make assumption that $\mu_\theta(s)$ is Lipschitz continuous, which we have justified in Lemma 14. We present the proof for the completeness.

We have $\omega^*(\theta) = -A_{\theta,\phi}^{-1}b_{\theta,\phi}$, where $A_{\theta,\phi}$ is defined in (22). The Jacobian of $\omega^*(\theta)$ can be calculated as

$$
\begin{aligned}
\nabla_\theta\omega^*(\theta) &= -\nabla_\theta(A_{\theta,\phi}^{-1}b_{\theta,\phi}) \\
&= -A_{\theta,\phi}^{-1}(\nabla_\theta A_{\theta,\phi})A_{\theta,\phi}^{-1}b_{\theta,\phi} - A_{\theta,\phi}(\nabla_\theta b_{\theta,\phi}). \quad (30)
\end{aligned}
$$

We can utilize Lemma 11 to show the Lipschitz continuity of $\nabla\omega^*(\theta)$. We have to verify the Lipschitz continuity and boundedness of $A_{\theta,\phi}^{-1}$, $b_{\theta,\phi}$, $\nabla_\theta A_{\theta,\phi}$, and $\nabla_\theta b_{\theta,\phi}$.

The Lipschitz continuity and boundedness of $A_{\theta,\phi}^{-1}$ has been shown in (29). Let $b_1$ and $b_2$ represent $b_{\theta_1,\phi}, b_{\theta_2,\phi}$, we have

$$
\begin{aligned}
\|b_1 - b_2\| &= \|\mathbb{E}[\bar{r}(s,a,s')\phi(s)] - \mathbb{E}[r(\tilde{s},\tilde{a},\tilde{s}')\phi(\tilde{s})]\| \\
&\leq \sup_{s,a,s'}\|r(s,a,s')\phi(s)\|\|\mathbb{P}((s,a,s'\in\cdot)) - \mathbb{P}((\tilde{s},\tilde{a},\tilde{s}'\in\cdot))\|_{TV} \\
&\leq r_{\max}\|\mathbb{P}((s,a,s'\in\cdot)) - \mathbb{P}((\tilde{s},\tilde{a},\tilde{s}'\in\cdot))\|_{TV} \\
&\leq 2|\mathcal{A}|L_\pi(1 + \log_\rho\kappa^{-1} + (1-\rho)^{-1})\|\theta_1 - \theta_2\|,
\end{aligned}
$$

690 where the last inequality follows Lemma 8.

691 We now analyze $\nabla_\theta A_{\theta,\phi}$. We first define

$$A(s, s') := \phi(s)(\gamma\phi(s') - \phi(s))^T, \quad b(s, a, s') := r(s, a, s')\phi(s).$$

692 as

$$\nabla_\theta A_{\theta,\phi} = \nabla_\theta \left( \sum_{s,a,s'} \mu_\theta(s)\pi_\theta(a|s)P(s'|s,a)A(s,s') \right)$$

$$= \sum_{s,a,s'} \left[ \nabla_\theta\mu_\theta(s)\pi_\theta(a|s)P(s'|s,a)A(s,s') + \mu_\theta\nabla_\theta\pi_\theta(a|s)P(s'|s,a)A(s,s') \right].$$

693 By Lemma 14 and Lemma 8, and Assumption 3, $\mu_\theta(s)$, $\pi_\theta(a|s)$, $\nabla_\theta\mu_\theta(s)$, $\nabla_\theta\pi_\theta(a|s)$ are Lipschitz
694 continuous and bounded. Therefore, $\nabla_\theta A_{\theta,\phi}$ is Lipschitz and bounded.

695 Finally, we analyze $\nabla_\theta b_{\theta,\phi}$ by following the same technique.

$$\nabla_\theta b_{\theta,\phi} = \nabla_\theta \left( \sum_{s,a,s'} \mu_\theta(s)\pi_\theta(a|s)P(s'|s,a)b(s,a,s') \right)$$

$$= \sum_{s,a,s'} \left[ \nabla_\theta\mu_\theta(s)\pi_\theta(a|s)P(s'|s,a)b(s,a,s') + \mu_\theta(s)\nabla_\theta\pi_\theta(a|s)P(s'|s,a)b(s,a,s') \right].$$

696 By Lemma 14 and Lemma 8, and Assumption 3, $\mu_\theta(s)$, $\pi_\theta(a|s)$, $\nabla_\theta\mu_\theta(s)$, $\nabla_\theta\pi_\theta(a|s)$ are Lipschitz
697 continuous and bounded. Thus, $\nabla_\theta b_{\theta,\phi}$ is bounded and Lipschitz continuous.

698 We have shown the Lipschitz continuity and boundedness of $A_{\theta,\phi}^{-1}$, $b_{\theta,\phi}$, $\nabla_\theta A_{\theta,\phi}$, and $\nabla_\theta b_{\theta,\phi}$.
699 Therefore, by applying Lemma 11, we conclude that there exists a positive constant $L_{\omega,2}$ such that
700 $\nabla_\theta\omega^*(\theta)$ in (30) is $L_{\omega,2}$-Lipschitz continuous. $\square$

701 **Lemma 15** (descent of critic's optimal gap (i.i.d. sampling))**.** *Suppose Assumptions 1-4 hold, with*
702 *$\omega_{k+1}$ generated by Algorithm 1 given $\omega_k$ and $\theta_k$ under i.i.d. sampling, then the following holds*

$$\mathbb{E}\|\bar{\omega}_{k+1} - \omega^*(\theta_{k+1})\|^2 \leq (1 + 4L_{\omega,2}^2 N\alpha_k + \frac{L_{\omega,2}^2}{2}C_\theta^2 N^2\alpha_k^2)\mathbb{E}\|\bar{\omega}_{k+1} - \omega^*(\theta_k)\|^2$$

$$+ (\frac{L_{\omega,2}^2}{2}C_\theta^2 N^2 + L_\omega^2 C_\theta^2 N^2)\alpha_k^2 + \frac{\alpha_k}{4}\sum_{i=1}^N \|\mathbb{E}[g_a^i(\xi_k, \omega_{k+1}^i, \lambda_{k+1}^i)]\|^2. \tag{31}$$

703
$$\mathbb{E}\|\bar{\omega}_{k+1} - \omega^*(\theta_k)\|^2 \leq (1 - 2\lambda_\phi\beta_k)\mathbb{E}\|\bar{\omega}_k - \omega^*(\theta_k)\|^2 + C_\delta^2\beta_k^2. \tag{32}$$

704 *Proof.* We begin with the optimality gap of averaged critic variables

$$\|\bar{\omega}_{k+1} - \omega^*(\theta_{k+1})\|^2$$
$$= \|\bar{\omega}_{k+1} - \omega^*(\theta_k) + \omega^*(\theta_k) - \omega^*(\theta_{k+1})\|^2$$
$$= \|\bar{\omega}_{k+1} - \omega^*(\theta_k)\|^2 + \|\omega^*(\theta_k) - \omega^*(\theta_{k+1})\|^2 + 2\langle\bar{\omega}_{k+1} - \omega^*(\theta_k), \omega^*(\theta_k) - \omega^*(\theta_{k+1})\rangle$$
$$\leq \|\bar{\omega}_{k+1} - \omega^*(\theta_k)\|^2 + N^2 L_\omega^2 C_\theta^2\alpha_k^2 + 2\langle\bar{\omega}_{k+1} - \omega^*(\theta_k), \nabla\omega^*(\theta_k)^T(\theta_k - \theta_{k+1})\rangle$$
$$+ 2\langle\bar{\omega}_{k+1} - \omega^*(\theta_k), \omega^*(\theta_k) - \omega^*(\theta_{k+1}) - \nabla\omega^*(\theta_k)^T(\theta_k - \theta_{k+1})\rangle, \tag{33}$$

705 where the inequality is due to

$$\|\omega^*(\theta_k) - \omega^*(\theta_{k+1})\|^2 \leq L_\omega\|\theta_k - \theta_{k+1}\|^2,$$

$$\|\theta_k - \theta_{k+1}\|^2 = \|\sum_{i=1}^N \hat{\delta}(\xi_k, \omega_k^i, \lambda_k^i)\psi_{\theta_k^i}(s_k, a_k^i)\|^2 \leq N^2\alpha_k^2 C_\theta^2, \tag{34}$$

706 with $C_\theta := C_\delta C_\psi$.

The third term in (33) can be bounded as

$$\langle\bar{\omega}_{k+1}-\omega^*(\theta_k),\nabla\omega^*(\theta_k)^T(\theta_k-\theta_{k+1})\rangle$$
$$\leq \|\bar{\omega}_{k+1}-\omega^*(\theta_k)\|\|\nabla\omega^*(\theta_k)^T(\theta_k-\theta_{k+1})\|$$
$$\leq L_{\omega,2}\|\bar{\omega}_{k+1}-\omega^*(\theta_k)\|\|\theta_k-\theta_{k+1}\|$$
$$\leq \sum_{i=1}^{N}L_{\omega,2}\alpha_k\|\bar{\omega}_{k+1}-\omega^*(\theta_k)\|\|g_a^i(\xi_k,\omega_{k+1}^i,\lambda_{k+1}^i)\|$$
$$\leq \sum_{i=1}^{N}(2L_{\omega,2}\alpha_k\|\bar{\omega}_{k+1}-\omega^*(\theta_k)\|^2+\frac{\alpha_k}{8}\|g_a^i(\xi_k,\omega_{k+1}^i,\lambda_{k+1}^i)\|^2), \tag{35}$$

where the second inequality follows Proposition 1, the third inequality uses triangle inequality, and the last inequality uses Young's inequality.

The last term in (33) can be bounded as

$$\mathbb{E}\langle\bar{\omega}_{k+1}-\omega^*(\theta_k),\omega^*(\theta_k)-\omega^*(\theta_{k+1})-\nabla\omega^*(\theta_k)^T(\theta_k-\theta_{k+1})\rangle$$
$$\leq \frac{L_{\omega,2}^2}{2}\mathbb{E}\|\bar{\omega}_{k+1}-\omega^*(\theta_k)\|\|\theta_{k+1}-\theta_k\|^2$$
$$\leq \frac{L_{\omega,2}^2}{4}\mathbb{E}\|\bar{\omega}_{k+1}-\omega^*(\theta_k)\|^2\|\theta_{k+1}-\theta_k\|^2+\frac{L_{\omega,2}^2}{4}\|\theta_{k+1}-\theta_k\|^2$$
$$\leq \frac{L_{\omega,2}^2}{4}N^2C_\theta^2\alpha_k^2\mathbb{E}\|\bar{\omega}_{k+1}-\omega^*(\theta_k)\|^2+\frac{L_{\omega,2}^2}{4}N^2C_\theta^2\alpha_k^2. \tag{36}$$

The first inequality uses Lemma 10, and the second inequality is induced by Young's inequality. The last inequality follows (34).

Plug (35) and (36) into (33) will yield (31).

We now prove (32).

$$\|\bar{\omega}_{k+1}-\omega^*(\theta_k)\|^2 = \|\prod_{R_\omega}(\bar{\omega}_k+\beta_k\bar{g}_c(\xi_k,\bar{\omega}_k))-\prod_{R_\omega}\omega^*(\theta_k)\|^2$$
$$\leq \|\bar{\omega}_k+\beta_k\bar{g}_c(\xi,\bar{\omega}_k)-\omega^*(\theta_k)\|^2$$
$$= \|\bar{\omega}_k-\omega^*(\theta_k)\|^2+\beta_k^2\|\bar{g}_c(\xi_k,\bar{\omega}_k)\|^2+2\beta_k\mathbb{E}[\langle\bar{\omega}_k-\omega^*(\theta_k),\bar{g}_c(\xi_k,\bar{\omega}_k)\rangle]$$
$$\leq \|\bar{\omega}_k-\omega^*(\theta_k)\|^2+\beta_k^2C_\delta^2+2\beta_k\langle\bar{\omega}_k-\omega^*(\theta_k),\bar{g}_c(\xi_k,\bar{\omega}_k)\rangle. \tag{37}$$

The first inequality is due to the non-expansiveness of projection to convex set. The last inequality follows

$$\|\bar{g}_c(\xi,\omega)\|\leq|r(s,a)+\gamma\phi(s')^T\omega-\phi(s)^T\omega|\leq r_{\max}+(1+\gamma)R_\omega := C_\delta.$$

Let $\xi\sim\mu_\theta$ to represent $s\sim\mu_{\pi_\theta}, a\sim\pi_\theta(\cdot|s), s'\sim\mathcal{P}(\cdot|s,a)$, the last term in (37) can be bounded as

$$\mathbb{E}[\langle\bar{\omega}_k-\omega^*(\theta_k),\bar{g}_c(\xi_k,\bar{\omega}_k)\rangle]$$
$$= \langle\bar{\omega}_k-\omega^*(\theta_k),\mathbb{E}[\bar{g}_c(\xi_k,\bar{\omega}_k)-g_c(\theta_k,\omega^*(\theta_k))]\rangle$$
$$= \beta_k\langle\bar{\omega}_k-\omega^*(\theta_k),\mathbb{E}_{\xi\sim\mu_{\theta_k}}[\phi(s)(\gamma\phi(s')-\phi(s))^T|\theta_k](\bar{\omega}_k-\omega^*(\theta_k))\rangle$$
$$= \beta_k\langle\bar{\omega}_k-\omega^*(\theta_k),A_{\theta_k,\phi}(\bar{\omega}_k-\omega^*(\theta_k))\rangle$$
$$\leq -\lambda_\phi\beta_k\|\bar{\omega}_k-\omega^*(\theta_k)\|^2. \tag{38}$$

Here the first equality is due to critic's optimality condition $g_c(\theta_k,\omega^*(\theta_k)) = \mathbb{E}_{\xi_k\sim\mu_{\theta_k}}[\bar{g}_c(\xi_k,\omega^*(\theta_k))|\theta_k] = 0$. The last inequality uses the negative definiteness of $A_{\theta_k,\phi}$. Plug (38) into (37) gives us (36). $\square$

The next lemma describes the descent property of averaged critic variables under Markovian sampling.

722 **Lemma 16** (descent of critic's optimal gap (Markovian sampling)). *Under Assumptions 1-4, with*
723 $\omega_{k+1}$ *generated by Algorithm 1 given $\omega_k$ and $\theta_k$ under Markovian sampling, then the following holds*

$$
\mathbb{E}\|\bar{\omega}_{k+1} - \omega^*(\theta_{k+1})\|^2 \leq (1 + 4L_{\omega,2}^2 N\alpha_k + \frac{L_{\omega,2}^2}{2}C_\theta^2 N^2\alpha_k^2)\mathbb{E}\|\bar{\omega}_{k+1} - \omega^*(\theta_k)\|^2
$$
$$
+ (\frac{L_{\omega,2}^2}{2}C_\theta^2 N^2 + L_\omega^2 C_\theta^2 N^2)\alpha_k^2 + \frac{\alpha_k}{4}\sum_{i=1}^N \|\mathbb{E}[g_a^i(\xi_k, \omega_{k+1}^i, \lambda_{k+1}^i)]\|^2.
$$
(39)

724

$$
\mathbb{E}\|\bar{\omega}_{k+1} - \omega^*(\theta_k)\|^2 \leq (1 - 2\lambda_\phi\beta_k)\mathbb{E}\|\bar{\omega}_k - \omega^*(\theta_k)\|^2 + C_{K_1}\beta_k\beta_{k-Z_K} + C_{K_2}\alpha_{k-Z_K}\beta_k. \quad (40)
$$

725 *where* $C_{K_1} := 4C_2C_\delta Z_K + C_\delta^2$, $C_{K_2} := 4C_1C_\theta Z_K + 2C_3C_\theta Z_K^2 + C_8, Z_K := \min\{z \in$
726 $\mathbb{N}^+|\kappa\rho^{z-1} \leq \min\{\alpha_k, \beta_k, \eta_k\}\}$.

727 *Proof.* (39) has already been derived in the proof of i.i.d. sampling setting, please check the derivation
728 of (31).

729 We now prove (40). Follow the derivation of (37), we have

$$
\mathbb{E}\|\bar{\omega}_{k+1} - \omega^*(\theta_k)\|^2 \leq \|\bar{\omega}_k - \omega^*(\theta_k)\|^2 + \beta_k^2 C_\delta^2 + 2\beta_k\mathbb{E}[\langle\bar{\omega}_k - \omega^*(\theta_k), \bar{g}_c(\xi_k, \bar{\omega}_k)\rangle]
$$
$$
= \|\bar{\omega}_k - \omega^*(\theta_k)\|^2 + \beta_k^2 C_\delta^2 + 2\beta_k\mathbb{E}\langle\bar{\omega}_k - \omega^*(\theta_k), g_c(\theta_k, \bar{\omega}_k)\rangle
$$
$$
+ 2\beta_k\mathbb{E}\langle\bar{\omega}_k - \omega^*(\theta_k), \bar{g}_c(\xi_k, \bar{\omega}_k) - g_c(\theta_k, \bar{\omega}_k)\rangle
$$
$$
\leq (1 - 2\lambda_\phi\beta_k)\|\bar{\omega}_k - \omega^*(\theta_k)\|^2 + \beta_k^2 C_\delta^2
$$
$$
+ 2\beta_k\mathbb{E}\langle\bar{\omega}_k - \omega^*(\theta_k), \bar{g}_c(\xi_k, \bar{\omega}_k) - g_c(\theta_k, \bar{\omega}_k)\rangle. \quad (41)
$$

730 Here, the last inequality bound the third term using the same technique of (38).

731 We now bound the last term in (41). By Lemma 17, for any $z \in \mathbb{N}^+$, we have

$$
\mathbb{E}\langle\bar{\omega}_k - \omega^*(\theta_k), \bar{g}_c(\xi_k, \bar{\omega}_k) - g_c(\theta_k, \bar{\omega}_k)\rangle
$$

$$
\leq C_1\mathbb{E}\|\theta_k - \theta_{k-z}\| + C_2\mathbb{E}\|\bar{\omega}_k - \bar{\omega}_{k-z}\| + C_3\sum_{m=0}^{z-1}\mathbb{E}\|\theta_{k-m} - \theta_{k-z}\| + C_8\kappa\rho^{z-1}
$$

$$
\overset{(i)}{\leq} C_1\sum_{n=1}^z \mathbb{E}\|\theta_{k-n+1} - \theta_{k-n}\| + C_2\sum_{n=1}^z \mathbb{E}\|\bar{\omega}_{k-n+1} - \bar{\omega}_{k-n}\|
$$

$$
+ C_3\sum_{m=0}^{z-1}\sum_{n=1}^{z-m}\mathbb{E}\|\theta_{k-m-n+1} - \theta_{k-m-n}\| + C_8\kappa\rho^{z-1}
$$

$$
\leq 2C_1C_\theta\sum_{n=1}^z \alpha_{k-n} + 2C_2C_\delta\sum_{n=1}^z \beta_{k-n} + C_3C_\theta\sum_{m=0}^{z-1}\sum_{n=1}^{z-m}\alpha_{k-m-n} + C_8\kappa\rho^{z-1}
$$

$$
\overset{(ii)}{\leq} 2C_1C_\theta z\alpha_{k-z} + 2C_2C_\delta z\beta_{k-z} + C_3C_\theta z(z-1)\alpha_{k-z} + C_8\kappa\rho^{z-1}, \quad (42)
$$

732 where the $(i)$ uses triangle inequality, $(ii)$ uses the non-increasing property of step sizes.

733 Let $z = Z_K := \min\{z \in \mathbb{N}^+|\kappa\rho^{z-1} \leq \min\{\alpha_k, \beta_k, \eta_k\}\}$, we have

$$
\mathbb{E}\langle\bar{\omega}_k - \omega^*(\theta_k), \bar{g}_c(\xi_k, \bar{\omega}_k) - g_c(\theta_k, \bar{\omega}_k)\rangle
$$
$$
\leq 2C_1C_\theta Z_K\alpha_{k-Z_K} + 2C_2C_\delta Z_K\beta_{k-Z_K} + C_3C_\theta Z_K^2\alpha_{k-Z_K} + C_8\alpha_{k-Z_K}. \quad (43)
$$

734 Plug (43) into (41) will yield

$$
\|\bar{\omega}_{k+1} - \omega^*(\theta_k)\|^2 \leq (1 - 2\lambda_\phi\beta_k)\|\bar{\omega}_k - \omega^*(\theta_k)\|^2 + C_\delta^2\beta_k^2
$$
$$
+ 4C_1C_\theta Z_K\alpha_{k-Z_K} + 4C_2C_\delta Z_K\beta_{k-Z_K} + 2C_3C_\theta Z_K^2\alpha_{k-Z_K} + 2C_8\alpha_{k-Z_K}.
$$

735 By defining $C_{K_1} := 4C_2C_\delta Z_K + C_\delta^2$, $C_{K_2} := 4C_1C_\theta Z_K + 2C_3C_\theta Z_K^2 + C_8$, we complete the
736 proof. $\qquad\square$

**Lemma 17.** *Consider the sequence generated by Algorithm 1, for any $z \in \mathbb{N}^+$, we have*

$$\mathbb{E}\langle \bar{\omega}_k - \omega^*(\theta_k), \bar{g}_c(\xi_k, \bar{\omega}_k) - g_c(\theta_k, \bar{\omega}_k)\rangle \leq C_1\|\theta_k - \theta_{k-z}\| + C_2\|\bar{\omega}_k - \bar{\omega}_{k-z}\|$$

$$+ C_3 \sum_{m=0}^{z-1} \|\theta_{k-m} - \theta_{k-z}\| + C_8\kappa\rho^{z-1},$$

*where $C_1 := 4R_\omega C_\delta|\mathcal{A}|L_\pi(1 + \log_\rho \kappa^{-1} + (1-\rho)^{-1}) + 2C_\delta L_\omega$, $C_2 := 4(1+\gamma)R_\omega + 2C_\delta$, $C_3 := 4R_\omega C_\delta|\mathcal{A}|L_\pi$, $C_8 := 8R_\omega C_\delta$.*

*Proof.* Consider the Markov chain since timestep $k - z$:

$$s_{k-z} \xrightarrow{\theta_{k-z}} a_{k-z} \xrightarrow{\mathcal{P}} s_{k-z+1} \xrightarrow{\theta_{k-z+1}} a_{k-z+1} \cdots \xrightarrow{\theta_{k-1}} a_{k-1} \xrightarrow{\mathcal{P}} s_k \xrightarrow{\theta_k} a_k \xrightarrow{\mathcal{P}} s_{k+1}.$$

Also consider the auxiliary Markov chain with fixed policy since timestep $k - z$:

$$s_{k-z} \xrightarrow{\theta_{k-z}} a_{k-z} \xrightarrow{\mathcal{P}} s_{k-z+1} \xrightarrow{\theta_{k-z}} \tilde{a}_{k-z+1} \cdots \xrightarrow{\theta_{k-z}} \tilde{a}_{k-1} \xrightarrow{\mathcal{P}} \tilde{s}_k \xrightarrow{\theta_{k-z}} \tilde{a}_k \xrightarrow{\mathcal{P}} \tilde{s}_{k+1}.$$

Throughout the proof of this lemma, we will use $\theta, \theta', \bar{\omega}, \bar{\omega}', \xi, \tilde{\xi}$ as shorthand notations of $\theta_k, \theta_{k-z}, \bar{\omega}_k, \bar{\omega}_{k-z}, \xi_k, \tilde{\xi}_k$.

For the ease of expression, define

$$\Delta_1(\xi, \theta, \omega) := \langle \omega - \omega^*(\theta), \bar{g}_c(\xi, \omega) - g_c(\theta, \omega)\rangle.$$

Therefore, we have

$$\langle \bar{\omega}_k - \omega^*(\theta_k), \bar{g}_c(\xi_k, \bar{\omega}_k) - g_c(\theta_k, \bar{\omega}_k)\rangle = \Delta_1(\xi, \theta, \bar{\omega})$$

$$= \underbrace{\Delta_1(\xi, \theta, \bar{\omega}) - \Delta_1(\xi, \theta', \bar{\omega})}_{I_1} + \underbrace{\Delta_1(\xi, \theta', \bar{\omega}) - \Delta_1(\xi, \theta', \bar{\omega}')}_{I_2}$$

$$+ \underbrace{\Delta_1(\xi, \theta', \bar{\omega}') - \Delta_1(\tilde{\xi}, \theta', \bar{\omega}')}_{I_3} + \underbrace{\Delta_1(\tilde{\xi}, \theta', \bar{\omega}')}_{I_4}. \quad (44)$$

$I_1$ can be expressed as

$$I_1 = \langle \bar{\omega} - \omega^*(\theta), \bar{g}_c(\xi, \bar{\omega}) - g_c(\theta, \bar{\omega})\rangle - \langle \bar{\omega} - \omega^*(\theta'), \bar{g}_c(\xi, \bar{\omega}) - g_c(\theta', \bar{\omega})\rangle$$

$$= \langle \bar{\omega} - \omega^*(\theta), \bar{g}_c(\xi, \bar{\omega}) - g_c(\theta, \bar{\omega})\rangle - \langle \bar{\omega} - \omega^*(\theta), \bar{g}_c(\xi, \bar{\omega}) - g_c(\theta', \bar{\omega})\rangle$$

$$+ \langle \omega^*(\theta) - \omega^*(\theta'), \bar{g}_c(\xi, \bar{\omega}) - g_c(\theta', \bar{\omega})\rangle$$

$$\leq \|\bar{\omega} - \omega^*(\theta)\|\|g_c(\theta', \bar{\omega}) - g_c(\theta, \bar{\omega})\| + \|\omega^*(\theta) - \omega^*(\theta')\|\|\bar{g}_c(\xi, \bar{\omega}) - g_c(\theta', \bar{\omega})\|. \quad (45)$$

The first term can be bounded as

$$\|\bar{\omega} - \omega^*(\theta)\|\|g_c(\theta', \bar{\omega}) - g_c(\theta, \bar{\omega})\| \leq 2R_\omega\|\mathbb{E}_{\xi\sim\mu'_\theta}[\bar{g}_c(\xi, \bar{\omega})] - \mathbb{E}_{\xi\sim\mu_\theta}[\bar{g}_c(\xi, \bar{\omega})]\|$$

$$\leq 4R_\omega \sup_\xi \|\bar{g}_c(\xi, \bar{\omega})\| d_{TV}(\mu'_\theta \otimes \pi'_\theta \otimes \mathcal{P}, \mu_\theta \otimes \pi_\theta \otimes \mathcal{P})$$

$$\leq 4R_\omega C_\delta d_{TV}(\mu'_\theta \otimes \pi'_\theta \otimes \mathcal{P}, \mu_\theta \otimes \pi_\theta \otimes \mathcal{P})$$

$$\leq 4R_\omega C_\delta|\mathcal{A}|L_\pi(1 + \log_\rho \kappa^{-1} + (1-\rho)^{-1})\|\theta - \theta'\|, \quad (46)$$

where the first inequality follows the projection update of each critic step, the third inequality is due to $\|\bar{g}_c(\xi, \bar{\omega})\| \leq C_\delta$, and the last inequality follows Lemma 8.

By the Lipschitz conitinuous of $\omega^*(\theta)$ proposed in Proposition 1, the second term can be bounded as

$$\|\omega^*(\theta) - \omega^*(\theta')\|\|\bar{g}_c(\xi, \bar{\omega}) - g_c(\theta, \bar{\omega})\| \leq 2C_\delta L_\omega\|\theta - \theta'\| \quad (47)$$

Plug (46) and (47) into (45), we can bound $I_1$ as

$$I_1 \leq (4R_\omega C_\delta|\mathcal{A}|L_\pi(1 + \log_\rho \kappa^{-1} + (1-\rho)^{-1}) + 2C_\delta L_\omega)\|\theta - \theta'\|. \quad (48)$$

Next we bound $I_2$ as

$$I_2 = \langle \bar{\omega} - \omega^*(\theta'), \bar{g}_c(\xi, \bar{\omega}) - g_c(\theta', \bar{\omega}) \rangle - \langle \bar{\omega}' - \omega^*(\theta'), \bar{g}_c(\xi, \bar{\omega}') - g_c(\theta', \bar{\omega}') \rangle$$
$$= \langle \bar{\omega} - \omega^*(\theta'), \bar{g}_c(\xi, \bar{\omega}) - g_c(\theta', \bar{\omega}) \rangle - \langle \bar{\omega}' - \omega^*(\theta'), \bar{g}_c(\xi, \bar{\omega}) - g_c(\theta', \bar{\omega}) \rangle$$
$$+ \langle \bar{\omega}' - \omega^*(\theta'), \bar{g}_c(\xi, \bar{\omega}) - \bar{g}_c(\xi, \bar{\omega}') - g_c(\theta', \bar{\omega}) + g_c(\theta', \bar{\omega}') \rangle.$$

The first two terms can be bounded as

$$\langle \bar{\omega} - \bar{\omega}', \bar{g}_c(\xi, \bar{\omega}) - g_c(\theta', \bar{\omega}) \rangle \le 2C_\delta \|\bar{\omega} - \bar{\omega}'\|. \tag{49}$$

The last term can be bounded as

$$\langle \bar{\omega}' - \omega^*(\theta'), \bar{g}_c(\xi, \bar{\omega}) - \bar{g}_c(\xi, \bar{\omega}') - g_c(\theta', \bar{\omega}) + g_c(\theta', \bar{\omega}') \rangle$$
$$\le \|\bar{\omega} - \omega^*(\theta')\|(\|\bar{g}_c(\xi, \bar{\omega}) - \bar{g}_c(\xi, \bar{\omega}')\| + \|g_c(\theta', \bar{\omega}') - g_c(\theta', \bar{\omega})\|)$$
$$\le 2R_\omega(\|\bar{g}_c(\xi, \bar{\omega}) - \bar{g}_c(\xi, \bar{\omega}')\| + \|g_c(\theta', \bar{\omega}') - g_c(\theta', \bar{\omega})\|)$$
$$\le 4R_\omega(1 + \gamma)\|\bar{\omega} - \bar{\omega}'\|, \tag{50}$$

where the second inequality follows the projection of each critic step. The last inequality is due to

$$\|\bar{g}_c(\xi, \bar{\omega}) - \bar{g}_c(\xi, \bar{\omega}')\| = \|\phi(s)(\gamma\phi(s')^T(\bar{\omega} - \bar{\omega}') - \phi(s)^T(\bar{\omega} - \bar{\omega}'))\|$$
$$\le \gamma\|\phi(s')^T(\bar{\omega} - \bar{\omega}')\| + \|\phi(s)^T(\bar{\omega} - \bar{\omega}')\|$$
$$\le (1 + \gamma)\|\bar{\omega} - \bar{\omega}'\|.$$

Combine (49) and (50), we can bound $I_2$ as

$$I_2 \le (4(1 + \gamma)R_\omega + 2C_\delta)\|\bar{\omega} - \bar{\omega}'\|. \tag{51}$$

We bound $I_3$ as

$$\mathbb{E}[I_3|\theta', s_{k-z+1}] = \mathbb{E}[\Delta_1(\xi, \theta', \bar{\omega}') - \Delta_1(\tilde{\xi}, \theta', \bar{\omega}')|\theta', s_{k-z+1}]$$
$$\le 2\sup_\xi |\Delta_1(\xi, \theta', \bar{\omega}')| \, d_{TV}(\mathbb{P}(\xi \in \cdot|\theta', s_{k-z+1}), \mathbb{P}(\tilde{\xi} \in \cdot|\theta', s_{k-z+1}))$$
$$\le 8R_\omega C_\delta d_{TV}(\mathbb{P}(\xi \in \cdot|\theta', s_{k-z+1}), \mathbb{P}(\tilde{\xi} \in \cdot|\theta', s_{k-z+1}))$$
$$\le 4R_\omega C_\delta |\mathcal{A}| L_\pi \sum_{m=0}^{z-1} \|\theta_{k-m} - \theta_{k-z}\|. \tag{52}$$

Here, the second inequality is due to $\|\Delta_1(\xi, \theta', \bar{\omega}')\| \le \|\omega' - \omega^*(\theta')\|\|\bar{g}_c(\xi, \omega') - g_c(\theta', \omega')\| \le 4R_\omega C_\delta$, and the last inequality is according to Lemma 13.

We now bound $I_4$

$$\mathbb{E}[I_4|\theta', \bar{\omega}', s_{k+z-1}] = \mathbb{E}[\Delta_1(\tilde{\xi}, \theta', \bar{\omega}')|\theta', \bar{\omega}', s_{k-z+1}]$$
$$\le \sup_\xi |\Delta_1(\xi, \theta', \bar{\omega}')| \|\mathbb{P}(\xi \in \cdot|\theta', s_{k-z+1}) - \mu_{\theta'} \otimes \pi_{\theta'} \otimes \mathcal{P}\|$$
$$\le 8R_\omega C_\delta d_{TV}(\mathbb{P}(\tilde{x} \in \cdot|\theta', s_{t-z+1}), \mu_{\theta'} \otimes \pi_{\theta'} \otimes \mathcal{P})$$
$$\le 8R_\omega C_\delta \kappa \rho^{z-1}, \tag{53}$$

where the last inequality follows Lemma 5.

Plug (48), (51), (52), and (53) into (44), we get

$$\mathbb{E}[\Delta_1(\xi, \theta, \bar{\omega})] \le (4R_\omega C_\delta |\mathcal{A}| L_\pi (1 + \log_\rho \kappa^{-1} + (1 - \rho)^{-1}) + 2C_\delta L_\omega)\mathbb{E}\|\theta_k - \theta_{k-z}\|$$
$$+ (4(1 + \gamma)R_\omega + 2C_\delta)\mathbb{E}\|\bar{\omega}_k - \bar{\omega}_{k-z}\|$$
$$+ (4R_\omega C_\delta |\mathcal{A}| L_\pi) \sum_{m=0}^{z-1} \mathbb{E}\|\theta_{k-m} - \theta_{k-z}\|$$
$$+ (8R_\omega C_\delta)\kappa \rho^{z-1},$$

which completes the proof. $\qquad \square$

 **D.2   Error of reward estimator**

765 The analysis for the error of reward estimator is similar to critic. To see this, we only need to change
766 $\bar{g}_c(\xi, \bar{\omega})$ into $\bar{g}_r(\xi, \bar{\lambda}) := (r(s, a) - \varphi(s, a)^T \bar{\lambda})\varphi(s, a)$ to recover most of the proofs. We provide the
767 reward estimator's analysis for the completeness. For the ease of discussion, we define

$$g_r^i(\xi, \lambda) := \varphi(s, a)(r^i(s, a) - \varphi(s, a)^T \lambda),$$
$$\bar{g}_r(\xi, \lambda) := \varphi(s, a)(\bar{r}(s, a) - \varphi(s, a)^T \lambda),$$
$$g_r(\theta, \lambda) := \mathbb{E}_{\xi \sim \mu_\theta}[\bar{g}_r(\xi, \lambda)].$$

768 Note here $g_r^i(\xi, \lambda)$ and $\bar{g}_r(\xi, \lambda)$ do not depend on the next state $s'$. We use $\xi$ for notational convience.

769 The following lemma is the counter part of Lemma 15 for reward estimator.

770 **Lemma 18** (descent of reward estimator's optimal gap (i.i.d. sampling))**.** *Suppopse Assumptions 1-4*
771 *hold, with $\lambda_{k+1}$ generated by Algorithm 1 given $\lambda_k$ and $\theta_k$ under i.i.d. sampling, then the following*
772 *holds*

$$\mathbb{E}\|\bar{\lambda}_{k+1} - \lambda^*(\theta_{k+1})\|^2 \leq (1 + 4L_{\lambda,2}^2 N\alpha_k + \frac{L_{\lambda,2}^2}{2}C_\theta^2 N^2 \alpha_k^2)\mathbb{E}\|\bar{\lambda}_{k+1} - \lambda^*(\theta_k)\|^2$$

$$+ (\frac{L_{\lambda,2}^2}{2}C_\theta^2 N^2 + L_\lambda^2 C_\theta^2 N^2)\alpha_k^2 + \frac{\alpha_k}{4}\sum_{i=1}^{N}\|\mathbb{E}[g_a^i(\xi_k, \lambda_{k+1}^i, \lambda_{k+1}^i)]\|^2.$$

(54)

773

$$\mathbb{E}\|\bar{\lambda}_{k+1} - \lambda^*(\theta_k)\|^2 \leq (1 - 2\eta_k \lambda_\varphi)\|\bar{\lambda}_k - \lambda^*(\theta_k)\|^2 + \eta_k^2 C_\lambda^2. \tag{55}$$

774 *Proof.* We begin with the optimal gap

$$\|\bar{\lambda}_{k+1} - \lambda^*(\theta_{k+1})\|^2$$
$$= \|\bar{\lambda}_{k+1} - \lambda^*(\theta_k) + \lambda^*(\theta_k) - \lambda^*(\theta_{k+1})\|^2$$
$$= \|\bar{\lambda}_{k+1} - \lambda^*(\theta_k)\|^2 + \|\lambda^*(\theta_k) - \lambda^*(\theta_{k+1})\|^2 + 2\langle \bar{\lambda}_{k+1} - \lambda^*(\theta_k), \lambda^*(\theta_k) - \lambda^*(\theta_{k+1}) \rangle$$
$$\leq \|\bar{\lambda}_{k+1} - \lambda^*(\theta_k)\|^2 + N^2 L_\lambda^2 C_\theta^2 \alpha_k^2 + 2\langle \bar{\lambda}_{k+1} - \lambda^*(\theta_k), \nabla\lambda^*(\theta_k)^T(\theta_k - \theta_{k+1}) \rangle$$
$$\quad + 2\langle \bar{\lambda}_{k+1} - \lambda^*(\theta_k), \lambda^*(\theta_k) - \lambda^*(\theta_{k+1}) - \nabla\lambda^*(\theta_k)^T(\theta_k - \theta_{k+1}) \rangle$$
$$\leq \|\bar{\lambda}_{k+1} - \lambda^*(\theta_k)\|^2 + N^2 L_\lambda^2 C_\theta^2 \alpha_k^2 + 2\alpha_k L_{\lambda,2}\sum_{i=1}^{N}\mathbb{E}\|\bar{\lambda}_{k+1} - \lambda^*(\theta_k)\|\|\mathbb{E}[g_a^i(\xi_k, \omega_{k+1}^i, \lambda_{k+1}^i)]\|$$
$$\quad + 2\langle \bar{\lambda}_{k+1} - \lambda^*(\theta_k), \lambda^*(\theta_k) - \lambda^*(\theta_{k+1}) - \nabla\lambda^*(\theta_k)^T(\theta_k - \theta_{k+1}) \rangle$$
$$\leq \|\bar{\lambda}_{k+1} - \lambda^*(\theta_k)\|^2 + N^2 L_\lambda^2 C_\theta^2 \alpha_k^2 + 4\alpha_k N L_{\lambda,2}^2 \mathbb{E}\|\bar{\lambda}_{k+1} - \lambda^*(\theta_k)\|^2 + \frac{\alpha_k}{4}\sum_{i=1}^{N}\|\mathbb{E}[g_a^i(\xi_k, \omega_{k+1}^i, \lambda_{k+1}^i)]\|^2$$
$$\quad + 2\langle \bar{\lambda}_{k+1} - \lambda^*(\theta_k), \lambda^*(\theta_k) - \lambda^*(\theta_{k+1}) - \nabla\lambda^*(\theta_k)^T(\theta_k - \theta_{k+1}) \rangle. \tag{56}$$

775 where the first inequality uses the Lipschitz continuous of $\lambda^*(\theta)$ and $\|\theta_k - \theta_{k+1}\|^2 \leq N^2 \alpha_k^2 C_\theta^2$. The
776 second inequality uses triangle inequality and the Lemma 2. The last inequality is due to Young's
777 inequality.

778 The last term in (56) can be bounded as

$$\mathbb{E}\langle \bar{\lambda}_{k+1} - \lambda^*(\theta_k), \lambda^*(\theta_k) - \lambda^*(\theta_{k+1}) - \nabla\lambda^*(\theta_k)^T(\theta_k - \theta_{k+1}) \rangle$$
$$\leq \frac{L_{\lambda,2}^2}{2}\mathbb{E}\|\bar{\lambda}_{k+1} - \lambda^*(\theta_k)\|\|\theta_{k+1} - \theta_k\|^2$$
$$\leq \frac{L_{\lambda,2}^2}{4}\mathbb{E}\|\bar{\lambda}_{k+1} - \lambda^*(\theta_k)\|^2\|\theta_{k+1} - \theta_k\|^2 + \frac{L_{\lambda,2}^2}{4}\|\theta_{k+1} - \theta_k\|^2$$
$$\leq \frac{L_{\lambda,2}^2}{4}N^2 C_\theta^2 \alpha_k^2 \mathbb{E}\|\bar{\lambda}_{k+1} - \lambda^*(\theta_k)\|^2 + \frac{L_{\lambda,2}^2}{4}N^2 C_\theta^2 \alpha_k^2. \tag{57}$$

The first inequality uses Lemma 10, and the second inequality is induced by Young's inequality. Plug (57) into (56) will yield (54).

We now prove (55)

$$
\begin{aligned}
\|\bar{\lambda}_{k+1} - \lambda^*(\theta_k)\|^2 &= \|\prod_{R_\lambda}(\bar{\lambda}_k - \eta_k \bar{g}_r(\xi_k, \bar{\lambda}_k)) - \prod_{R_\lambda} \lambda^*(\theta_k)\|^2 \\
&\leq \|\bar{\lambda}_k - \eta_k \bar{g}_r(\xi_k, \bar{\lambda}_k) - \lambda^*(\theta_k)\|^2 \\
&\leq \|\bar{\lambda}_k - \lambda^*(\theta_k)\|^2 + \eta_k^2 \|\bar{g}_r(\xi_k, \bar{\lambda}_k)\|^2 + 2\eta_k \mathbb{E}[\langle \bar{\lambda}_k - \lambda^*(\theta_k), \bar{g}_r(\xi_k, \bar{\lambda}_k)\rangle] \\
&\leq \|\bar{\lambda}_k - \lambda^*(\theta_k)\|^2 + C_\lambda \eta_k^2 - 2\eta_k \mathbb{E}[\langle \bar{\lambda}_k + \lambda^*(\theta_k), \bar{g}_r(s_k, a_k, \bar{\lambda}_k)\rangle], \quad (58)
\end{aligned}
$$

where the last inequality is due to $\|\bar{g}_r(\xi_k, \bar{\lambda}_k)\| \leq |r(s,a) - \varphi(s,a)^T \lambda| \leq r_{\max} + R_\lambda := C_\lambda$.

The last term can be bounded as

$$
\begin{aligned}
\mathbb{E}[\langle \bar{\lambda}_k - \lambda^*(\theta_k), \bar{g}_r(\xi_k, \bar{\lambda}_k)\rangle] &= \langle \bar{\lambda}_k - \lambda^*(\theta_k), \mathbb{E}[\bar{g}_r(\xi_k, \bar{\lambda}_k) - g_r(\theta_k, \lambda^*(\theta_k))]\rangle \\
&= \langle \bar{\lambda}_k - \lambda^*(\theta_k), \mathbb{E}_{\xi \sim \mu_{\theta_k}}[\varphi(s_k, a_k)\varphi(s_k, a_k)^T | \bar{\lambda}_k](\lambda^*(\theta_k) - \bar{\lambda}_k)\rangle \\
&= \langle \bar{\lambda}_k - \lambda^*(\theta_k), A_{\theta,\varphi}(\lambda^*(\theta_k) - \bar{\lambda}_k)\rangle \\
&\leq -\lambda_\varphi \|\bar{\lambda}_k - \lambda^*(\theta_k)\|^2, \quad (59)
\end{aligned}
$$

where the first equality is according to the optimality condition of reward estimator

$$
\mathbb{E}_{\xi \sim \mu_{\theta_k}}[\varphi(s,a)(r(s,a) - \varphi(s,a)^T \lambda^*(\theta_k))] = 0.
$$

Plug (59) into (58) will give us (55), which completes the proof. $\qquad \square$

**Lemma 19** (descent of reward estimator's optimal gap (Markovian sampling)). *Suppose Assumptions 1-4 hold, with $\lambda_{k+1}$ generated by Algorithm 1 given $\lambda_k$ and $\theta_k$ under Markovian sampling, then the following holds*

$$
\begin{aligned}
\mathbb{E}\|\bar{\lambda}_{k+1} - \lambda^*(\theta_{k+1})\|^2 &\leq (1 + 4L_{\lambda,2}^2 N\alpha_k + \frac{L_{\lambda,2}^2}{2} C_\theta^2 N^2 \alpha_k^2)\mathbb{E}\|\bar{\lambda}_{k+1} - \lambda^*(\theta_k)\|^2 \\
&+ (\frac{L_{\lambda,2}^2}{2} C_\theta^2 N^2 + L_\lambda^2 C_\theta^2 N^2)\alpha_k^2 + \frac{\alpha_k}{4} \sum_{i=1}^N \|\mathbb{E}[g_a^i(\xi_k, \lambda_{k+1}^i, \lambda_{k+1}^i)]\|^2.
\end{aligned}
$$
(60)

$$
\mathbb{E}\|\bar{\lambda}_{k+1} - \lambda^*(\theta_k)\|^2 \leq (1 - 2\eta_k \lambda_\varphi)\|\bar{\lambda}_k - \lambda^*(\theta_k)\|^2 + C_{K_3}\eta_k \eta_{k-Z_K} + C_{K_4}\eta_k \alpha_{k-Z_K}, \quad (61)
$$

*where* $C_{K_3} := 4C_6 C_\lambda Z_K + C_\lambda^2$, $C_{K_4} := 4C_5 C_\theta Z_K + 2C_7 C_\theta Z_K^2 + C_8$, $Z_K := \min\{z \in \mathbb{N}^+ | \kappa \rho^{z-1} \leq \min\{\alpha_k, \eta_k, \eta_k\}\}$.

*Proof.* Since analysis of (60) does not involve the update of $\bar{\lambda}_k$, it can be directly recovered from (54).

We now prove (61). Following the derivation of (58), we obtain

$$
\begin{aligned}
\|\bar{\lambda}_{k+1} - \lambda^*(\theta_k)\|^2 &\leq \|\bar{\lambda}_k - \lambda^*(\theta_k)\|^2 + C_\lambda^2 \eta_k^2 + 2\eta_k \mathbb{E}[\langle \bar{\lambda}_k - \lambda^*(\theta_k), \bar{g}_r(\xi_k, \bar{\lambda}_k)\rangle] \\
&= \|\bar{\lambda}_k - \lambda^*(\theta_k)\|^2 + C_\lambda^2 \eta_k^2 + 2\eta_k \mathbb{E}[\langle \bar{\lambda}_k - \lambda^*(\theta_k), g_r(\theta_k, \bar{\lambda}_k)\rangle] \\
&+ 2\eta_k \mathbb{E}[\langle \bar{\lambda}_k - \lambda^*(\theta_k), \bar{g}_r(\xi_k, \bar{\lambda}_k) - g_r(\theta_k, \bar{\lambda}_k)\rangle] \\
&\leq (1 - 2\lambda_\varphi \eta_k)\|\bar{\lambda}_k - \lambda^*(\theta_k)\|^2 + C_\lambda^2 \eta_k^2 \\
&+ 2\eta_k \mathbb{E}[\langle \bar{\lambda}_k - \lambda^*(\theta_k), \bar{g}_r(\xi_k, \bar{\lambda}_k) - g_r(\theta_k, \bar{\lambda}_k)\rangle], \quad (62)
\end{aligned}
$$

where the last inequality is obtained by (61).

We now bound the last term. By Lemma 20, for any $z \in \mathbb{N}^+$, we have

$$\mathbb{E}\langle \bar{\lambda}_k - \lambda^*(\theta_k), \bar{g}_r(\xi_k, \bar{\lambda}_k) - g_r(\theta_k, \bar{\lambda}_k)\rangle$$

$$\leq C_5 \mathbb{E}\|\theta_k - \theta_{k-z}\| + C_6 \mathbb{E}\|\bar{\lambda}_k - \bar{\lambda}_{k-z}\| + C_7 \sum_{m=0}^{z-1} \mathbb{E}\|\theta_{k-m} - \theta_{k-z}\| + C_8 \kappa \rho^{z-1}$$

$$\stackrel{(i)}{\leq} C_5 \sum_{n=1}^{z} \mathbb{E}\|\theta_{k-n+1} - \theta_{k-n}\| + C_6 \sum_{n=1}^{z} \mathbb{E}\|\bar{\lambda}_{k-n+1} - \bar{\lambda}_{k-n}\|$$

$$+ C_7 \sum_{m=0}^{z-1} \sum_{n=1}^{z-m} \mathbb{E}\|\theta_{k-m-n+1} - \theta_{k-m-n}\| + C_8 \kappa \rho^{z-1}$$

$$\leq 2C_5 C_\theta \sum_{n=1}^{z} \alpha_{k-n} + 2C_6 C_\lambda \sum_{n=1}^{z} \eta_{k-n} + C_7 C_\theta \sum_{m=0}^{z-1} \sum_{n=1}^{z-m} \alpha_{k-m-n} + C_8 \kappa \rho^{z-1}$$

$$\stackrel{(ii)}{\leq} 2C_5 C_\theta z \alpha_{k-z} + 2C_6 C_\lambda z \eta_{k-z} + C_7 C_\theta z(z-1)\alpha_{k-z} + C_8 \kappa \rho^{z-1}, \tag{63}$$

797 where the $(i)$ uses triangle inequality, $(ii)$ uses the non-increasing property of step sizes.

798 Let $z = Z_K$, recall $Z_K := \min\{z \in \mathbb{N}^+ | \kappa \rho^{z-1} \leq \min\{\alpha_k, \eta_k, \eta_k\}\}$, we have
$$\mathbb{E}\langle \bar{\lambda}_k - \lambda^*(\theta_k), \bar{g}_r(\xi_k, \bar{\lambda}_k) - g_r(\theta_k, \bar{\lambda}_k)\rangle$$
$$\leq 2C_5 C_\theta Z_K \alpha_{k-Z_K} + 2C_6 C_\lambda Z_K \eta_{k-Z_K} + C_7 C_\theta Z_K^2 \alpha_{k-Z_K} + C_8 \alpha_{k-Z_K}. \tag{64}$$

799 Plug (64) into (62) will yield
$$\|\bar{\lambda}_{k+1} - \lambda^*(\theta_k)\|^2 \leq (1 - 2\lambda_\phi \eta_k)\|\bar{\lambda}_k - \lambda^*(\theta_k)\|^2 + C_\lambda^2 \eta_k^2$$
$$+ 4C_5 C_\theta Z_K \alpha_{k-Z_K} + 4C_6 C_\lambda Z_K \eta_{k-Z_K} + 2C_7 C_\theta Z_K^2 \alpha_{k-Z_K} + 2C_8 \alpha_{k-Z_K}.$$

800 By defining $C_{K_3} := 4C_6 C_\lambda Z_K + C_\lambda^2$, $C_{K_4} := 4C_5 C_\theta Z_K + 2C_7 C_\theta Z_K^2 + C_8$, we complete the
801 proof.

802 $\hfill\square$

803 **Lemma 20.** *Consider the sequence generated by Algorithm 1, for any $z \in \mathbb{N}^+$, we have*
$$\mathbb{E}[\langle \bar{\lambda}_k - \lambda^*(\theta), \bar{g}_r(\xi_k, \bar{\lambda}_k) - g_r(\theta_k, \bar{\lambda}_k)\rangle] \leq C_5\|\theta_k - \theta_{k-z}\| + C_6\|\lambda_k - \lambda_{k-z}\|$$
$$+ C_7 \sum_{m=0}^{z-1} \|\theta_{k-m} - \theta_{k-z}\| + C_8 \kappa \rho^{z-1}, \tag{65}$$

804 *where $C_5 := 4R_\lambda C_\lambda |\mathcal{A}| L_\pi (1 + \log_\rho \kappa^{-1} + (1-\rho)^{-1}) + 2C_\lambda L_\lambda$, $C_6 := 4R_\lambda + 2C_\lambda$, $C_7 :=$*
805 *$4R_\lambda C_\lambda |\mathcal{A}| L_\pi$, $C_8 := 8R_\lambda C_\lambda$.*

806 *Proof.* Consider the Markov chain since timestep $k - z$:
$$s_{k-m} \xrightarrow{\theta_{k-m}} a_{k-m} \xrightarrow{\mathcal{P}} s_{k-m+1} \xrightarrow{\theta_{k-m+1}} a_{k-m+1} \cdots \xrightarrow{\theta_{k-1}} a_{k-1} \xrightarrow{\mathcal{P}} s_k \xrightarrow{\theta_k} a_k \xrightarrow{\mathcal{P}} s_{k+1}.$$
807 Also consider the auxiliary Markov chain with fixed policy since timestep $k - z$:
$$s_{k-m} \xrightarrow{\theta_{k-m}} a_{k-m} \xrightarrow{\mathcal{P}} s_{k-m+1} \xrightarrow{\theta_{k-m}} \tilde{a}_{k-m+1} \cdots \xrightarrow{\theta_{k-m}} \tilde{a}_{k-1} \xrightarrow{\mathcal{P}} \tilde{s}_k \xrightarrow{\theta_{k-m}} \tilde{a}_k \xrightarrow{\mathcal{P}} \tilde{s}_{k+1}.$$
808 Throughout the proof, we will use $\theta, \theta', \bar{\lambda}, \bar{\lambda}', \xi, \tilde{\xi}$ to represent $\theta_k, \theta_{k-z}, \bar{\lambda}_k, \bar{\lambda}_{k-z}, \xi_k, \xi_{k-z}$, respec-
809 tively.

810 For the ease of expression, define
$$\Delta_2(\xi, \lambda, \theta) := \langle \lambda - \lambda^*(\theta), \bar{g}_r(\xi, \lambda) - g_r(\theta, \lambda)\rangle.$$

811 We have
$$\langle \bar{\lambda}_k - \lambda^*(\theta), \bar{g}_r(\xi_k, \bar{\lambda}_k) - g_r(\theta_k, \bar{\lambda}_k)\rangle = \Delta_2(\xi, \bar{\lambda}, \theta)$$
$$= \underbrace{\Delta_2(\xi, \bar{\lambda}, \theta) - \Delta_2(\xi, \bar{\lambda}, \theta')}_{I_1} + \underbrace{\Delta_2(\xi, \bar{\lambda}, \theta') - \Delta_2(\xi, \bar{\lambda}', \theta')}_{I_2}$$
$$+ \underbrace{\Delta_2(\xi, \bar{\lambda}', \theta') - \Delta_2(\tilde{\xi}, \bar{\lambda}', \theta')}_{I_3} + \underbrace{\Delta_2(\tilde{\xi}, \bar{\lambda}', \theta')}_{I_4}.$$

$I_1$ can be expressed as

$$\begin{aligned}
I_1 &= \langle \bar{\lambda} - \lambda^*(\theta), \bar{g}_r(\xi, \bar{\lambda}) - g_r(\theta, \bar{\lambda}) \rangle - \langle \bar{\lambda} - \lambda^*(\theta'), \bar{g}_r(\xi, \bar{\lambda}) - g_r(\theta', \bar{\lambda}) \rangle \\
&= \langle \bar{\lambda} - \lambda^*(\theta), \bar{g}_r(\xi, \bar{\lambda}) - g_r(\theta, \bar{\lambda}) \rangle - \langle \bar{\lambda} - \lambda^*(\theta), \bar{g}_r(\xi, \bar{\lambda}) - g_r(\theta', \bar{\lambda}) \rangle \\
&\quad + \langle \lambda^*(\theta) - \lambda^*(\theta'), \bar{g}_r(\xi, \bar{\lambda}) - g_r(\theta', \bar{\lambda}) \rangle \\
&\leq \|\bar{\lambda} - \lambda^*(\theta)\| \|g_r(\theta', \bar{\lambda}) - g_r(\theta, \bar{\lambda})\| + \|\lambda^*(\theta) - \lambda^*(\theta')\| \|\bar{g}_r(\xi, \bar{\lambda}) - g_r(\theta', \bar{\lambda})\|. \quad (66)
\end{aligned}$$

The first term can be bounded as

$$\begin{aligned}
\|\bar{\lambda} - \lambda^*(\theta)\| \|g_r(\theta', \bar{\lambda}) - g_r(\theta, \bar{\lambda})\| &\leq 2R_\lambda \|\mathbb{E}_{\xi \sim \mu'_\theta}[\bar{g}_r(\xi, \bar{\lambda})] - \mathbb{E}_{\xi \sim \mu_\theta}[\bar{g}_r(\xi, \bar{\lambda})]\| \\
&\leq 4R_\lambda \sup_\xi \|\bar{g}_r(\xi, \bar{\lambda})\| d_{TV}(\mu'_\theta \otimes \pi'_\theta \otimes \mathcal{P}, \mu_\theta \otimes \pi_\theta \otimes \mathcal{P}) \\
&\leq 4R_\lambda C_\lambda d_{TV}(\mu'_\theta \otimes \pi'_\theta \otimes \mathcal{P}, \mu_\theta \otimes \pi_\theta \otimes \mathcal{P}) \\
&\leq 4R_\lambda C_\lambda |\mathcal{A}| L_\pi (1 + \log_\rho \kappa^{-1} + (1 - \rho)^{-1}) \|\theta - \theta'\|, \quad (67)
\end{aligned}$$

where the first inequality follows the projection update of each lambda step, the third inequality is due to $\|\bar{g}_r(\xi, \bar{\lambda})\| \leq C_\lambda$, and the last inequality follows Lemma 8.

The second term can be bounded as

$$\|\lambda^*(\theta) - \lambda^*(\theta')\| \|\bar{g}_r(\xi, \bar{\lambda}) - g_r(\theta, \bar{\lambda})\| \leq 2C_\lambda L_\lambda \|\theta - \theta'\| \quad (68)$$

Plug (67) and (68) into (66), we can bound $I_1$ as

$$I_1 \leq (4R_\lambda C_\lambda |\mathcal{A}| L_\pi (1 + \log_\rho \kappa^{-1} + (1 - \rho)^{-1}) + 2C_\lambda L_\lambda) \|\theta - \theta'\|. \quad (69)$$

Next we bound $I_2$ as

$$\begin{aligned}
I_2 &= \langle \bar{\lambda} - \lambda^*(\theta'), \bar{g}_r(\xi, \bar{\lambda}) - g_r(\theta', \bar{\lambda}) \rangle - \langle \bar{\lambda}' - \lambda^*(\theta'), \bar{g}_r(\xi, \bar{\lambda}') - g_r(\theta', \bar{\lambda}') \rangle \\
&= \langle \bar{\lambda} - \lambda^*(\theta'), \bar{g}_r(\xi, \bar{\lambda}) - g_r(\theta', \bar{\lambda}) \rangle - \langle \bar{\lambda}' - \lambda^*(\theta'), \bar{g}_r(\xi, \bar{\lambda}) - g_r(\theta', \bar{\lambda}) \rangle \\
&\quad + \langle \bar{\lambda}' - \lambda^*(\theta'), \bar{g}_r(\xi, \bar{\lambda}) - \bar{g}_r(\xi, \bar{\lambda}') - g_r(\theta', \bar{\lambda}) + g_r(\theta', \bar{\lambda}') \rangle.
\end{aligned}$$

The first two terms can be bounded as

$$\langle \bar{\lambda} - \bar{\lambda}', \bar{g}_r(\xi, \bar{\lambda}) - g_r(\theta', \bar{\lambda}) \rangle \leq 2C_\lambda \|\bar{\lambda} - \bar{\lambda}'\|. \quad (70)$$

The last term can be bounded as

$$\begin{aligned}
&\langle \bar{\lambda}' - \lambda^*(\theta'), \bar{g}_r(\xi, \bar{\lambda}) - \bar{g}_r(\xi, \bar{\lambda}') - g_r(\theta', \bar{\lambda}) + g_r(\theta', \bar{\lambda}') \rangle \\
&\leq \|\bar{\lambda} - \lambda^*(\theta')\| (\|\bar{g}_r(\xi, \bar{\lambda}) - \bar{g}_r(\xi, \bar{\lambda}')\| + \|g_r(\theta', \bar{\lambda}') - g_r(\theta', \bar{\lambda})\|) \\
&\leq 2R_\lambda (\|\bar{g}_r(\xi, \bar{\lambda}) - \bar{g}_r(\xi, \bar{\lambda}')\| + \|g_r(\theta', \bar{\lambda}') - g_r(\theta', \bar{\lambda})\|) \\
&\leq 4R_\lambda \|\bar{\lambda} - \bar{\lambda}'\|, \quad (71)
\end{aligned}$$

where the second inequality follows the projection of each lambda step. The last inequality is due to

$$\begin{aligned}
\|\bar{g}_r(\xi, \bar{\lambda}) - \bar{g}_r(\xi, \bar{\lambda}')\| &= \|\varphi(s, a)(\varphi(s, a)^T(\bar{\lambda} - \bar{\lambda}'))\| \\
&\leq \|\bar{\lambda} - \bar{\lambda}'\|
\end{aligned}$$

Combine (70) and (71), we can bound $I_2$ as

$$I_2 \leq (4R_\lambda + 2C_\lambda) \|\bar{\lambda} - \bar{\lambda}'\|. \quad (72)$$

We bound $I_3$ as

$$\begin{aligned}
\mathbb{E}[I_3 | \theta', s_{k-z+1}] &= \mathbb{E}[\Delta_2(\xi, \theta', \bar{\lambda}') - \Delta_2(\tilde{\xi}, \theta', \bar{\lambda}') | \theta', s_{k-z+1}] \\
&\leq 2 \sup_\xi |\Delta_2(\xi, \theta', \bar{\lambda}')| \, d_{TV}(\mathbb{P}(\xi \in \cdot | \theta', s_{k-z+1}), \mathbb{P}(\tilde{\xi} \in \cdot | \theta', s_{k-z+1})) \\
&\leq 8R_\lambda C_\lambda d_{TV}(\mathbb{P}(\xi \in \cdot | \theta', s_{k-z+1}), \mathbb{P}(\tilde{\xi} \in \cdot | \theta', s_{k-z+1})) \\
&\leq 4R_\lambda C_\lambda |\mathcal{A}| L_\pi \sum_{m=0}^{z-1} \|\theta_{k-m} - \theta_{k-z}\|. \quad (73)
\end{aligned}$$

824 Here, the second inequality is due to $\|\Delta_2(\xi, \theta', \bar{\lambda}')\| \leq \|\lambda' - \lambda^*(\theta')\|\|\bar{g}_r(\xi, \lambda') - g_r(\theta', \lambda')\| \leq$
825 $4R_\lambda C_\lambda$, and the last inequality is according to Lemma 13.

826 We now bound $I_4$

$$
\begin{aligned}
\mathbb{E}[I_4|\theta', \bar{\lambda}', s_{k+z-1}] &= \mathbb{E}[\Delta_2(\tilde{\xi}, \theta', \bar{\lambda}')|\theta', \bar{\lambda}', s_{k-z+1}] \\
&\leq \sup_\xi |\Delta_2(\xi, \theta', \bar{\lambda}')\|\mathbb{P}(\xi \in \cdot|\theta', s_{k-z+1}) - \mu_{\theta'} \otimes \pi_{\theta'} \otimes \mathcal{P}\| \\
&\leq 8R_\lambda C_\lambda d_{TV}(\mathbb{P}(\tilde{x} \in \cdot|\theta', s_{t-z+1}), \mu_{\theta'} \otimes \pi_{\theta'} \otimes \mathcal{P}) \\
&\leq 8R_\lambda C_\lambda \kappa \rho^{z-1},
\end{aligned}
\tag{74}
$$

827 where the last inequality follows Lemma 5.

828 Plug (69), (72), (73), and (74) into (65), we get

$$
\begin{aligned}
\mathbb{E}[\Delta_2(\xi, \theta, \bar{\lambda})] &\leq (4R_\lambda C_\lambda|\mathcal{A}|L_\pi(1 + \log_\rho \kappa^{-1} + (1-\rho)^{-1}) + 2C_\lambda L_\lambda)\mathbb{E}\|\theta_k - \theta_{k-z}\| \\
&\quad + (4R_\lambda + 2C_\lambda)\mathbb{E}\|\bar{\lambda}_k - \bar{\lambda}_{k-z}\| \\
&\quad + 4R_\lambda C_\lambda|\mathcal{A}|L_\pi \sum_{m=0}^{z-1} \mathbb{E}\|\theta_{k-m} - \theta_{k-z}\| \\
&\quad + 8R_\lambda C_\lambda \kappa \rho^{z-1},
\end{aligned}
$$

829 which completes the proof. $\qquad\square$

## D.3 Consensus error

831 **Lemma 21** (bound of consensus error). *Suppose Asssumptions 1 and 5 hold. Let $\omega_k, \lambda_k$ be the*
832 *sequence generated by the algorithm 1, then for $k \geq 1$, the following hold*

$$
\sum_{i=1}^N \|\omega_k^i - \bar{\omega}_k\|^2 \leq \nu^{2k}\|\omega_0\|_F + \frac{16NC_\delta^2}{1-\nu}\beta_k^2 + \frac{8\sqrt{N}C_\delta\|\omega_0\|_F}{1-\nu}\nu^k\beta_k.
\tag{75}
$$

$$
\sum_{i=1}^N \|\lambda_k^i - \bar{\lambda}_k\|^2 \leq \nu^{2k}\|\lambda_0\|_F + \frac{16NC_\lambda^2}{1-\nu}\eta_k^2 + \frac{8\sqrt{N}C_\lambda\|\lambda_0\|_F}{1-\nu}\nu^k\eta_k,
\tag{76}
$$

833 *where $\nu \in [0, 1]$ is the second largest singular value of $W$. $\omega_k, \lambda_k$ are defined as*

$$
\omega_k := \begin{bmatrix} (\omega_k^1)^T \\ \vdots \\ (\omega_k^N)^T \end{bmatrix}, \qquad \lambda_k := \begin{bmatrix} (\lambda_k^1)^T \\ \vdots \\ (\lambda_k^N)^T \end{bmatrix}.
$$

834 *Proof.* We will prove the bound in (75) for critic variables. The analysis for reward estimator in (76)
835 follows the same routine. To simplify the notation, we will use $g_k^i$ to represent $g_c^i(\xi_k, \omega_k^i)$ throughout
836 the proof of this lemma. We also use $e_k^i$ to represent the projection error $e_k^i := \prod_{R_\omega}(\omega_k^i - \beta_k g_k^i) -$
837 $(\omega_k^i - \beta_k g_k^i)$. Also define $\bar{g}_k := \frac{1}{N}\sum_{i=1}^N g_k^i; \bar{e}_k := \frac{1}{N}\sum_{i=1}^N e_k^i$. The corresponding matrix exressions
838 are

$$
G_k := \begin{bmatrix} (g_k^1)^T, \\ \vdots \\ (g_k^N)^T \end{bmatrix}, E_k := \begin{bmatrix} (e_k^1)^T, \\ \vdots \\ (e_k^N)^T \end{bmatrix}.
$$

839 Then the following equality holds by the update rule of critic variables

$$
\omega_{k+1} = \begin{cases} W\omega_k - \beta_k G_k + E_k, & \text{if } k \bmod K_c = 0 \\ \omega_k - \beta_k G_k + E_k, & \text{otherwise.} \end{cases}
\tag{77}
$$

840 Let $Q := I - \frac{1}{N}\mathbf{1}\mathbf{1}^T$, then the consensus error can be expressed as $\|\omega_k - \mathbf{1}\bar{\omega}_k^T\|_F = \|Q\omega_k\|_F$.

We bound the consensus error of critic's first

$$\|QG_k\| = \sqrt{\sum_{i=1}^{N} \|g_k^i - \bar{g}_k\|} \overset{(i)}{\leq} \sqrt{\sum_{i=1}^{N} 2\|g_k^i\|^2 + 2\|\bar{g}_k\|^2} \leq 2\sqrt{N}C_\delta. \tag{78}$$

$$\|QE_k\| = \sqrt{\sum_{i=1}^{N} \|e_t^i - \bar{e}_t\|} \leq \sqrt{\sum_{i=1}^{N} 2\|e_k^i\|^2 + 2\|\bar{e}_k\|^2} \overset{(ii)}{\leq} \sqrt{\sum_{i=1}^{N} 2\|g_k^i\|^2 + 2\|\bar{g}_k\|^2} \leq 2\beta_k\sqrt{N}C_\delta, \tag{79}$$

where $(i)$ is due to $\|g_k^i\| \leq C_\delta$, $(ii)$ is ensured by the convexity of the projection set.

We now study the consensus error of critic variables. Let $k' = \lfloor \frac{k}{K_c} \rfloor * K_c$. Without loss of generality, assume $k \bmod K_c \neq 0$. We have

$$\begin{aligned}
Q\boldsymbol{\omega}_{k+1} &= QW\boldsymbol{\omega}_k - \beta_k QG_k + QE_k \\
&= WQ\boldsymbol{\omega}_k + \beta_k QG_k + QE_k \\
&= W^{k+1}Q\boldsymbol{\omega}_0 + \sum_{t=0}^{k} \beta_t W^{k-t}QG_t + \sum_{t=0}^{k} W^{k-t}QE_k,
\end{aligned} \tag{80}$$

where the first equality follows (77). The second equality is due to the doubly stochasticity of matrix $W$ (see Assumption 5): $QW = W - \frac{1}{N}\mathbf{1}\mathbf{1}^T W = W - \frac{1}{N}W\mathbf{1}\mathbf{1}^T = WQ$. The last equality expands the recursion of the second equation.

Take Frobenius norm on each side of (80) and apply triangle inequality, we get

$$\begin{aligned}
\|Q\boldsymbol{\omega}_{k+1}\|_F &\leq \|W^k\boldsymbol{\omega}_0\|_F + \sum_{t=0}^{k} \beta_t\|W^{k-t}QG_t\|_F + \sum_{t=0}^{k} \|W^{k-t}QE_k\|_F \\
&\leq \nu^k\|\boldsymbol{\omega}_0\|_F + 4\sum_{t=0}^{k} \beta_t \nu^{k-t}\sqrt{N}C_\delta \\
&\leq \nu^k\|\boldsymbol{\omega}_0\|_F + \frac{4\sqrt{N}C_\delta\beta_k}{1-\nu}.
\end{aligned} \tag{81}$$

The $\nu$ in (81) denotes the second largest singular value of $W$, which satisfies $\nu < 1$ as specified by Assumption 5. The second inequality uses (78), (79) and Lemma 9.

Take square on each side, we obtain

$$\|Q\boldsymbol{\omega}_{k+1}\|_F^2 \leq \nu^{2k}\|\boldsymbol{\omega}_0\|_F + \frac{16NC_\delta^2}{1-\nu}\beta_k^2 + \frac{8\sqrt{N}C_\delta\|\boldsymbol{\omega}_0\|_F}{1-\nu}\nu^k\beta_k$$

which completes the proof for (75). The proof of (76) follows similar procedure, we leave it as an exercise to reader.

$\square$

## D.4 Error of actor

**Lemma 22.** *Consider the sequence generated by Algorithm 1, for any $z \geq 1$ we have*

$$\|\mathbb{E}_{\xi\sim\mu_{\theta_k}}[\delta(\xi, \theta_k)\psi_{\theta_k^i}(s_k, a_k^i)] - \mathbb{E}[\delta(\xi_k, \theta_k)\psi_{\theta_k^i}(s_k, a_k^i)]\|$$

$$\leq 2C_\theta\kappa\rho^{z-1} + C_{12}\sum_{m=0}^{z-1} \|\theta_{k-m} - \theta_{k-z}\| + C_{13}\|\theta_k - \theta_{k-z}\| + C_{14}\|\theta_k^i - \theta_{k-z}^i\|, \tag{82}$$

*where $C_{12} := 2C_\theta|\mathcal{A}|L_\pi$, $C_{13} := |\mathcal{A}|L(\log_\rho \kappa^{-1} + (1-\rho)^{-1})C_\theta + 2(1+\gamma)L_V$, $C_{14} := 2C_\delta L_\psi$.*

*Proof.* Consider the Markov chain since timestep $k - z$:

$$s_{k-z} \xrightarrow{\theta_{k-z}} a_{k-z} \xrightarrow{\mathcal{P}} s_{k-z+1} \xrightarrow{\theta_{k-z+1}} a_{k-z+1} \cdots \xrightarrow{\theta_{k-1}} a_{k-1} \xrightarrow{\mathcal{P}} s_k \xrightarrow{\theta_k} a_k \xrightarrow{\mathcal{P}} s_{k+1}.$$

Also consider the auxiliary Markov chain with fixed policy since timestep $k - z$:

$$s_{k-z} \xrightarrow{\theta_{k-z}} a_{k-z} \xrightarrow{\mathcal{P}} s_{k-z+1} \xrightarrow{\theta_{k-z}} \tilde{a}_{k-z+1} \cdots \xrightarrow{\theta_{k-z}} \tilde{a}_{k-1} \xrightarrow{\mathcal{P}} \tilde{s}_k \xrightarrow{\theta_{k-z}} \tilde{a}_k \xrightarrow{\mathcal{P}} \tilde{s}_{k+1}.$$

Throughout the proof of this lemma, we wil use $\psi_{\theta^i}$ to represent $\psi_{\theta^i}(s_k, a_k^i)$ for brevity.

We define the following notation for the ease of discussion

$$\Delta_3(\xi, \theta) := \mathbb{E}_{\xi \sim \mu_\theta}[\delta(\xi, \theta)\psi_{\theta^i}] - \delta(\xi, \theta)\psi_{\theta^i}.$$

Then our objective is to bound

$$\mathbb{E}[\|\Delta_3(\xi_k, \theta_k)\| \,\|\, \theta_{k-z}].$$

We decompose $\|\Delta_3(\xi_k, \theta_k)\|$ by applying triangle inequality

$$\|\Delta_3(\xi_k, \theta_k)\| \leq \underbrace{\|\Delta_3(\xi_k, \theta_k) - \Delta_3(\xi_k, \theta_{k-z})\|}_{I_1}$$

$$+ \underbrace{\|\Delta_3(\xi_k, \theta_{k-z}) - \Delta_3(\tilde{\xi}_k, \theta_{k-z})\|}_{I_2}$$

$$+ \underbrace{\|\Delta_3(\tilde{\xi}_k, \theta_{k-z})\|}_{I_3}. \tag{83}$$

We apply triangle inequality again to bound $I_1$ as

$$I_1 \leq \underbrace{\|\delta(\xi_k, \theta_{k-z})\psi_{\theta_{k-z}^i} - \delta(\xi_k, \theta_k)\psi_{\theta_k^i}\|}_{I_1^{(1)}}$$

$$+ \underbrace{\|\mathbb{E}_{\xi \sim \mu_{\theta_k}}[\delta(\xi, \theta_k)\psi_{\theta_k^i}] - \mathbb{E}_{\xi \sim \mu_{\theta_{k-z}}}[\delta(\xi, \theta_{k-z})\psi_{\theta_{k-z}^i}]\|}_{I_1^{(2)}} \tag{84}$$

$I_1^{(1)}$ can be bounded as

$$\begin{aligned}
I_1^{(1)} &= \|\delta(\xi_k, \theta_{k-z})\psi_{\theta_{k-z}^i} - \delta(\xi_k, \theta_k)\psi_{\theta_k^i}\| \\
&\leq \|\delta(\xi_k, \theta_{k-z})\psi_{\theta_{k-z}^i} - \delta(\xi_k, \theta_k)\psi_{\theta_{k-z}^i}\| \\
&\quad + \|\delta(\xi_k, \theta_k)\psi_{\theta_{k-z}^i} - \delta(\xi_k, \theta_k)\psi_{\theta_k^i}\| \\
&\leq \||\gamma(V_{\theta_{k-z}}(s') - V_{\theta_k}(s')) + (V_{\theta_{k-z}}(s) - V_{\theta_{k-z}}(s'))|\psi_{k-z}^i\| \\
&\quad + \|\delta(\xi_k, \theta_k)\psi_{\theta_{k-z}^i} - \delta(\xi_k, \theta_k)\psi_{\theta_k^i}\| \\
&\leq (1+\gamma)L_V\|\theta_k - \theta_{k-z}\| + \|\delta(\xi_k, \theta_k)\psi_{\theta_{k-z}^i} - \delta(\xi_k, \theta_k)\psi_{\theta_k^i}\| \\
&\leq (1+\gamma)L_V\|\theta_k - \theta_{k-z}\| + C_\delta L_\psi\|\theta_k^i - \theta_{k-z}^i\|, \tag{85}
\end{aligned}$$

where the second last inequality follows the Lipschitz continuous of value function in Lemma 7, and the last inequality uses Lipschitz continuous of $\psi_{\theta^i}$.

$I_1^{(2)}$ can be bounded as

$$\begin{aligned}
I_1^{(2)} &= \|\mathbb{E}_{\xi \sim \mu_{\theta_k}}[\delta(\xi, \theta_k)\psi_{\theta_k^i}] - \mathbb{E}_{\xi \sim \mu_{\theta_{k-z}}}[\delta(\xi, \theta_{k-z})\psi_{\theta_{k-z}^i}]\| \\
&= \|\mathbb{E}_{\xi \sim \mu_{\theta_k}}[\delta(\xi, \theta_{k-z})\psi_{\theta_{k-z}^i}] - \mathbb{E}_{\xi \sim \mu_{\theta_{k-z}}}[\delta(\xi, \theta_{k-z})\psi_{\theta_{k-z}^i}] \\
&\quad + \mathbb{E}_{\xi \sim \mu_{\theta_k}}[\delta(\xi, \theta_k)\psi_{\theta_k^i} - \delta(\xi, \theta_{k-z})\psi_{\theta_{k-z}^i}]\| \\
&\leq |\mathcal{A}|L(\log_\rho \kappa^{-1} + (1-\rho)^{-1})C_\theta\|\theta_k - \theta_{k-z}\| \\
&\quad + \|\mathbb{E}_{\xi \sim \mu_{\theta_k}}[\delta(\xi, \theta_k)\psi_{\theta_k^i} - \delta(\xi, \theta_{k-z})\psi_{\theta_{k-z}^i}]\| \\
&\leq |\mathcal{A}|L(\log_\rho \kappa^{-1} + (1-\rho)^{-1})C_\theta\|\theta_k - \theta_{k-z}\| \\
&\quad + (1+\gamma)L_V\|\theta_k - \theta_{k-z}\| + C_\delta L_\psi\|\theta_k^i - \theta_{k-z}^i\|, \tag{86}
\end{aligned}$$

869 where the first inequality applies Lemma 8, and the last inequality uses the derivation in (85).

870 Combine (85) and (86), we have

$$
\begin{aligned}
I_1 \leq\ & |\mathcal{A}| L(\log_\rho \kappa^{-1} + (1-\rho)^{-1}) C_\theta \|\theta_k - \theta_{k-z}\| \\
& + 2(1+\gamma) L_V \|\theta_k - \theta_{k-z}\| + 2 C_\delta L_\psi \|\theta_k^i - \theta_{k-z}^i\|
\end{aligned}
\tag{87}
$$

871 We now bound $I_2$ as

$$
\begin{aligned}
\mathbb{E}[I_2] &= \mathbb{E}\|\delta(\tilde{\xi}_k, \theta_{k-z})\psi_{\theta_{k-z}}^i - \delta(\xi_k, \theta_{k-z})\psi_{\theta_{k-z}}^i\| \\
&\leq 2 \sup_\xi \|\delta(\xi, \theta_{k-z})\psi_{\theta_{k-z}^i}\| d_{TV}(P(\tilde{\xi}_k \in \cdot | \theta_{k-z}, s_{k-z}), P(\xi_k \in \cdot | \theta_{k-z}, s_{k-z})) \\
&\leq 2 C_\theta \sum_{m=0}^{z-1} |\mathcal{A}| L_\pi \|\theta_{k-m} - \theta_{k-z}\|,
\end{aligned}
\tag{88}
$$

872 where the last inequality follows Lemma 13.

873 $I_3$ can be bounded as

$$
\begin{aligned}
I_3 &= \mathbb{E}\|\mathbb{E}_{\xi \sim \mu_{\theta_{k-z}}}[\delta(\xi, \theta_{k-z})\psi_{k-z}^i - \delta(\tilde{\xi}_k, \theta_{k-z}\psi_{\theta_{k-z}}^i)]\| \\
&\leq 2 \sup_\xi \|\delta(\xi, \theta_{k-z})\psi_{\theta_{k-z}}^i\| d_{TV}(P(\tilde{\xi} \in \cdot | \theta_{k-z}, s_{k-z}), \mu_{\theta_{k-z}} \otimes \pi_{\theta_{k-z}} \otimes \mathcal{P}) \\
&\leq 2 C_\theta \kappa \rho^{z-1},
\end{aligned}
\tag{89}
$$

874 where the last inequality follows Lemma 5.

875 Plug (87), (88), and (89), we have

$$
\begin{aligned}
&\|\mathbb{E}_{\xi \sim \mu_{\theta_k}}[\delta(\xi, \theta_k)\psi_{\theta_k^i}(s_k, a_k^i)] - \mathbb{E}[\delta(\xi_k, \theta_k)\psi_{\theta_k^i}(s_k, a_k^i)]\| \\
&\leq 2 C_\theta \kappa \rho^{z-1} + 2 C_\delta L_\psi \|\theta_k^i - \theta_{k-z}^i\| + 2 C_\theta \sum_{m=0}^{z-1} |\mathcal{A}| L_\pi \|\theta_{k-m} - \theta_{k-z}\| \\
&\quad + (|\mathcal{A}| L(\log_\rho \kappa^{-1} + (1-\rho)^{-1}) C_\theta + 2(1+\gamma) L_V) \|\theta_k - \theta_{k-z}\|,
\end{aligned}
$$

876 which completes the proof.

877 $\hfill\square$

 # E   Proof of main results

 ## E.1   Proof of Theorem 1

880  In this section, we provide the analysis for i.i.d. sampling. By Lemma 4, we have

$$
\begin{aligned}
\mathbb{E}[J(\theta_{k+1})] - J(\theta_k) &\geq \mathbb{E}[\langle \nabla J(\theta_k), \theta_{k+1} - \theta_k \rangle] - \frac{L}{2}\|\theta_{k+1} - \theta_k\|^2 \\
&= \sum_{i=1}^{N} \mathbb{E}[\langle \nabla_{\theta^i} J(\theta_k), \theta_{k+1}^i - \theta_k^i \rangle] - \frac{L}{2}\sum_{i=1}^{N}\|\theta_{k+1}^i - \theta_k^i\|^2 \\
&= \sum_{i=1}^{N} \mathbb{E}[\alpha_k \langle \nabla_{\theta^i} J(\theta_k), g_a^i(\xi_k, \omega_{k+1}^i, \lambda_{k+1}^i) \rangle] - \frac{L}{2}\alpha_k^2 \sum_{i=1}^{N}\mathbb{E}\|g_a^i(\xi_k, \omega_{k+1}^i, \lambda_{k+1}^i)\|^2 \\
&\geq \sum_{i=1}^{N} [\frac{\alpha_k}{2}\|\nabla_{\theta^i} J(\theta_k)\|^2 + \frac{\alpha_k}{2}\|\mathbb{E}[g_a^i(\xi_k, \omega_{k+1}^i, \lambda_{k+1}^i)]\|^2 \\
&\quad - \frac{\alpha_k}{2}\|\nabla_{\theta^i} J(\theta_k) - \mathbb{E}[g_a^i(\xi_k, \omega_{k+1}^i, \lambda_{k+1}^i)]\|^2] - \frac{L}{2}NC_\theta^2 \alpha_k^2, \quad\quad (90)
\end{aligned}
$$

881  where the last inequality is due to $\|g_a^i(\xi_k, \omega_{k+1}^i, \lambda_{k+1}^i)\| = \|\hat{\delta}(\xi_k, \omega_k^i, \lambda_k^i)\psi_{\theta_k^i}(s_k, a_k^i)\| \leq C_\delta C_\psi :=$
882  $C_\theta$.

883  For brevity, we will use $\psi_{\theta_k^i}$ to represent $\psi_{\theta_k^i}(s_k, a_k^i)$. The gradient bias can be bounded as

$$
\begin{aligned}
&\|\nabla_{\theta^i} J(\theta_k) - \mathbb{E}[g_a^i(\xi_k, \omega_{k+1}^i, \lambda_{k+1}^i)|\omega_{k+1}^i, \lambda_{k+1}^i]\|^2 \\
&\leq 4\underbrace{\|\nabla_{\theta^i} J(\theta_k) - \mathbb{E}[\delta(\xi_k, \theta_k)\psi_{\theta_k^i}]\|^2}_{I_1} \\
&\quad + 4\underbrace{\|\mathbb{E}[(\delta(\xi_k, \theta_k) - \tilde{\delta}(\xi_k, \omega^*(\theta_k)))\psi_{\theta_k^i}]\|^2}_{I_2} \\
&\quad + 4\underbrace{\|\mathbb{E}[(\tilde{\delta}(\xi_k, \omega^*(\theta_k)) - \tilde{\delta}(\xi_k, \omega_{k+1}^i))\psi_{\theta_k^i}]\|^2}_{I_3} \\
&\quad + 4\underbrace{\|\mathbb{E}[(\tilde{\delta}(\xi_k, \omega_{k+1}^i) - \hat{\delta}(\xi_k, \omega_{k+1}^i, \lambda_{k+1}^i))\psi_{\theta_k^i}]\|^2}_{I_4}, \quad\quad (91)
\end{aligned}
$$

884  where the inequality uses $\|a + b + c + c\|^2 \leq 4\|a\|^2 + 4\|b\|^2 + 4\|c\|^2 + 4\|d\|^2$.

885  From now on, we will use $\xi \sim d_\theta$ to denote $s \sim d_{\pi_\theta}, a \sim \pi(\cdot|s), s' \sim \mathcal{P}$ for notational simplicity.

886  $I_1$ reflects the sampling error under perfect value function estimation of critic. It can be bounded as

$$
\begin{aligned}
\mathbb{E}[I_1|\theta_k] &= \|\nabla_{\theta^i} J(\theta_k) - \mathbb{E}[\delta(\xi_k, \theta_k)\psi_{\theta_k^i}|\theta_k]\|^2 \\
&= \|\mathbb{E}_{\xi \sim d_{\theta_k}}[\delta(\xi, \theta_k)\psi_{\theta_k^i}|\theta_k] - \mathbb{E}_{\xi \sim \mu_{\theta_k}}[\delta(\xi, \theta_k)\psi_{\theta_k^i}|\theta_k]\|^2 \\
&\leq (2\sup_\xi |\bar{r}(s, a) + \gamma V_{\theta_k}(s') - V_{\theta_k}(s)| \, d_{TV}(\mu_{\theta_k} \otimes \pi_{\theta_k} \otimes \mathcal{P}, d_{\theta_k} \otimes \pi_{\theta_k} \otimes \mathcal{P}))^2 \\
&\leq (2r_{\max} C_\psi d_{TV}(\mu_{\theta_k}, d_{\theta_k}))^2 \\
&\leq 16C_\theta^2 (\log_\rho \kappa^{-1} + \frac{1}{\rho})^2 (1 - \gamma^2),
\end{aligned}
$$

887  where the last inequality follows Lemma 6.
888  Define $\varepsilon_{sp} := 4C_\theta^2 (\log_\rho \kappa^{-1} + \frac{1}{\rho})^2 (1 - \gamma^2)$, then $I_1$ can be bounded as

$$
I_1 \leq 4\varepsilon_{sp}. \quad\quad (92)
$$

The term $I_2$ describe the approximation quality of linear function class, it can be bounded as

$$
I_2 = \|\mathbb{E}[(\delta(\xi_k, \theta_k) - \tilde{\delta}(\xi_k, \omega^*(\theta_k)))\psi_{\theta_k^i}]\|^2
$$

$$
\overset{(i)}{\leq} \mathbb{E}[|\delta(\xi_k, \theta_k) - \tilde{\delta}(\xi_k, \omega^*(\theta_k))|^2 \|\psi_{\theta_k^i}\|^2]
$$

$$
\overset{(ii)}{\leq} C_\psi^2 \mathbb{E}[|\gamma(V_{\theta_k}(s_{k+1}) - \phi(s_{k+1})^T \omega^*(\theta_k)) + (V_{\theta_k}(s_k) - \phi(s_k)^T \omega^*(\theta_k))|^2]
$$

$$
\overset{(iii)}{\leq} C_\psi^2 (2\mathbb{E}[\gamma^2(V_{\theta_k}(s_{k+1}) - \phi(s_{k+1})^T \omega^*(\theta_k))^2] + 2\mathbb{E}[(V_{\theta_k}(s_k) - \phi(s_k)^T \omega^*(\theta_k))^2])
$$

$$
\overset{(iiii)}{\leq} 2C_\psi^2(1 + \gamma^2)\varepsilon_{app}^c \leq 4C_\psi^2 \varepsilon_{app}^c. \tag{93}
$$

where $(i)$ applies triangle inequality and Cauchy Schwarz inequality, $(ii)$ follows Assumption 3, $(iii)$ uses $\|a + b\|^2 \leq 2\|a\|^2 + 2\|b\|^2$, and $(iiii)$ follows the definition of $\varepsilon_{app}^c :=$ $\max_{\theta,a} \sqrt{\mathbb{E}_{s \sim \mu_\theta}[|V_{\pi_\theta}(s) - \hat{V}_{\omega^*(\theta)}(s)|^2]}$.

$I_3$ can be bounded as

$$
\mathbb{E}[I_3] = \|\mathbb{E}[(\tilde{\delta}(\xi_k, \omega^*(\theta_k)) - \tilde{\delta}(\xi_k, \omega_{k+1}^i))\psi_{\theta_k^i}]\|^2
$$

$$
\leq \mathbb{E}[|\tilde{\delta}(\xi_k, \omega^*(\theta_k)) - \tilde{\delta}(\xi_k, \omega_{k+1}^i)|^2 \|\psi_{\theta_k^i}\|^2]
$$

$$
\leq C_\psi^2 \mathbb{E}[|\gamma\phi(s_k + 1)^T(\omega^*(\theta_k) - \omega_{k+1}^i) - \phi(s_k)^T(\omega^*(\theta_k) - \omega_{k+1}^i)|^2]
$$

$$
\leq C_\psi^2 (2\mathbb{E}[|\gamma\phi(s_{k+1})^T(\omega^*(\theta_k) - \omega_{k+1}^i)|^2] + 2\mathbb{E}[|\phi(s_k)^T(\omega^*(\theta_k - \omega_{k+1}^i))|^2])
$$

$$
\leq C_\psi^2 (2\gamma^2\mathbb{E}[\|\phi(s_{k+1})\|^2\|\omega^*(\theta_k) - \omega_{k+1}^i\|^2] + 2\mathbb{E}[\|\phi(s_k)\|^2\|\omega^*(\theta_k) - \omega_{k+1}^i\|^2])
$$

$$
\overset{(i)}{\leq} 2C_\psi^2(1 + \gamma^2)\|\omega^*(\theta_k) - \omega_{k+1}^i\|^2 \leq 4C_\psi^2\|\omega^*(\theta_k) - \omega_{k+1}^i\|^2. \tag{94}
$$

where the last inequality is due to $\|\phi(s)\| \leq 1$, as specified by Assumption 1.

$I_4$ can be bounded as

$$
\mathbb{E}[I_4] = \|\mathbb{E}[(\tilde{\delta}(\xi_k, \omega_{k+1}^i) - \hat{\delta}(\xi_k, \omega_{k+1}^i, \lambda_{k+1}^i))\psi_{\theta_k^i}|\lambda_{k+1}^i]\|^2
$$

$$
\leq \mathbb{E}[|\tilde{\delta}(\xi_k, \omega_{k+1}^i) - \hat{\delta}(\xi_k, \omega_{k+1}^i, \lambda_{k+1}^i)|^2 \|\psi_{\theta_k^i}\|^2|\lambda_{k+1}^i]
$$

$$
\leq C_\psi^2 \mathbb{E}[|\bar{r}(s_k, a_k) - \varphi(s_k, a_k)^T\lambda_{k+1}^i|^2|\lambda_{k+1}^i]
$$

$$
\leq C_\psi^2 (2\mathbb{E}[|\bar{r}(s_k, a_k) - \varphi(s_k, a_k)^T\lambda^*(\theta_k)|^2] + 2\mathbb{E}[|\varphi(s_k, a_k)^T\lambda^*(\theta_k) - \varphi(s_k, a_k)^T\lambda_{k+1}^i|^2|\lambda_{k+1}^i])
$$

$$
\leq 2C_\psi^2\varepsilon_{app}^r + 2C_\psi^2\|\lambda^*(\theta_k) - \lambda_{k+1}^i\|^2 \tag{95}
$$

Thus, the gradient bias for $i_{th}$ agent can be bounded as

$$
\|\nabla_{\theta^i} F(\theta_k) - \mathbb{E}[g_a^i(\xi_k, \omega_{k+1}^i, \lambda_{k+1}^i)]\|^2
$$

$$
\leq 16\varepsilon_{sp} + 16C_\psi^2\varepsilon_{app}^c + 16C_\psi^2\|\omega^*(\theta_k) - \omega_{k+1}^i\|^2
$$

$$
+ 8C_\psi^2\varepsilon_{app}^r + 8C_\psi^2\|\lambda^*(\theta_k) - \lambda_{k+1}^i\|^2
$$

$$
\leq 16(\varepsilon_{sp} + C_\psi^2\varepsilon_{app}) + 16C_\psi^2\|\omega^*(\theta_k) - \omega_{k+1}^i\|^2 + 8C_\psi^2\|\lambda^*(\theta_k) - \lambda_{k+1}^i\|^2, \tag{96}
$$

where the last inequality follows the definition of $\varepsilon_{app}$.

Plug (96) into (90) gives us

$$
\mathbb{E}[J(\theta_{k+1})] - J(\theta_k) \geq \sum_{i=1}^N (\frac{\alpha_k}{2}\mathbb{E}\|\nabla_{\theta^i}J(\theta_k)\|^2 + \frac{\alpha_k}{2}\mathbb{E}\|g_a^i(\xi_k, \omega_{k+1}^i, \lambda_{k+1}^i)\|^2
$$

$$
- 8C_\psi^2\alpha_k\mathbb{E}\|\omega^*(\theta_k) - \omega_{k+1}^i\|^2 - 4C_\psi^2\alpha_k\mathbb{E}\|\lambda^*(\theta_k) - \lambda_{k+1}^i\|^2)
$$

$$
- \frac{L}{2}NC_\theta^2\alpha_k^2 - 8(\varepsilon_{sp} + C_\psi^2\varepsilon_{app})N\alpha_k. \tag{97}
$$

Consider the Lyapunov function

$$
\mathbb{V}_k := -J(\theta_k) + \|\bar{\omega}_k - \omega^*(\theta_k)\|^2 + \|\bar{\lambda}_k - \lambda^*(\theta_k)\|^2.
$$

The difference between two Lyapunov functions will be

$$\mathbb{E}[\mathbb{V}_{k+1}] - \mathbb{E}[\mathbb{V}_k] = \mathbb{E}[J(\theta_k)] - \mathbb{E}[J(\theta_{k+1})] + \mathbb{E}\|\bar{\omega}_{k+1} - \omega^*(\theta_{k+1})\|^2 - \mathbb{E}\|\bar{\omega}_k - \omega^*(\theta_k)\|^2$$
$$+ \mathbb{E}\|\bar{\lambda}_{k+1} - \lambda^*(\theta_k)\|^2 - \mathbb{E}\|\bar{\lambda}_k - \lambda^*(\theta_k)\|^2$$
$$\leq \sum_{i=1}^{N} \left( -\frac{\alpha_k}{2}\|\nabla_{\theta^i} J(\theta_k)\|^2 - \frac{\alpha_k}{2}\mathbb{E}\|g_a^i(\xi_k, \omega_{k+1}^i)\|^2 \right) + \frac{L}{2}NC_\theta^2\alpha_k^2 + 8(\varepsilon_{sp} + C_\psi^2\varepsilon_{app})N\alpha_k$$
$$+ \underbrace{\sum_{i=1}^{N} 8C_\psi^2\alpha_k\mathbb{E}\|\omega^*(\theta_k) - \omega_{k+1}^i\|^2 + \mathbb{E}\|\bar{\omega}_{k+1} - \omega^*(\theta_{k+1})\|^2 - \mathbb{E}\|\bar{\omega}_k - \omega^*(\theta_k)\|^2}_{I_5}$$
$$+ \underbrace{\sum_{i=1}^{N} 4C_\psi^2\alpha_k\mathbb{E}\|\lambda^*(\theta_k) - \lambda_{k+1}^i\|^2 + \mathbb{E}\|\bar{\lambda}_{k+1} - \lambda^*(\theta_{k+1})\|^2 - \mathbb{E}\|\bar{\lambda}_k - \lambda^*(\theta_k)\|^2}_{I_6}$$

$$(98)$$

The first two terms of $I_5$ can be bounded as

$$\sum_{i=1}^{N} 8C_\psi^2\alpha_k\mathbb{E}\|\omega^*(\theta_k) - \bar{\omega}_{k+1} + \bar{\omega}_{k+1} - \omega_{k+1}^i\|^2 + \mathbb{E}\|\bar{\omega}_{k+1} - \omega^*(\theta_{k+1})\|^2$$
$$= \sum_{i=1}^{N} 8C_\psi^2\alpha_k\mathbb{E}\|\bar{\omega}_{k+1} - \omega_{k+1}^i\|^2 + 8C_\psi^2\alpha_k\mathbb{E}\|\bar{\omega}_{k+1} - \omega^*(\theta_k)\|^2 + \mathbb{E}\|\bar{\omega}_{k+1} - \omega^*(\theta_{k+1})\|^2$$
$$\leq 8C_\psi^2\alpha_k\left(\nu^{2k}\|\boldsymbol{\omega}_0\|_F + \frac{16NC_\delta^2}{1-\nu}\beta_k^2 + \frac{8\sqrt{N}C_\delta\|\boldsymbol{\omega}_0\|}{1-\nu}\nu^k\beta_k\right)$$
$$+ 8C_\psi^2\alpha_k\mathbb{E}\|\bar{\omega}_{k+1} - \omega^*(\theta_k)\|^2 + \mathbb{E}\|\bar{\omega}_{k+1} - \omega^*(\theta_{k+1})\|^2, \qquad (99)$$

where the second equality is due to

$$\sum_{i=1}^{N} \langle \omega^*(\theta_k) - \bar{\omega}_{k+1}, \bar{\omega}_{k+1} - \omega_{k+1}^i \rangle = \langle \omega^*(\theta_k) - \bar{\omega}_{k+1}, \bar{\omega}_{k+1} - \bar{\omega}_{k+1} \rangle = 0,$$

and the last inequality follows the Lemma 21.

For the ease of expression, we define

$$M_{k_1} := 8C_\psi^2\left(\nu^{2k}\|\boldsymbol{\omega}_0\|_F + \frac{16NC_\delta^2}{1-\nu}\beta_k^2 + \frac{8\sqrt{N}C_\delta\|\boldsymbol{\omega}_0\|_F}{1-\nu}\nu^k\beta_k\right). \qquad (100)$$

Plug (100) into (99), we have

$$I_5 \leq 8C_\psi^2\alpha_k\mathbb{E}\|\bar{\omega}_{k+1} - \omega^*(\theta_k)\|^2 + \mathbb{E}\|\bar{\omega}_{k+1} - \omega^*(\theta_{k+1})\|^2 + \alpha_k M_{k_1}$$
$$\leq \left(1 + 4L_{\omega,2}^2 N\alpha_k + 8C_\psi^2\alpha_k + \frac{L_{\omega,2}^2}{2}C_\theta^2 N^2\alpha_k^2\right)\mathbb{E}\|\bar{\omega}_{k+1} - \omega^*(\theta_k)\|^2$$
$$+ \left(\frac{L_{\omega,2}^2 C_\theta^2 N^2}{2} + L_\omega^2\right)\alpha_k^2 + \frac{\alpha_k}{4}\sum_{i=1}^{N}\|\mathbb{E}[g_a^i(\xi_k, \omega_{k+1}^i, \lambda_{k+1}^i)]\|^2 + \alpha_k M_{k_1}, \qquad (101)$$

where the second inequality follows (31) in Lemma 15.

907  Let $C_9 := \min\{c \mid 4L_{\omega,2}^2 N\alpha_k + 8C_\psi^2\alpha_k + \frac{L_{\omega,2}^2}{2}C_\theta^2 N^2\alpha_k^2 \le c\alpha_k\}$. Plug the definition into (101),
908  we get

$$
\begin{aligned}
I_5 &\le (1 + C_9\alpha_k)\mathbb{E}\|\bar{\omega}_{k+1} - \omega^*(\theta_k)\|^2 + (\frac{L_{\omega,2}^2 C_\theta^2 N^2}{2} + L_\omega^2)\alpha_k^2 \\
&\quad + \frac{\alpha_k}{4}\sum_{i=1}^N \|\mathbb{E}[g_a^i(\xi_k, \omega_{k+1}^i, \lambda_{k+1}^i)]\|^2 + \alpha_k M_{k_1} \\
&\le (1 + C_9\alpha_k)(1 - 2\lambda_\phi\beta_k)\mathbb{E}\|\bar{\omega}_{k+1} - \omega^*(\theta_k)\|^2 + (1 + C_9\alpha_k)C_\delta^2\beta_k^2 \\
&\quad + (\frac{L_{\omega,2}^2 C_\theta^2 N^2}{2} + L_\omega^2)\alpha_k^2 + \frac{\alpha_k}{4}\sum_{i=1}^N \|\mathbb{E}[g_a^i(\xi_k, \omega_{k+1}^i, \lambda_{k+1}^i)]\|^2 + \alpha_k M_{k_1}, \quad (102)
\end{aligned}
$$

909  where the last inequality follows (32) in Lemma 15.

910  By letting $\beta_k = \frac{C_9}{2\lambda_\phi}\alpha_k$, we can ensure

$$(1 + C_9\alpha_k)(1 - 2\lambda_\phi\beta_k) < 0.$$

911  Therefore, $I_5$ can be bounded as

$$
I_5 \le (1 + C_9\alpha_k)C_\delta^2\beta_k^2 + \frac{\alpha_k}{4}\sum_{i=1}^N \|\mathbb{E}[g_a^i(\xi_k, \omega_{k+1}^i, \lambda_{k+1}^i)]\|^2 + \alpha_k M_{k_1} + (\frac{L_{\omega,2}^2 C_\theta^2 N^2}{2} + L_\omega^2)\alpha_k^2.
\tag{103}
$$

912  By applying Lemma 18 and following the similar procedure, we can bound $I_6$ as

$$
I_6 \le (1 + C_{10}\alpha_k)C_\lambda^2\eta_k^2 + \frac{\alpha_k}{4}\sum_{i=1}^N \|\mathbb{E}[g_a^i(\xi_k, \omega_{k+1}^i, \lambda_{k+1}^i)]\|^2 + \alpha_k M_{k_2} + (\frac{L_{\lambda,2}^2 C_\theta^2 N^2}{2} + L_\lambda^2)\alpha_k^2,
\tag{104}
$$

913  with $\eta_k = \frac{C_{10}}{2\lambda_\varphi}\alpha_k$ and

$$
\begin{aligned}
C_{10} &:= \min\{c \mid 4\frac{L_{\lambda,2}^2}{2}C_\theta^2\alpha_k + 8C_\psi^2\alpha_k + \frac{L_{\lambda,2}^2 C_\delta^2}{2}\alpha_k^2 \le c\alpha_k\}, \\
M_{k_2} &:= 8C_\psi^2(\nu^{2k}\|\boldsymbol{\lambda}_0\|_F + \frac{16NC_\lambda^2}{1-\nu}\eta_k^2 + \frac{8\sqrt{N}C_\lambda\|\boldsymbol{\lambda}_0\|_F}{1-\nu}\nu^k\eta_k). \quad (105)
\end{aligned}
$$

914  Plug (103) and (104) into (98), we have

$$
\begin{aligned}
\mathbb{E}[\mathbb{V}_{k+1}] - \mathbb{E}[\mathbb{V}_k] &\le \sum_{i=1}^N (-\frac{\alpha_k}{2}\|\nabla_{\theta^i}J(\theta_k)\|^2 - \frac{\alpha_k}{2}\mathbb{E}\|g_a^i(\xi_k, \omega_{k+1}^i)\|^2) + \frac{\alpha_k}{2}\sum_{i=1}^N \|\mathbb{E}[g_a^i(\xi_k, \omega_{k+1}^i, \lambda_{k+1}^i)]\|^2 \\
&\quad + (1 + C_9\alpha_k)C_\delta^2\beta_k^2 + (1 + C_{10}\alpha_k)C_\lambda^2\eta_k^2 + (\frac{L}{2}NC_\theta^2 + C_{11})\alpha_k^2 \\
&\quad + (M_{k_1} + M_{k_2})\alpha_k + 8(\varepsilon_{sp} + C_\psi^2\varepsilon_{app}N)\alpha_k, \\
&= \sum_{i=1}^N (-\frac{\alpha_k}{2}\|\nabla_{\theta^i}J(\theta_k)\|^2) + (M_{k_1} + M_{k_2})\alpha_k + 8(\varepsilon_{sp} + C_\psi^2\varepsilon_{app}N)\alpha_k \\
&\quad + (1 + C_9\alpha_k)C_\delta^2\beta_k^2 + (1 + C_{10}\alpha_k)C_\lambda^2\eta_k^2 + (\frac{L}{2}NC_\theta^2 + C_{11})\alpha_k^2, \quad (106)
\end{aligned}
$$

915  where $C_{11} := \frac{L_{\omega,2}^2 C_\theta^2 N^2}{2} + \frac{L_{\lambda,2}^2 C_\theta^2 N^2}{2} + L_\omega^2 + L_\lambda^2$.

916  By telescoping (106), we get

$$
\begin{aligned}
\frac{1}{K}\sum_{k=0}^K \sum_{i=1}^N \mathbb{E}\|\nabla_{\theta^i}J(\theta_k)\|^2 &\le \frac{2\mathbb{E}[\mathbb{V}_0]}{K\alpha_k} + 16(\varepsilon_{sp} + C_\psi^2\varepsilon_{app}N) + \frac{2}{K}\sum_{k=0}^K (M_{k_1} + M_{k_2}) \\
&\quad + (1 + C_9\alpha_k)C_\delta^2\frac{\beta_k^2}{\alpha_k} + (1 + C_{10}\alpha_k)C_\lambda^2\frac{\eta_k^2}{\alpha_k} + (\frac{L}{2}NC_\theta^2 + C_{11})\alpha_k. \quad (107)
\end{aligned}
$$

The third term can be bounded as

$$
\frac{2}{K} \sum_{k=0}^{K} (M_{k_1} + M_{k_2})
$$

$$
= \frac{16C_\psi^2}{K}(\|\boldsymbol{\omega}_0\|_F + \|\boldsymbol{\lambda}_0\|_F) \sum_{k=1}^{K} \nu^{2k} + \frac{256NC_\psi^2}{(1-\nu)K} \sum_{k=0}^{K} (C_\delta^2 \beta_k^2 + C_\lambda^2 \eta_k^2)
$$

$$
\quad + \frac{128\sqrt{N}C_\psi^2}{(1-\nu)K} (\sum_{k=1}^{K} C_\delta \|\boldsymbol{\omega}_0\|_F \nu^k \beta_k + \sum_{k=1}^{K} C_\lambda \|\boldsymbol{\lambda}_0\|_F \nu^k \eta_k)
$$

$$
\leq \frac{16C_\psi^2}{K(1-\nu^2)}(\|\boldsymbol{\omega}_0\|_F + \|\boldsymbol{\lambda}_0\|_F) + \frac{256NC_\psi^2}{(1-\nu)}(C_\delta^2 \beta_k^2 + C_\lambda^2 \eta_k^2)
$$

$$
\quad + \frac{128\sqrt{N}C_\psi^2}{(1-\nu)^2 K}(C_\delta \|\boldsymbol{\omega}_0\|_F \beta_k + C_\lambda \|\boldsymbol{\lambda}_0\|_F \eta_k)
$$

$$
= o(\frac{1}{\sqrt{K}}), \tag{108}
$$

where we use $\sum_{k=0}^{K} \nu^k \leq \frac{1}{1-\nu}$ for the inequality.

Plug (108) back into (107) and let $\alpha_k = \frac{\bar{\alpha}}{\sqrt{K}}$ for some positive constant $\bar{\alpha}$, $\beta_k = \frac{C_9}{2\lambda_\phi}\alpha_k$, $\eta_k = \frac{C_{10}}{2\lambda_\varphi}\alpha_k$, we obtain the desired result.

## E.2   Proof of Theorem 2

Following the proof under i.i.d. sampling in (90), we have

$$
\mathbb{E}[J(\theta_{k+1})] - J(\theta_k)
$$

$$
\geq \sum_{i=1}^{N} [\frac{\alpha_k}{2} \|\nabla_{\theta^i} J(\theta_k)\|^2 + \frac{\alpha_k}{2} \|\mathbb{E}[g_a^i(\xi_k, \omega_{k+1}^i, \lambda_{k+1}^i)]\|^2
$$

$$
\quad - \frac{\alpha_k}{2} \|\nabla_{\theta^i} J(\theta_k) - \mathbb{E}[g_a^i(\xi_k, \omega_{k+1}^i, \lambda_{k+1}^i)]\|^2 - \frac{L}{2}NC_\theta^2\alpha_k^2. \tag{109}
$$

By following the derivation of (91), the gradient bias can be bounded as (crf. $\psi_{\theta_k^i} := \psi_{\theta_k^i}(s_k, a_k^i)$)

$$
\|\nabla_{\theta^i} J(\theta_k) - \mathbb{E}[g_a^i(\xi_k, \omega_{k+1}^i, \lambda_{k+1}^i)|\omega_{k+1}^i, \lambda_{k+1}^i]\|^2
$$

$$
\leq 4 \underbrace{\|\nabla_{\theta^i} J(\theta_k) - \mathbb{E}[\delta(\xi_k, \theta_k)\psi_{\theta_k^i}]\|^2}_{I_1}
$$

$$
\quad + 4 \underbrace{\|\mathbb{E}[(\delta(\xi_k, \theta_k) - \tilde{\delta}(\xi_k, \omega^*(\theta_k)))\psi_{\theta_k^i}]\|^2}_{I_2}
$$

$$
\quad + 4 \underbrace{\|\mathbb{E}[(\tilde{\delta}(\xi_k, \omega^*(\theta_k)) - \tilde{\delta}(\xi_k, \omega_{k+1}^i))\psi_{\theta_k^i}]\|^2}_{I_3}
$$

$$
\quad + 4 \underbrace{\|\mathbb{E}[(\tilde{\delta}(\xi_k, \omega_{k+1}^i) - \hat{\delta}(\xi_k, \omega_{k+1}^i, \lambda_{k+1}^i))\psi_{\theta_k^i}]\|^2}_{I_4}, \tag{110}
$$

We bound $I_1$ as

$$\begin{aligned}
I_1 &= \|\nabla_{\theta^i} J(\theta_k) - \mathbb{E}[\delta(\xi_k, \theta_k)\psi_{\theta_k^i}|\theta_k]\|^2 \\
&= \|\mathbb{E}_{\xi \sim d_{\theta_k}}[\delta(\xi, \theta_k)\psi_{\theta_k^i}|\theta_k] - \mathbb{E}[\delta(\xi_k, \theta_k)\psi_{\theta_k^i}|\theta_k]\|^2 \\
&\leq 2 \underbrace{\|\mathbb{E}_{\xi \sim d_{\theta_k}}[\delta(\xi, \theta_k)\psi_{\theta_k^i}|\theta_k] - \mathbb{E}_{\xi \sim \mu_{\theta_k}}[\delta(\xi, \theta_k)\psi_{\theta_k^i}|\theta_k]\|^2}_{I_1^{(1)}} \\
&\quad + 2 \underbrace{\|\mathbb{E}_{\xi \sim \mu_\theta}[\delta(\xi, \theta_k)\psi_{\theta_k^i}|\theta_k] - \mathbb{E}[\delta(\xi_k, \theta_k)\psi_{\theta_k^i}|\theta_k]\|^2}_{I_1^{(2)}}
\end{aligned} \tag{111}$$

Follow the derivation of (92), we have

$$I_1^{(1)} \leq 4\varepsilon_{sp}.$$

By Lemma 22, $I_1^{(2)}$ can be bounded as

$$\begin{aligned}
I_1^{(2)} &\leq (2C_\theta \kappa \rho^{z-1} + C_{12}\sum_{m=0}^{z-1}\|\theta_{k-m} - \theta_{k-z}\| + C_{13}\|\theta_k - \theta_{k-z}\| + C_{14}\|\theta_k^i - \theta_{k-z}^i\|)^2 \\
&\leq (2C_\theta \kappa \rho^{z-1} + C_{12}\sum_{m=0}^{z-1}\sum_{n=1}^{z-m}\|\theta_{k-m-n+1} - \theta_{k-m}\| + C_{13}\sum_{n=1}^{z}\|\theta_{k-n+1} - \theta_{k-n}\| + C_{14}\sum_{n=1}^{z}\|\theta_{k-n+1}^i - \theta_{k-n}^i\|)^2 \\
&\leq (2C_\theta \kappa \rho^{z-1} + C_{12}NC_\theta\frac{z(z+1)}{2}\alpha_{k-z} + C_{13}NzC_\theta\alpha_{k-z} + C_{14}zC_\theta\alpha_{k-z})^2 \\
&\leq 16C_\theta^2\kappa^2\rho^{2z-2} + 2C_{12}^2C_\theta^2 z^2\alpha_{k-z}^2 + 4C_{13}^2N^2z^2C_\theta^2\alpha_{k-z}^2 + 4C_{14}^2z^2C_\theta^2\alpha_{k-z}^2, \tag{112}
\end{aligned}$$

where the second inequality uses triangle inequality, and the last inequality applies $(a+b+c+d)^2 \leq 4a^2 + 4b^2 + 4c^2 + 4d^2$.

Let $z = Z_K$. Recall $Z_K$ is defined as $Z_K := \min\{z \in \mathbb{N}^+ | \kappa\rho^{z-1} \leq \min\{\alpha_k, \beta_k, \eta_k\}\}$. Then we have

$$I_1^{(2)} \leq C_{K_5}\alpha_{k-Z_K}^2, \tag{113}$$

where we define $C_{K_5} := 16C_\theta^2 + 2C_{12}^2C_\theta^2 Z_K^2 + 4C_{13}^2N^2Z_K^2C_\theta^2 + 4C_{14}^2Z_K^2C_\theta^2$.

Thus, we have

$$I_1 \leq 4\varepsilon_{sp} + C_{K_5}\alpha_{k-Z_K}^2. \tag{114}$$

The bound of $I_2, I_3$, and $I_4$ follows the analysis under i.i.d. sampling. Plug in (93), (94), and (95) will give us the bound of gradient bias

$$\begin{aligned}
&\|\nabla_{\theta^i} F(\theta_k) - \mathbb{E}[g_a^i(\xi_k, \omega_{k+1}^i, \lambda_{k+1}^i)]\|^2 \\
&\leq 16(\varepsilon_{sp} + C_\psi^2\varepsilon_{app}) + 16C_\psi^2\|\omega^*(\theta_k) - \omega_{k+1}^i\|^2 \\
&\quad + 8C_\psi^2\|\lambda^*(\theta_k) - \lambda_{k+1}^i\|^2 + 4C_{K_5}\alpha_{k-Z_K}^2.
\end{aligned}$$

Thus, we have

$$\begin{aligned}
\mathbb{E}[J(\theta_{k+1})] - J(\theta_k) &\geq \sum_{i=1}^{N}(\frac{\alpha_k}{2}\mathbb{E}\|\nabla_{\theta^i} J(\theta_k)\|^2 + \frac{\alpha_k}{2}\mathbb{E}\|g_a^i(\xi_k, \omega_{k+1}^i, \lambda_{k+1}^i)\|^2 \\
&\quad - 8C_\psi^2\alpha_k\mathbb{E}\|\omega^*(\theta_k) - \omega_{k+1}^i\|^2 - 4C_\psi^2\alpha_k\mathbb{E}\|\lambda^*(\theta_k) - \lambda_{k+1}^i\|^2) \\
&\quad - \frac{L}{2}NC_\theta^2\alpha_k^2 - 2NC_{K_5}\alpha_{k-Z_K}^2 - 8(\varepsilon_{sp} + C_\psi^2\varepsilon_{app})N\alpha_k. \tag{115}
\end{aligned}$$

Consider the Lyapunov function

$$\mathbb{V}_k := -J(\theta_k) + \|\bar{\omega}_k - \omega^*(\theta_k)\|^2 + \|\bar{\lambda}_k - \lambda^*(\theta_k)\|^2. \tag{116}$$

937 The difference between two Lyapunov functions will be

$$
\begin{aligned}
\mathbb{E}[\mathbb{V}_{k+1}] - \mathbb{E}[\mathbb{V}_k] = {}& \mathbb{E}[J(\theta_k)] - \mathbb{E}[J(\theta_{k+1})] + \mathbb{E}\|\bar{\omega}_{k+1} - \omega^*(\theta_{k+1})\|^2 - \mathbb{E}\|\bar{\omega}_k - \omega^*(\theta_k)\|^2 \\
& + \mathbb{E}\|\bar{\lambda}_{k+1} - \lambda^*(\theta_k)\|^2 - \mathbb{E}\|\bar{\lambda}_k - \lambda^*(\theta_k)\|^2 \\
\leq {}& \sum_{i=1}^N (-\frac{\alpha_k}{2}\|\nabla_{\theta^i} J(\theta_k)\|^2 - \frac{\alpha_k}{2}\mathbb{E}\|g_a^i(\xi_k, \omega_{k+1}^i)\|^2) \\
& + 2NC_{K_5}\alpha_{k-Z_K} + \frac{L}{2}NC_\theta^2\alpha_k^2 + 8(\varepsilon_{sp} + C_\psi^2\varepsilon_{app})N\alpha_k \\
& + \underbrace{\sum_{i=1}^N 8C_\psi^2\alpha_k\mathbb{E}\|\omega^*(\theta_k) - \omega_{k+1}^i\|^2 + \mathbb{E}\|\bar{\omega}_{k+1} - \omega^*(\theta_{k+1})\|^2 - \mathbb{E}\|\bar{\omega}_k - \omega^*(\theta_k)\|^2}_{I_5} \\
& + \underbrace{\sum_{i=1}^N 4C_\psi^2\alpha_k\mathbb{E}\|\lambda^*(\theta_k) - \lambda_{k+1}^i\|^2 + \mathbb{E}\|\bar{\lambda}_{k+1} - \lambda^*(\theta_{k+1})\|^2 - \mathbb{E}\|\bar{\lambda}_k - \lambda^*(\theta_k)\|^2}_{I_6}.
\end{aligned}
\tag{117}
$$

938 The first two terms of $I_5$ can be bounded as

$$
\begin{aligned}
& \sum_{i=1}^N 8C_\psi^2\alpha_k\mathbb{E}\|\omega^*(\theta_k) - \bar{\omega}_{k+1} + \bar{\omega}_{k+1} - \omega_{k+1}^i\|^2 + \mathbb{E}\|\bar{\omega}_{k+1} - \omega^*(\theta_{k+1})\|^2 \\
& = \sum_{i=1}^N 8C_\psi^2\alpha_k\mathbb{E}\|\bar{\omega}_{k+1} - \omega_{k+1}^i\|^2 + 8C_\psi^2\alpha_k\mathbb{E}\|\bar{\omega}_{k+1} - \omega^*(\theta_k)\|^2 + \mathbb{E}\|\bar{\omega}_{k+1} - \omega^*(\theta_{k+1})\|^2 \\
& \leq 8C_\psi^2\alpha_k\mathbb{E}\|\bar{\omega}_{k+1} - \omega^*(\theta_k)\|^2 + \mathbb{E}\|\bar{\omega}_{k+1} - \omega^*(\theta_{k+1})\|^2 + \alpha_k M_{k_1} \\
& \leq (1 + 4L_{\omega,2}^2 N\alpha_k + 8C_\psi^2\alpha_k + \frac{L_{\omega,2}^2}{2}C_\theta^2 N^2\alpha_k^2)\mathbb{E}\|\bar{\omega}_{k+1} - \omega^*(\theta_k)\|^2 \\
& \quad + (\frac{L_{\omega,2}^2 C_\theta^2 N^2}{2} + L_\omega^2)\alpha_k^2 + \frac{\alpha_k}{4}\sum_{i=1}^N \|\mathbb{E}[g_a^i(\xi_k, \omega_{k+1}^i, \lambda_{k+1}^i)]\|^2 + \alpha_k M_{k_1},
\end{aligned}
\tag{118}
$$

939 where the equality is due to

$$
\sum_{i=1}^N \langle \omega^*(\theta_k) - \bar{\omega}_{k+1}, \bar{\omega}_{k+1} - \omega_{k+1}^i \rangle = \langle \omega^*(\theta_k) - \bar{\omega}_{k+1}, \bar{\omega}_{k+1} - \bar{\omega}_{k+1} \rangle = 0.
$$

940 The first inequality follows the Lemma 21, with $M_{k_1}$ is defined in (100). The last inequality follows
941 (39) in Lemma 16.

942 Plug (118) into (117), and recall $C_9 := \min\{c \mid 4L_{\omega,2}^2 N\alpha_k + 8C_\psi^2\alpha_k + \frac{L_{\omega,2}^2}{2}C_\theta^2 N^2\alpha_k^2 \leq c\alpha_k\}$, we
943 get

$$
\begin{aligned}
I_5 \leq {}& (1 + C_9\alpha_k)\mathbb{E}\|\bar{\omega}_{k+1} - \omega^*(\theta_k)\|^2 + (\frac{L_{\omega,2}^2 C_\theta^2 N^2}{2} + L_\omega^2)\alpha_k^2 \\
& + \frac{\alpha_k}{4}\sum_{i=1}^N \|\mathbb{E}[g_a^i(\xi_k, \omega_{k+1}^i, \lambda_{k+1}^i)]\|^2 + \alpha_k M_{k_1} \\
\leq {}& (1 + C_9\alpha_k)(1 - 2\lambda_\phi\beta_k)\mathbb{E}\|\bar{\omega}_{k+1} - \omega^*(\theta_k)\|^2 \\
& + (1 + C_9\alpha_k)(C_{K_1}\beta_k\beta_{k-Z_K} + C_{K_2}\beta_k\alpha_{k-Z_K}) \\
& + (\frac{L_{\omega,2}^2 C_\theta^2 N^2}{2} + L_\omega^2)\alpha_k^2 + \frac{\alpha_k}{4}\sum_{i=1}^N \|\mathbb{E}[g_a^i(\xi_k, \omega_{k+1}^i, \lambda_{k+1}^i)]\|^2 + \alpha_k M_{k_1},
\end{aligned}
\tag{119}
$$

944 where the last inequality follows (40) in Lemma 16.

945 By letting $\beta_k = \frac{C_9}{2\lambda_\phi}\alpha_k$, we can ensure

$$(1 + C_9\alpha_k)(1 - 2\lambda_\phi\beta_k) < 0.$$

946 Therefore, $I_5$ can be bounded as

$$I_5 \le \frac{\alpha_k}{4}\sum_{i=1}^{N}\|\mathbb{E}[g_a^i(\xi_k, \omega_{k+1}^i, \lambda_{k+1}^i)]\|^2 + \alpha_k M_{k_1} + (\frac{L_{\omega,2}^2 C_\theta^2 N^2}{2} + L_\omega^2)\alpha_k^2$$
$$+ (1 + C_9\alpha_k)(C_{K_1}\beta_k\beta_{k-Z_K} + C_{K_2}\beta_k\alpha_{k-Z_K}). \tag{120}$$

947 By applying Lemma 19 and following the similar procedure, we can bound $I_6$ as

$$I_6 \le \frac{\alpha_k}{4}\sum_{i=1}^{N}\|\mathbb{E}[g_a^i(\xi_k, \omega_{k+1}^i, \lambda_{k+1}^i)]\|^2 + \alpha_k M_{k_2} + (\frac{L_{\lambda,2}^2 C_\theta^2 N^2}{2} + L_\lambda^2)\alpha_k^2$$
$$+ (1 + C_{10}\alpha_k)(C_{K_3}\eta_k\eta_{k-Z_K} + C_{K_4}\eta_k\alpha_{k-Z_K}). \tag{121}$$

948 with $\eta_k = \frac{C_{10}}{2\lambda_\varphi}\alpha_k$, and $M_{k_2}$ defined in (105).

949 Plug (120) and (121) into (117), we have

$$\mathbb{E}[\mathbb{V}_{k+1}] - \mathbb{E}[\mathbb{V}_k] \le \sum_{i=1}^{N} -\frac{\alpha_k}{2}\|\nabla_{\theta^i}J(\theta_k)\|^2 + (M_{k_1} + M_{k_2})\alpha_k$$
$$+ (1 + C_9\alpha_k)(C_{K_1}\beta_k\beta_{k-Z_K} + C_{K_2}\beta_k\alpha_{k-Z_K})$$
$$+ (1 + C_{10}\alpha_k)(C_{K_3}\eta_k\eta_{k-Z_K} + C_{K_4}\eta_k\alpha_{k-Z_K})$$
$$+ (\frac{L}{2}NC_\theta^2 + C_{11})\alpha_k^2 + 8(\varepsilon_{sp} + C_\psi^2\varepsilon_{app}N)\alpha_k, \tag{122}$$

950 where we recall $C_{11} := \frac{L_{\omega,2}^2 C_\theta^2 N^2}{2} + \frac{L_{\lambda,2}^2 C_\theta^2 N^2}{2} + L_\omega^2 + L_\lambda^2$.

951 By letting $\alpha_k = \frac{\bar{\alpha}}{\sqrt{K}}$ for some positive constant $\bar{\alpha}$, and recall $\beta_k = \frac{C_9}{2\lambda_\phi}\alpha_k, \eta_k = \frac{C_{10}}{2\lambda_\varphi}\alpha_k$, we can
952 telescope (122) as

$$\frac{1}{K}\sum_{k=0}^{K}\sum_{i=1}^{N}\mathbb{E}\|\nabla_{\theta^i}J(\theta_k)\|^2 \le \frac{2\mathbb{E}[\mathbb{V}_0]}{K\alpha_k} + 16(\varepsilon_{sp} + C_\psi^2\varepsilon_{app}N) + \frac{2}{K}\sum_{k=0}^{K}(M_{k_1} + M_{k_2})$$
$$+ (1 + C_9\alpha_k)(C_{K_1}\frac{\beta_k}{\alpha_k}\beta_{k-Z_K} + C_{K_2}\frac{\beta_k}{\alpha_k}\alpha_{k-Z_K})$$
$$+ (1 + C_{10}\alpha_k)(C_{K_3}\frac{\eta_k}{\alpha_k}\eta_{k-Z_K} + C_{K_4}\frac{\eta_k}{\alpha_k}\alpha_{k-Z_K})$$
$$+ (\frac{L}{2}NC_\theta^2 + C_{11})\alpha_k. \tag{123}$$

953 The third term can be bounded as

$$\frac{2}{K}\sum_{k=0}^{K}(M_{k_1} + M_{k_2}) = \frac{16C_\psi^2}{K}(\|\boldsymbol{\omega}_0\|_F + \|\boldsymbol{\lambda}_0\|_F)\sum_{k=1}^{K}\nu^{2k} + \frac{256NC_\psi^2}{(1-\nu)K}\sum_{k=0}^{K}(C_\delta^2\beta_k^2 + C_\lambda^2\eta_k^2)$$
$$+ \frac{128\sqrt{N}C_\psi^2}{(1-\nu)K}(\sum_{k=1}^{K}C_\delta\|\boldsymbol{\omega}_0\|_F\nu^k\beta_k + \sum_{k=1}^{K}C_\lambda\|\boldsymbol{\lambda}_0\|_F\nu^k\eta_k)$$
$$\le \frac{16C_\psi^2}{K(1-\nu^2)}(\|\boldsymbol{\omega}_0\|_F + \|\boldsymbol{\lambda}_0\|_F) + \frac{256NC_\psi^2}{(1-\nu)}(C_\delta^2\beta_k^2 + C_\lambda^2\eta_k^2)$$
$$+ \frac{128\sqrt{N}C_\psi^2}{(1-\nu)^2K}(C_\delta\|\boldsymbol{\omega}_0\|_F\beta_k + C_\lambda\|\boldsymbol{\lambda}_0\|_F\eta_k)$$
$$= o(\frac{1}{\sqrt{K}}), \tag{124}$$

954 where we use $\sum_{k=0}^{K} \nu^k \leq \frac{1}{1-\nu}$ for the inequality.

955 Plug (124) back into (123). By noticing $C_{K_1} = \mathcal{O}(\log \frac{1}{\alpha_k}), C_{K_2} = \mathcal{O}(\log^2 \frac{1}{\alpha_k}), C_{K_3} =$
956 $\mathcal{O}(\log \frac{1}{\alpha_k}), C_{K_4} = \mathcal{O}(\log^2 \frac{1}{\alpha_k})$, we obtain the desired result.

### E.3 Proof of Theorem 3

958 Define the update of actor $i$ using the noisy reward as

$$g_a^i(\epsilon_k, \omega_{k+1}^i) := \tilde{r}_{k,K_r}^i(s_k, a_k) + \gamma\phi(s')^T \omega_{k+1}^i - \phi(s)^T \omega_{k+1}^i. \tag{125}$$

959 Following the derivation of (90), we have

$$\mathbb{E}[J(\theta_{k+1}) - J(\theta_k) \geq \sum_{i=1}^{N} [\frac{\alpha_k}{2} \|\nabla_{\theta^i} J(\theta_k)\|^2 + \frac{\alpha_k}{2} \|\mathbb{E}[g_a^i(\xi_k, \omega_{k+1}^i)]\|^2$$

$$- \frac{\alpha_k}{2} \|\nabla_{\theta^i} J(\theta_k) - \mathbb{E}[g_a^i(\xi_k, \omega_{k+1}^i)]\|^2] - \frac{L}{2} N C_\theta^2 \alpha_k^2. \tag{126}$$

960 Similarly to the proof of Theorem 1 and 2, the gradient bias term can be decomposed as as

$$\|\nabla_{\theta^i} J(\theta_k) - \mathbb{E}[g_a^i(\xi_k, \omega_{k+1}^i)]\|^2 \leq 4 \underbrace{\|\nabla_{\theta^i} J(\theta_k) - \mathbb{E}[\delta(\xi_k, \theta_k)\psi_{\theta_k^i}]\|^2}_{I_1}$$

$$+ 4 \underbrace{\|\mathbb{E}[(\delta(\xi_k, \theta_k) - \tilde{\delta}(\xi_k, \omega^*(\theta_k)))\psi_{\theta_k^i}]\|^2}_{I_2}$$

$$+ 4 \underbrace{\|\mathbb{E}[(\tilde{\delta}(\xi_k, \omega^*(\theta_k)) - \tilde{\delta}(\xi_k, \omega_{k+1}^i))\psi_{\theta_k^i}]\|^2}_{I_3}$$

$$+ 4 \underbrace{\|\mathbb{E}[(\bar{r}_k(s_k, a_k) - \tilde{r}_{k,K_r}(s_k, a_k))\psi_{\theta_k^i}]\|^2}_{I_4} \tag{127}$$

961 $I_1, I_2, I_3$ can be bounded following the derivation of (114), (91), and (96), respectively. Plug these
962 bounds into (127), we have

$$\mathbb{E}[J(\theta_{k+1})] - J(\theta_k) \geq \sum_{i=1}^{N} (\frac{\alpha_k}{2} \mathbb{E}\|\nabla_{\theta^i} J(\theta_k)\|^2 + \frac{\alpha_k}{2} \mathbb{E}\|g_a^i(\xi_k, \omega_{k+1}^i)\|^2 - 8C_\psi^2 \alpha_k \mathbb{E}\|\omega^*(\theta_k) - \omega_{k+1}^i\|^2)$$

$$- \sum_{i=1}^{N} \frac{\alpha_k}{2} C_\psi^2 \|\bar{r}_k(s_k, a_k) - \tilde{r}_{k,K_r}^i(s_k, a_k)\|^2 - \frac{L}{2} N C_\theta^2 \alpha_k^2$$

$$- 2N C_{K_5} \alpha_{k-Z_K}^2 - 8(\varepsilon_{sp} + C_\psi^2 \varepsilon_{app}) N \alpha_k. \tag{128}$$

963 Define $\tilde{r}_{k,K_r} := [r_{k,K_r}^1, \cdots, r_{k,K_r}^N]^T$. The reward bias can be bounded as

$$\sum_{i=1}^{N} \|\bar{r}_k(s_k, a_k) - \tilde{r}_{k,K_r}^i(s_k, a_k)\|^2 = \|Q\tilde{r}_{k,K_r}\|^2$$

$$= \|QW^{K_r} \tilde{r}_{k,0}(s_k, a_k)\|^2$$

$$\leq \nu^{2K_r} \|\tilde{r}_{k,0}(s_k, a_k)\|^2$$

$$= \nu^{2K_r} \sum_{i=1}^{N} (\|\tilde{r}_{k,0}^i(s_k, a_k) - \bar{r}_k(s_k, a_k)\|^2 + \|\bar{r}_k(s_k, a_k)\|^2)$$

$$\leq \nu^{2K_r} N(\sigma^2 + r_{\max}), \tag{129}$$

964 where $\sigma^2$ is the variance of the reward noise. Let $K_r = \frac{1}{2} \log_\nu \alpha_k$ and define $C_{15} := \sigma^2 + r_{\max}^2$.
965 Plug (128) back to (127), we have

$$\mathbb{E}[J(\theta_{k+1})] - J(\theta_k) \geq \sum_{i=1}^{N} (\frac{\alpha_k}{2} \mathbb{E}\|\nabla_{\theta^i} J(\theta_k)\|^2 + \frac{\alpha_k}{2} \mathbb{E}\|g_a^i(\xi_k, \omega_{k+1}^i)\|^2 - 8C_\psi^2 \alpha_k \mathbb{E}\|\omega^*(\theta_k) - \omega_{k+1}^i\|^2)$$

$$+ \frac{N}{2} (C_{15} + C_\theta^2 L) \alpha_k^2 - 2N C_{K_5} \alpha_{k-Z_K}^2 - 8(\varepsilon_{sp} + C_\psi^2 \varepsilon_{app}) N \alpha_k.$$

Consider the Lyapunov function

$$\mathbb{V}_k := -J(\theta_k) + \|\bar{\omega}_k - \omega^*(\theta_k)\|^2.$$

The difference between two Lyapunov functions is

$$\mathbb{E}[\mathbb{V}_{k+1}] - \mathbb{E}[\mathbb{V}_k] \leq \sum_{i=1}^{N}(-\frac{\alpha_k}{2}\|\nabla_{\theta^i}J(\theta_k)\|^2 - \frac{\alpha_k}{2}\mathbb{E}\|g_a^i(\xi_k, \omega_{k+1}^i)\|^2)$$
$$+ \frac{N}{2}C_{16}\alpha_k^2 - 2NC_{K_5}\alpha_{k-Z_K}^2 - 8(\varepsilon_{sp} + C_\psi^2\varepsilon_{app})N\alpha_k$$
$$+ \underbrace{\sum_{i=1}^{N}8C_\psi^2\alpha_k\mathbb{E}\|\omega^*(\theta_k) - \omega_{k+1}^i\|^2 + \mathbb{E}\|\bar{\omega}_{k+1} - \omega^*(\theta_{k+1})\|^2 - \mathbb{E}\|\bar{\omega}_k - \omega^*(\theta_k)\|^2}_{I_5}.$$

$I_5$ can be bounded by following the derivation of (120). Thus, we have

$$\mathbb{E}[\mathbb{V}_{k+1}] - \mathbb{E}[\mathbb{V}_k]$$
$$\leq \sum_{i=1}^{N}-\frac{\alpha_k}{2}\|\nabla_{\theta^i}J(\theta_k)\|^2 + \frac{N}{2}C_{16}\alpha_k^2 - 2NC_{K_5}\alpha_{k-Z_K}^2 - 8(\varepsilon_{sp} + C_\psi^2\varepsilon_{app})N\alpha_k$$
$$+ (1 + C_9\alpha_k)(C_{K_1}\beta_k\beta_{k-Z_K} + C_{K_2}\beta_k\alpha_{k-Z_K}) + M_{k_1}\alpha_k, \tag{130}$$

where $C_{16} := C_{15} + C_\theta^2 L + \frac{L_{\omega,2}^2 C_\theta^2 N^2}{2} + L_\omega^2$.

Telescoping (130), we have

$$\frac{1}{K}\sum_{k=0}^{K}\sum_{i=1}^{N}\mathbb{E}\|\nabla_{\theta^i}J(\theta_k)\|^2 \leq \frac{2\mathbb{E}[\mathbb{V}_0]}{K\alpha_k} + 16(\varepsilon_{sp} + C_\psi^2\varepsilon_{app}N) + \frac{2}{K}\sum_{k=0}^{K}M_{k_1} + C_{16}\alpha_k$$
$$+ (1 + C_9\alpha_k)(C_{K_1}\frac{\beta_k}{\alpha_k}\beta_{k-Z_K} + C_{K_2}\frac{\beta_k}{\alpha_k}\alpha_{k-Z_K}).$$

The term $\frac{2}{K}\sum_{k=0}^{K}M_{k_1}$ has been bounded in (124). Let $\alpha_k = \frac{\bar{\alpha}}{\sqrt{K}}$ for some positive constant $\bar{\alpha}$, $\beta_k = \frac{C_\theta}{2\lambda_\phi}\alpha_k$ will yield the desired rate.

# F  Natural AC variant and its convergence

In this section, we propose a natural Actor-Critic variant of Algorithm 1, where the approach of calculating the natural policy graident under the decentralized setting is mainly inspired by [6]. We show that the gradient norm square of such an algorithm will convergence with the optimal sample complexity of $\widetilde{\mathcal{O}}(\varepsilon^{-3})$. Moreover, the algorithm will converge to the *global optimum* with the sample complexity of $\widetilde{\mathcal{O}}(\varepsilon^{-4})$. In the rest of this section, we first explain the update of the algorithm, and then prove its convergence.

## F.1  Decentralized natural Actor-Critic

The natural policy gradient (NPG) algorithm [12] can be viewed as a preconditioned policy gradient algorithm, which updates as follow:

$$\theta_{k+1} = \theta_k - \alpha_k F(\theta_k)^{-1}\nabla J(\theta_k), \tag{131}$$

where $F(\theta) := \mathbb{E}_{s\sim d_{\pi_\theta}, a\sim\pi_\theta}\left[\psi_\theta(s,a)\psi_\theta(s,a)^T\right]$ is the Fisher information matrix (FIM).[3] The natural Actor-Critic (NAC) uses the critic variable to estimate the gradient. The main challenge for implementing NAC lies in the estimation of the inverse matrix-vector product $F(\theta_k)^{-1}\nabla J(\theta_k)$,

---

[3]Throughout the discussion, we assume that FIM is invertible and thus positive-definite.

---

**Algorithm 3:** Decentralized single-timescale NAC

---

1: **Initialize:** Actor parameter $\theta_0$, critic parameter $\omega_0$, reward estimator parameter $\lambda_0$, initial state $s_0$, natural policy gradient estimation $h_{k,0}$.
2: **for** $k = 0, \cdots, K - 1$ **do**
3:    **Option 1: i.i.d. sampling:**
4:    $s_k \sim \mu_{\theta_k}(\cdot), a_k \sim \pi_{\theta_k}(\cdot|s_k), s_{k+1} \sim \mathcal{P}(\cdot|s_k, a_k)$.
5:    **Option 2: Markovian sampling:**
6:    $a_k \sim \pi_{\theta_k}(\cdot|s_k), s_{k+1} \sim \mathcal{P}(\cdot|s_k, a_k)$.
7:
8:    **Periodical consensus:** Compute $\tilde{\omega}_k^i$ and $\tilde{\lambda}_k^i$ by (4) and (7).
9:
10:    **for** $i = 0, \cdots, N$ **in parallel do**
11:      **Reward estimator update:** Update $\lambda_{k+1}^i$ by (8).
12:      **Critic update:** Update $\omega_{k+1}^i$ by (5).
13:      **Actor update:**
14:        Collect $N_a$ transition samples based on Markovian/i.i.d sampling.
15:        **for** $k' = 1, \cdots, K_a$ **do**
16:          Estimate $\bar{z}_{k',n}, \forall n \in [N_a]$ using (133).
17:          Update $h_{k,k'+1}$ by (135).
18:        **end for**
19:        Update $\theta_{k+1}^i$ by (136).
20:    **end for**
21: **end for**

---

especially under the decentralized setting. The work [6] proposes to solve the following strongly convex problem in order to estimate the product in a decentralized way

$$h(\theta_k) = \arg\min_h f_{\theta_k}(h) := \frac{1}{2}h^T F(\theta_k)h - \nabla J(\theta_k)^T h. \tag{132}$$

Such a problem can be solved by using (stochastic) gradient descent, where the gradient is calculated by $F(\theta_k)h - \nabla J(\theta_k)$. For the centralized setting, the gradient w.r.t. each agent can be approximated as $\frac{1}{N_a}\sum_{n=1}^{N_a} \psi_{\theta_k}^i(s_n, a_n^i)\psi_{\theta_k}(s_n, a_n)^T h - g_a^i(\xi_n, \omega_{k+1}, \lambda_{k+1})$. However, when considering the decentralized setting, the term $\bar{z}_n := \psi_{\theta_k}(s_n, a_n)^T h = \sum_{i=1}^N \psi_{\theta_k}^i(s_n, a_n)^T h^i$ is not accessible for each agent. Therefore, to approximate this value, agents compute $z_{n,0}^i := \psi_{\theta_k}^i(s_n, a_n)^T h^i$ locally and then perform the following communication step for $K_z$ steps

$$z_{n,k'+1}^i = \sum_{j=1}^N W^{ij} z_{n,k'}^i, \ \forall n \in [N_a], \ k' = 0, \cdots, K_z - 1. \tag{133}$$

As we will see, $N z_{n,k'}^i$ converges to $\bar{z}_n$ linearly. Thus, the gradient of agent $i$ can be approximated as

$$\widetilde{\nabla} f_{\theta_k}^i(h_{k,k'}) := \frac{N}{N_a}\sum_{n=1}^{N_a} \psi_{\theta_k}^i(s_n, a_n^i) z_{n,K_z}^i - g_a^i(\xi_k, \omega_{k+1}, \lambda_{k+1}). \tag{134}$$

Then, each agent $i$ performs the following update for $K_a$ steps to estimate the natural policy gradient direction as

$$h_{k,k'+1}^i = \Pi_{C_h}(h_{k,k'}^i - \varrho \widetilde{\nabla} f_{\theta_k}^i(h_{k,k'})), \tag{135}$$

where $\varrho$ is a positive constant step size. Since the norm of optimal direction is bounded by $C_h := \lambda_{\max}(F(\theta)^{-1})C_\theta$, we project the vector into a ball of norm $C_h$ for each update. Finally, we perform the approximate natural policy gradient step as

$$\theta_{k+1}^i = \theta_k^i - \alpha_k h_{k,K_a}^i. \tag{136}$$

## F.2   Convergence of natural Actor-Critic

In this section, we establish the sample complexity of Algorithm 3. We first introduce an additional assumption.

**Assumption 6.** *(invertible FIM) There exists a positive constant $\lambda_F$ such that for all policy $\theta$,* $\lambda_{\min}(F(\theta)) \geq \lambda_F$.

Assumption 6 ensures that $F(\theta)$ is positive definite so that the problem (132) is strongly convex. Such an assumption is commonly adopted; see [6, 36, 17].

We now show the sample complexity of the Algroithm 3 in terms of gradient norm square and the global optimal gap. We consider the i.i.d. sampling to simplify the proof. We remark that the proof for Markovian sampling follows the similar analysis, with additional $\mathcal{O}(\log(\varepsilon^{-1}))$ error terms caused by Markov chain mixing.

**Theorem 4.** *Suppose Assumptions 1-6 hold. Consider the update of Algorithm 3 under i.i.d. sampling. Let $\alpha_k = \frac{\bar{\alpha}}{\sqrt{K}}$ for some positive constant $\bar{\alpha}$, $\beta_k = \frac{C_9}{2\lambda_\phi}\alpha_k$, $\varrho \leq \frac{1}{2C_\psi^2}$, $N_a = \mathcal{O}(\sqrt{K})$, $K_a = \mathcal{O}(\log(K^{1/2})), K_c = \mathcal{O}(\log(K^{1/4}))$. Then, the following hold*

$$\frac{1}{K}\sum_{k=1}^{K}\sum_{i=1}^{N}\mathbb{E}\left[\|\nabla_{\theta^i}F(\theta_k)\|^2\right] \leq \mathcal{O}\left(\frac{1}{\sqrt{K}}\right) + \mathcal{O}(\varepsilon_{app} + \varepsilon_{sp}) \tag{137}$$

$$\frac{1}{K}\sum_{k=0}^{K}J(\theta^*) - J(\theta_k) \leq \mathcal{O}\left(\frac{1}{K^{1/4}}\right) + \mathcal{O}(\varepsilon_{app} + \varepsilon_{sp} + \varepsilon_{actor}). \tag{138}$$

Based on Theorem 4, Algorithm 3 needs $K = \mathcal{O}(\varepsilon^{-2})$ iterations to achieve $\varepsilon$-error for gradient norm square, and thus attains sample complexity of $KN_aK_a = \widetilde{\mathcal{O}}(\varepsilon^{-3})$, which matches the best existing sample complexity of NAC [35, 6]. In terms of the global optimality gap, the algorithm requires $K = \mathcal{O}(\varepsilon^{-4})$ iterations to achieve $\varepsilon$-error, and thus has $KN_aK_a = \widetilde{\mathcal{O}}(\varepsilon^{-6})$ sample complexity. Such a sample complexity is much worse than the best existing sample complexity of $\widetilde{\mathcal{O}}(\varepsilon^{-3})$ [35, 6].

We now explain the intuition of the gap for the sample complexity. Mimicking the analysis of [6] allows to establish the following inequality

$$\frac{1}{K}\sum_{k=0}^{K}J\left(\theta^*\right) - \mathbb{E}[J(\theta_k)] \leq \mathcal{O}\left(\frac{1}{K}\sum_{k=1}^{K}\sum_{i=1}^{N}\mathbb{E}[\|\nabla_{\theta^i}J(\theta_k)\|^2]\right)$$
$$+ \mathcal{O}\left(\frac{1}{K}\sum_{k=1}^{K}\sum_{i=1}^{N}\|\omega_k^i - \omega^*(\theta_k)\|\right) + \mathcal{O}\left(\frac{1}{K\alpha_k}\right).$$

While our analysis can obtain the iteration complexity of $\mathcal{O}(\frac{1}{\sqrt{K}})$ for $\|\nabla J(\theta_k)\|^2$, we can only achieve $\mathcal{O}(\frac{1}{K^{1/4}})$ iteration complexity for critic's error $\|\omega_k - \omega^*(\theta_k)\|$. This is because our algorithm uses single-timescale update, where the critic's error inevitably converges slower than that of double-loop based algorithms which have $\mathcal{O}(\frac{1}{\sqrt{K}})$ complexity for the critic's error at each iteration. Therefore, the sample complexity in terms of global optimality gap of our single-timescale NAC is dominated by this critic's error term, resulting in the final complexity of $\widetilde{\mathcal{O}}(\varepsilon^{-6})$.

We remark that this sample complexity result is based on a straightforward application of the analysis of [6], which is designed for double-loop algorithm. Therefore, such a proof technique may not be the tightest one for our single-timescale NAC (intuitively, the result is not tight). We leave the research on the improvement of such highly suboptimal results of single-timescale NAC as a future work.

### F.3 Proof of Theorem 4

By Lemma 4, we have

$$\mathbb{E}[J(\theta_{k+1})] - J(\theta_k) \geq \sum_{i=1}^{N}\mathbb{E}\langle\nabla_{\theta^i}J(\theta_k), \theta_{k+1}^i - \theta_k^i\rangle - \frac{L}{2}\sum_{i=1}^{N}\|\theta_{k+1}^i - \theta_k^i\|^2$$
$$\overset{(i)}{\geq} \sum_{i=1}^{N}\alpha_k\mathbb{E}\langle\nabla_{\theta^i}J(\theta_k), h_k^i\rangle - \frac{L}{2}NC_h^2\alpha_k^2$$

$$
= \sum_{i=1}^{N} [\alpha_k \mathbb{E} \langle \nabla_{\theta^i} J(\theta_k), F(\theta_k)^{-1} g_a^i(\xi_k, \omega_{k+1}^i, \lambda_{k+1}^i) \rangle
$$
$$
+ \alpha_k \mathbb{E} \langle \nabla_{\theta^i} J(\theta_k), h_k^i - F(\theta_k)^{-1} g_a^i(\xi_k, \omega_{k+1}^i, \lambda_{k+1}^i) \rangle] - \frac{L}{2} N C_h^2 \alpha_k^2
$$
$$
\stackrel{(ii)}{=} \sum_{i=1}^{N} [\alpha_k \mathbb{E} \langle F(\theta_k)^{-1/2} \nabla_{\theta^i} J(\theta_k), F(\theta_k)^{-1/2} g_a^i(\xi_k, \omega_{k+1}^i, \lambda_{k+1}^i) \rangle
$$
$$
+ \alpha_k \mathbb{E} \langle \nabla_{\theta^i} J(\theta_k), h_k^i - F(\theta_k)^{-1} g_a^i(\xi_k, \omega_{k+1}^i, \lambda_{k+1}^i) \rangle] - \frac{L}{2} N C_h^2 \alpha_k^2
$$
$$
= \sum_{i=1}^{N} [\frac{\alpha_k}{2} \| F(\theta_k)^{-1/2} \nabla_{\theta^i} J(\theta_k) \|^2 + \frac{\alpha_k}{2} \| F(\theta_k)^{-1/2} \mathbb{E}[g_a^i(\xi_k, \omega_{k+1}^i, \lambda_{k+1}^i)] \|^2
$$
$$
- \frac{\alpha_k}{2} \| F(\theta_k)^{-1/2} \nabla_{\theta^i} J(\theta_k) - F(\theta_k)^{-1/2} \mathbb{E}[g_a^i(\xi_k, \omega_{k+1}^i, \lambda_{k+1}^i)] \|^2
$$
$$
+ \alpha_k \mathbb{E} \langle \nabla_{\theta^i} J(\theta_k), h_k^i - F(\theta_k)^{-1} g_a^i(\xi_k, \omega_{k+1}^i, \lambda_{k+1}^i) \rangle] - \frac{L}{2} N C_\theta^2 \alpha_k^2
$$
$$
\stackrel{(iii)}{\geq} \sum_{i=1}^{N} [\frac{\alpha_k}{4} C_\psi^{-2} \| \nabla_{\theta^i} J(\theta_k) \|^2 + \frac{\alpha_k}{2} \lambda_F \| F(\theta_k)^{-1} \mathbb{E}[g_a^i(\xi_k, \omega_{k+1}^i, \lambda_{k+1}^i)] \|^2
$$
$$
- \frac{\alpha_k}{2} \lambda_F^{-1} \underbrace{\| \nabla_{\theta^i} J(\theta_k) - \mathbb{E}[g_a^i(\xi_k, \omega_{k+1}^i, \lambda_{k+1}^i)] \|^2}_{I_1}
$$
$$
- \alpha_k C_\psi^2 \underbrace{\| \mathbb{E}[h_k^i] - F(\theta_k)^{-1} \mathbb{E}[g_a^i(\xi_k, \omega_{k+1}^i, \lambda_{k+1}^i)] \|^2}_{I_2}] - \frac{L}{2} N C_\theta^2 \alpha_k^2,
$$
$$
\tag{139}
$$

where $(i)$ is due to $\| \theta_{k+1}^i - \theta_k^i \| \leq C_h := \lambda_F C_\theta$. Note that we use $h_k^i$ to represent $h_{k, K_a}^i$ for simplifying the notation. $(ii)$ uses decomposition of positive definite (PD) matrix. Specifically, let $A$ be PD matrix, then by eigenvalue decomposition, $A = V \Lambda V^T$ for some orthonormal matrix $V$. Define $A^{-1/2} := V \Lambda^{1/2} V^T$, then $\langle x, Ay \rangle = \langle A^{1/2} x, A^{1/2} y \rangle$ for any $x$ and $y$. $(iii)$ uses $\lambda_F \leq \lambda(F(\theta)) \leq C_\psi^2, \ \forall \theta$.

$I_1$ represents the error of gradient bias, which we have bounded when analyzing the error of AC. By (96), we have

$$
I_1 \leq 16(\varepsilon_{sp} + C_\psi^2 \varepsilon_{app}) + 16 C_\psi^2 \| \omega^*(\theta_k) - \omega_{k+1}^i \|^2 + 8 C_\psi^2 \| \lambda^*(\theta_k) - \lambda_{k+1}^i \|^2. \tag{140}
$$

To bound $I_2$, we need to bound the error of $h_{k, k'}$. We start with the gradient bias when estimating $h_{k, k'}$. Define $\overline{\nabla} f_{k, k'}(h_{k, k'}) := \nabla F(\theta_k) h_{k, k'} - \mathbb{E}[g_a(\xi_k, \omega_{k+1}^i, \lambda_{k+1}^i)]$, then it is easy to see that $\overline{\nabla} f_{k, k'}(h_{k, k'})$ is the unbiased gradient of the following problem

$$
\frac{1}{2} h_{k, k'}^T \nabla F(\theta_k) h_{k, k'} - \mathbb{E}[g_a(\xi_k, \omega_{k+1}^i, \lambda_{k+1}^i)]^T h_{k, k'}.
$$

Define the following notation for the ease of expression

$$
\widehat{\nabla} f_{k, k'}^i(h_{k, k'}) := \frac{1}{N_a} \sum_{n=1}^{N_a} \psi_{\theta_k^i}(s_n, a_n^i) \psi_{\theta_k}(s_n, a_n)^T h_{k, k'} - g_a^i(\xi_{k, k'}, \omega_{k+1}^i, \lambda_{k+1}^i)
$$
$$
\widehat{\nabla} f_{k, k'}(h_{k, k'}) := [\widehat{\nabla} f_{k, k'}^1(h_{k, k'}), \cdots, \widehat{\nabla} f_{k, k'}^N(h_{k, k'})]
$$
$$
\widetilde{\nabla} f_{k, k'}^i(h_{k, k'}) := \frac{N}{N_a} \sum_{n=1}^{N_a} \psi_{\theta_k^i}(s_n, a_n^i) z_{n, K_z}^i - g_a^i(\xi_{k, k'}, \omega_{k+1}^i, \lambda_{k+1}^i)
$$
$$
\widetilde{\nabla} f_{k, k'}(h_{k, k'}) := [\widetilde{\nabla} f_{k, k'}^1(h_{k, k'}), \cdots, \widetilde{\nabla} f_{k, k'}^N(h_{k, k'})].
$$

We now analyze the error at outer-loop iteration $k$. For notational simplicity, we omit the subscript $k$ for the prementioned notations, e.g. we use $\widehat{\nabla} f_{k'}^i(h_{k'})$, $\widehat{\nabla} f_{k'}(h_{k'})$, $\widetilde{\nabla} f_{k'}^i(h_{k'})$, $\widetilde{\nabla} f_{k'}(h_{k'})$ to represent the above notations, respectively.

$$\|\overline{\nabla} f_{k'}(h_{k'}) - \widetilde{\nabla} f_{k'}(h_{k'})\|^2 \le 2 \underbrace{\|\overline{\nabla} f_{k'}(h_{k'}) - \widehat{\nabla} f_{k'}(h_{k'})\|^2}_{I_3} + 2 \underbrace{\|\widehat{\nabla} f_{k'}(h_{k'}) - \widetilde{\nabla} f_{k'}(h_{k'})\|^2}_{I_4}.$$

$I_3$ can be bounded as

$$\begin{aligned} I_3 &= \|\sum_{n=1}^{N_a}(\frac{1}{N_a}\psi_\theta(s_n,a_n)\psi_\theta(s_n,a_n)^T - F(\theta))h_{k'}\|^2 \\ &\le \|\sum_{n=1}^{N_a}(\frac{1}{N_a}\psi_\theta(s_n,a_n)\psi_\theta(s_n,a_n)^T - F(\theta))\|^2 C_h^2 \\ &\le \frac{1}{N_a}C_\psi^4 C_h^2. \end{aligned} \tag{141}$$

$I_4$ can be bounded as

$$\begin{aligned} I_4 &= \sum_{i=1}^{N}\left\|\psi_{\theta^i}(s_n,a_n^i)\left(\frac{1}{N_a}\sum_{n=1}^{N_a}N z_{n,K_z}^i - \psi_\theta(s_n,a_n)^T h_{k'}\right)\right\|^2 \\ &\le \frac{1}{N_a}N C_\psi^2 \sum_{i=1}^{N}\sum_{n=1}^{N_a}\|z_{n,K_z}^i - \bar{z}_{n,K_z}\|^2 \\ &= \frac{N C_\psi^2}{N_a}\sum_{n=}^{N_a}\|QW^{K_z}z_{n,0}\|^2 \\ &\le \frac{N C_\psi^2}{N_a}\sum_{n=1}^{N_a}\nu^{K_z}\|z_{n,0}\|^2 \le N C_\psi^4 C_h^2 \nu^{K_z}. \end{aligned} \tag{142}$$

Let $K_z = \min\{c \in \mathbb{N}^+ | \nu^c \le \frac{4}{N_a N}\}$, then $K_z = \mathcal{O}(\log \frac{1}{N_a})$. Combine (141) and (142) gives us

$$\|\overline{\nabla} f_{k'}(h_{k'}) - \widetilde{\nabla} f_{k'}(h_{k'})\|^2 \le \frac{4 C_\psi^4 C_h^2}{N_a}.$$

We now analyze the error of $h_{k,k'}$. Note that we omit the subscript $k$ here for simplifying notation.
Define

$$h^* = \arg\min_h \bar{f}_\theta(h) := h^T F(\theta)h := -\mathbb{E}_{\xi \sim \mu_\theta}[g_a(\xi,\omega,\lambda)]^T h. \tag{143}$$

It is easy to see that the function on the RHS is strongly convex, since $F(\theta)$ is positive definite w.r.t. $h$. We bound the optimal gap by

$$\begin{aligned} \mathbb{E}\|h_{k'+1} - h^*\|^2 &= \mathbb{E}\|h_{k'} - \varrho\widetilde{\nabla} f_{k'}(h_{k'}) - h^*\|^2 \\ &= \mathbb{E}\|h_{k'} - h^*\|^2 - 2\varrho\mathbb{E}\langle h_{k'} - h^*, \widetilde{\nabla} f_{k'}(h_{k'})\rangle + \varrho^2\|\widetilde{\nabla} f_{k'}(h_{k'})\|^2 \\ &\le \mathbb{E}\|h_{k'} - h^*\|^2 - 2\varrho\mathbb{E}\langle h_{k'} - h^*, \overline{\nabla} f_{k'}(h_{k'})\rangle + 2\varrho\mathbb{E}\langle h_{k'} - h^*, \overline{\nabla} f_{k'}(h_{k'}) - \widetilde{\nabla} f_{k'}(h_{k'})\rangle \\ &\quad + 2\varrho^2\|\overline{\nabla} f_{k'}(h_{k'})\|^2 + 2\varrho^2\|\widetilde{\nabla} f_{k'}(h_{k'}) - \overline{\nabla} f_{k'}(h_{k'})\|^2 \\ &\overset{(i)}{\le} (1 - \varrho\lambda_F)\mathbb{E}\|h_{k'} - h^*\|^2 - 2\varrho(f_{k'}(h_{k'}) - \overline{f}^*) + 2\varrho\mathbb{E}\langle h_{k'} - h^*, \overline{\nabla} f_{k'}(h_{k'}) - \widetilde{\nabla} f_{k'}(h_{k'})\rangle \\ &\quad + 2\varrho^2\|\overline{\nabla} f_{k'}(h_{k'})\|^2 + 2\varrho^2\|\widetilde{\nabla} f_{k'}(h_{k'}) - \overline{\nabla} f_{k'}(h_{k'})\|^2 \\ &\overset{(ii)}{\le} (1 - \varrho\lambda_F)\mathbb{E}\|h_{k'} - h^*\|^2 - 2\varrho(1 - 2\varrho C_\psi^2)(f_{k'}(h_{k'}) - \overline{f}^*) \\ &\quad + 2\varrho\mathbb{E}\langle h_{k'} - h^*, \overline{\nabla} f_{k'}(h_{k'}) - \widetilde{\nabla} f_{k'}(h_{k'})\rangle + 2\varrho^2\|\widetilde{\nabla} f_{k'}(h_{k'}) - \overline{\nabla} f_{k'}(h_{k'})\|^2 \\ &\overset{(iii)}{\le} (1 - \varrho\lambda_F)\mathbb{E}\|h_{k'} - h^*\|^2 + 2\varrho\mathbb{E}\langle h_{k'} - h^*, \overline{\nabla} f_{k'}(h_{k'}) - \widetilde{\nabla} f_{k'}(h_{k'})\rangle \\ &\quad + 2\varrho^2\|\widetilde{\nabla} f_{k'}(h_{k'}) - \overline{\nabla} f_{k'}(h_{k'})\|^2 \\ &\overset{(iiii)}{\le} (1 - \frac{\varrho\lambda_F}{2})\mathbb{E}\|h_{k'} - h^*\|^2 + (\frac{2\varrho}{\lambda_F} + 2\varrho^2)\|\widetilde{\nabla} f_{k'}(h_{k'}) - \overline{\nabla} f_{k'}(h_{k'})\|^2, \end{aligned}$$

where $\overline{f}^*$ is the optimal value of $\overline{f}(h)$ defined in (143), and the inequality follows the property of $\lambda_F$-strongly convex function: $\overline{f}(h_2) \geq \overline{f}(h_1) + \langle \nabla \overline{f}(h_1), h_2 - h_1 \rangle + \frac{\lambda_F}{2}\|h_1 - h_2\|^2$, $\forall h_1, h_2$. $(ii)$ uses the PL condition implied by $\lambda_F$-strong convexity: $\overline{f}(h^*) - \overline{f}(h) \leq -\frac{1}{2\lambda_F}\|\nabla \overline{f}(h)\|^2$, $\forall h$. $(iii)$ is due to step size rule that $\varrho \leq \frac{1}{2C_\psi^2}$. $(iiii)$ applies Young's inequality.

Use the above induction, we have

$$\mathbb{E}\|h_{K_a} - h^*\|^2 \leq (1 - \frac{\varrho\lambda_F}{2})^{K_a}\|h_0 - h^*\|^2 + \sum_{t=0}^{K_a}(1 - \frac{\varrho\lambda_F}{2})^t(\frac{2\varrho}{\lambda_F} + 2\varrho^2)\|\overline{\nabla}f_{K_a-t}(h_{K_a}) - \widetilde{\nabla}f_{K_a}(h_{K_a})\|^2$$

$$\leq 4C_h^2(1 - \frac{\varrho\lambda_F}{2})^{K_a} + (\frac{4\varrho}{\varrho\lambda_F^2} + \frac{4\varrho}{\lambda_F})C_\psi^4 C_h^2 \frac{4}{N_a}.$$

Let $K_a = \min\{c \in \mathbb{N}^+ | 4C_h^2(1 - \frac{\varrho\lambda_F}{2})^c = (\frac{4\varrho}{\varrho\lambda_F^2} + \frac{4\varrho}{\lambda_F})C_\psi^4 C_h^2 \frac{1}{N_a}\}$, then $K_a = \mathcal{O}(\log(\frac{1}{N_a}))$. Define $C_{18} := (\frac{16\varrho}{\varrho\lambda_F^2} + \frac{16\varrho}{\lambda_F})C_\psi^4 C_h^2$, we have

$$I_2 = \mathbb{E}\|h_{K_a} - h^*\|^2 \leq \frac{2C_{18}}{N_a}. \tag{144}$$

Plug (140) and (144) back to (139), we have

$$\mathbb{E}[J(\theta_{k+1})] - J(\theta_k) \geq \sum_{i=1}^{N}[\frac{\alpha_k}{4}C_\psi^{-2}\|\nabla_{\theta^i}J(\theta_k)\|^2 + \frac{\alpha_k}{2}\lambda_F\|F(\theta_k)^{-1}\mathbb{E}[g_a^i(\xi_k, \omega_{k+1}^i, \lambda_{k+1}^i)]\|^2 + \alpha_k C_\psi^2 \frac{2C_{18}}{N_a}$$

$$+ 8\lambda_F^{-1}(\varepsilon_{sp} + C_\psi^2\varepsilon_{app}) + 8\lambda_F^{-1}C_\psi^2\|\omega^*(\theta_k) - \omega_{k+1}^i\|^2 + 4\lambda_F^{-1}C_\psi^2\|\lambda^*(\theta_k) - \lambda_{k+1}^i\|^2]$$

Consider the Lyapunov function

$$\mathbb{V}^k = -J(\theta_k) + \lambda_F^{-1}(\|\omega_k - \omega^*(\theta_k)\|^2 + \|\lambda_k - \lambda^*(\theta_k)\|^2).$$

The difference of the Lyapunov function is

$$\mathbb{E}[\mathbb{V}^{k+1}] - \mathbb{E}[\mathbb{V}^k] = \mathbb{E}[J(\theta_k)] - \mathbb{E}[J(\theta_{k+1})] + \lambda_F^{-1}(\mathbb{E}\|\omega_{k+1} - \omega^*(\theta_{k+1})\|^2 - \mathbb{E}\|\omega_k - \omega^*(\theta_k)\|^2$$

$$+ \mathbb{E}\|\lambda_{k+1} - \lambda^*(\theta_{k+1})\|^2 - \mathbb{E}\|\lambda_k - \lambda^*(\theta_k)\|^2)$$

$$\leq \sum_{i=1}^{N}\left[\frac{\alpha_k}{4}C_\psi^{-2}\mathbb{E}\|\nabla_{\theta^i}J(\theta_k)\|^2 + \frac{\alpha_k}{2}\lambda_F\|F(\theta_k)^{-1}\mathbb{E}[g_a^i(\xi_k, \omega_{k+1}^i, \lambda_{k+1}^i)]\|^2 + \alpha_k C_\psi^2\frac{2C_{18}}{N_a}\right]$$

$$+ \lambda_F^{-1}\underbrace{\left[\sum_{i=1}^{N}8C_\psi^2\alpha_k\mathbb{E}\|\omega^*(\theta_k) - \omega_{k+1}^i\|^2 + \mathbb{E}\|\bar{\omega}_{k+1} - \omega^*(\theta_{k+1})\|^2 - \mathbb{E}\|\bar{\omega}_k - \omega^*(\theta_k)\|^2\right]}_{I_5}$$

$$+ \lambda_F^{-1}\underbrace{\left[\sum_{i=1}^{N}4C_\psi^2\alpha_k\mathbb{E}\|\lambda^*(\theta_k) - \lambda_{k+1}^i\|^2 + \mathbb{E}\|\bar{\lambda}_{k+1} - \lambda^*(\theta_{k+1})\|^2 - \mathbb{E}\|\bar{\lambda}_k - \lambda^*(\theta_k)\|^2\right]}_{I_6}$$

$$+ 8N\lambda_F^{-1}(\varepsilon_{sp} + C_\psi^2\varepsilon_{app}). \tag{145}$$

By following the similar procedures through (98) to (106), we can bound $I_5$ and $I_6$ as

$$I_5 \leq (1 + C_{19}\alpha_k)C_\delta^2\beta_k^2 + \frac{\alpha_k}{4}\lambda_F^{-1}\sum_{i=1}^{N}\mathbb{E}\|F(\theta_k)^{-1}g_a^i(\xi_k, \omega_{k+1}^i, \lambda_{k+1}^i)\|^2 + \alpha_k M_{k_1} + C_{20}\alpha_k^2$$

$$\tag{146}$$

$$I_6 \leq (1 + C_{21}\alpha_k)C_\lambda^2\eta_k^2 + \frac{\alpha_k}{4}\lambda_F^{-1}\sum_{i=1}^{N}\mathbb{E}\|F(\theta_k)^{-1}g_a^i(\xi_k, \omega_{k+1}^i, \lambda_{k+1}^i)\|^2 + \alpha_k M_{k_2} + C_{22}\alpha_k^2,$$

$$\tag{147}$$

where $C_{19}, C_{20}, C_{21}, C_{22}$ are some positive constants. Plug (146) and (147) back to (145), we have

$$\mathbb{E}[\mathbb{V}^{k+1}] - \mathbb{E}[\mathbb{V}^k] \leq \sum_{i=1}^{N} [\frac{\alpha_k}{4} C_\psi^{-2} \mathbb{E}\|\nabla_{\theta^i} J(\theta_k)\|^2 + \alpha_k C_\psi^2 \frac{2C_{18}}{N_a} + \mathcal{O}(\alpha_k^2 + \beta_k^2 + \eta_k^2)$$
$$+ (M_{k_1} + M_{k_2})\alpha_k + \mathcal{O}(\varepsilon_{sp} + \varepsilon_{app})\alpha_k]. \tag{148}$$

By telescoping (148), we can get

$$\frac{1}{K} \sum_{k=0}^{K} \sum_{i=1}^{N} \mathbb{E}\|\nabla_{\theta^i} J(\theta_k)\|^2 \leq \frac{4C_\psi^2 \mathbb{V}_0}{K\alpha_k} + \mathcal{O}(\varepsilon_{sp} + \varepsilon_{app}) + \frac{8C_\psi^2 C_{18}}{N_a} + \mathcal{O}(\alpha_k + \frac{\beta_k^2}{\alpha_k} + \frac{\eta_k^2}{\alpha_k})$$
$$+ 4C_\psi^2 (M_{k_1} + M_{k_2})$$

By (108), $M_{k_1} + M_{k_2} = \mathcal{O}(\frac{1}{\sqrt{K}})$ when $K_c \leq \mathcal{O}(K^{1/4})$. Therefore, let $C, \bar{\alpha}$ be some positive constants. Set $N_a = C\sqrt{K}$, $\alpha_k = \frac{\bar{\alpha}}{\sqrt{K}}$, $\beta_k = \frac{C_9}{2\lambda_\phi} \alpha_k$, $\eta_k = \frac{C_{10}}{2\lambda_\varphi} \alpha_k$, we obtain the desired result of (137).

We now prove (138). Let $\mathbb{E}_{\theta^*}$ denote the expectation over $s \sim d_{\pi_{\theta^*}}, a \sim \pi_{\theta^*}(\cdot|s)$. We begin with the descent of policy gap as

$$\mathbb{E}_{\theta^*}[\log \pi_{\theta_{k+1}}(a|s) - \log \pi_{\theta_k}(a|s)]$$
$$\geq \alpha_k \mathbb{E}_{\theta^*}[\psi_{\theta_k}(s,a)^T h_k] - \frac{L_\psi \alpha_k^2}{2} C_h^2$$
$$\geq \alpha_k \mathbb{E}_{\theta^*}[\psi_{\theta_k}(s,a)^T(h_k - h^*(\theta_k))] + \alpha_k \mathbb{E}_{\theta^*}[\psi_{\theta_k}(s,a)^T h^*(\theta_k) - A_{\theta_k}(s,a)]$$
$$+ \alpha_k \mathbb{E}_{\theta^*}[A_{\theta_k}(s,a)] - \frac{L_\psi \alpha_k^2}{2} C_h^2$$
$$\geq -\alpha_k C_\psi \|h_k - h^*(\theta_k)\| - \alpha_k \sqrt{\varepsilon_{actor}} + \alpha_k (J(\theta^*) - J(\theta_k)) - \frac{L_\psi \alpha_k^2}{2} C_h^2.$$

By telescoping the above inequality and rearranging terms, we have

$$\frac{1}{K} \sum_{k=1}^{K} (J(\theta^*) - J(\theta_k)) \leq \frac{1}{K\alpha_k} \mathbb{E}_{\theta^*}[\log \pi_K(a|s) - \log \pi_0(a|s)] + \sqrt{\varepsilon_{actor}}$$
$$+ \frac{1}{K} \sum_{k=1}^{K} C_\psi \|h_k - h^*(\theta_k)\| + \frac{1}{K} \sum_{k=1}^{K} \frac{L_\psi \alpha_k}{2}.$$

The term $\|h_k - h^*(\theta_k)\| \leq \|h_k - F(\theta_k)^{-1} \mathbb{E}[g_a(\xi_k, \omega_{k+1}, \lambda_{k+1})]\| + \|\mathbb{E}[g_a(\xi_k, \omega_{k+1}, \lambda_{k+1}) - F^{-1} \nabla J(\theta_k)\|$. Since by the (144) and (96), these two terms are of order $\mathcal{O}(\frac{1}{N_a^{1/2}})$ and $\mathcal{O}(\|\omega_k - \omega_{k+1}\| + \varepsilon_{app})$, respectively, we conclude that $\|h_k - h^*(\theta_k)\|$ is of order $\mathcal{O}(\|\omega_k - \omega^*(\theta_k)\| + \varepsilon_{app})$. By following the step size rule as suggested by Theorem 4, we obtain the desired result.

## G   Overview of communication complexity

The Table 1 compares related works in terms of sample complexity and communication complexity.

| Setting | Paper | Update | Sampling | Sample complexity | Communication complexity |
|---|---|---|---|---|---|
| Single-agent AC | [32] | Two-timescale | Markovian | $\widetilde{\mathcal{O}}(\varepsilon^{-\frac{5}{2}})$ | - |
| | [35] | Double-loop | Markovian | $\widetilde{\mathcal{O}}(\varepsilon^{-2})$ | - |
| Decentralized AC | [42] | Two-timescale | Markovian | Asymptotic | - |
| | [38] | Two-timescale | i.i.d. | $\mathcal{O}(\varepsilon^{-\frac{5}{2}})$ | $\mathcal{O}(\varepsilon^{-\frac{5}{2}})$ |
| | [6] | Double-loop | Markovian | $\widetilde{\mathcal{O}}(\varepsilon^{-2})$ | $\widetilde{\mathcal{O}}(\varepsilon^{-1})$ |
| | [11] | Double-loop | Markovian | $\widetilde{\mathcal{O}}(\varepsilon^{-2})$ | $\widetilde{\mathcal{O}}(\varepsilon^{-1})$ |
| | **This work** | **Single-timescale** | Markovian | $\widetilde{\mathcal{O}}(\varepsilon^{-2})$ | $\widetilde{\mathcal{O}}(\varepsilon^{-\frac{3}{2}})$ |

Table 1: Comparison of some existing sample complexity results. The symbol $\tilde{\mathcal{O}}(\cdot)$ hides the logarithmic terms.

## H   Policy gradient theorem

The following derivation establishes the policy gradient update of our algorithm.

$$
\nabla \mathbb{E}_{s_0 \sim \mu_0}[V_{\pi^\theta}(s_0)] = \mathbb{E}_{s_0 \sim \mu_0}\left[ \nabla \sum_{a_0} \pi_\theta(a_0|s_0) Q_{\pi_\theta}(s_0, a_0) \right]
$$

$$
= \mathbb{E}_{s_0 \sim \mu_0}\left[ \underbrace{\sum_{a_0} \nabla \pi_\theta(a_0|s_0) Q_{\pi_\theta}(s_0, a_0)}_{\text{1st term on RHS of (7)}} + \underbrace{\sum_{a_0} \pi_\theta(a_0|s_0) \nabla Q_{\pi_\theta}(s_0, a_0)}_{\text{2nd term on RHS of (7)}} \right]
$$

$$
= \mathbb{E}_{s_0 \sim \mu_0}\left[ \sum_{a_0} \pi_\theta(a_0|s_0) \nabla \log \pi_\theta(a_0|s_0) Q_{\pi_\theta}(s_0, a_0) \right]
$$

$$
+ \mathbb{E}_{s_0 \sim \mu_0}\left[ \sum_{a_0} \pi_\theta(a_0|s_0) \nabla \left( r(s_0, a_0) + \gamma \sum_{s_1} P(s_1|s_0, a_0) V_{\pi_\theta}(s_1) \right) \right]
$$

$$
= \mathbb{E}_{s_0 \sim \mu_0}\left[ \sum_{a_0} \pi_\theta(a_0|s_0) \nabla \log \pi_\theta(a_0|s_0) Q_{\pi_\theta}(s_0, a_0) + \gamma \sum_{a_0, s_1} \pi_\theta(a_0|s_0) \nabla V_{\pi_\theta}(s_1) \right]
$$

$$
= \mathbb{E}_\tau\left[ Q_{\pi_\theta}(s_0, a_0) \nabla \log \pi_\theta(a_0|s_0) \right] + \gamma \mathbb{E}_\tau[\nabla V_{\pi_\theta}(s_1)],
$$

where the (7) in the second inequality refers to equation (7) of [4], and the expectation on $\tau$ is taken over a trajectory: $a_0 \sim \pi_\theta(\cdot|s_0), s_1 \sim P(s_1|s_0, a_0), \cdots$. By expanding the above recursion, we can derive the policy gradient

$$
\nabla \mathbb{E}_{s_0 \sim \mu_0}[V_{\pi_\theta}(s_0)] = \mathbb{E}_\tau\left[ \sum_{k=0}^{\infty} \gamma^k Q_{\pi_\theta}(s_k, a_k) \nabla \log \pi_\theta(a_k, s_k) \right]
$$

$$
= \frac{1}{1-\gamma} \mathbb{E}_{s \sim d_{\pi_\theta}, a \sim \pi_\theta}\left[ Q_{\pi_\theta}(s, a) \nabla \log \pi_\theta(a|s) \right].
$$