# OpenReview forum: "Finite-Time Analysis of Fully Decentralized Single-Timescale Actor Critic "
_NeurIPS.cc/2022/Conference — NeurIPS 2022 Submitted_

### Official Review · Reviewer_hKdD · 2022-07-02

**Rating:** 6
**Confidence:** 4
**Soundness:** 4 excellent
**Presentation:** 4 excellent
**Contribution:** 3 good

**Summary:**

The submission solves corporated MARL via decentralized AC. Specifically, the authors consider single timescale update of AC (i.e., the actor, critic, and reward estimation take stepsizes of the same order). In theory, the authors show that under linear function approximation, AC converges to a stationary point under i.i.d. and Markovian sample schemes, respectively.


**Questions:**

Questions:
 - A discussion on the communication frequency $K_c$ with the final rate of convergence would be interesting. The choice of $K_c$ seems only appear in the introduction (line 71) without much further explanation in the paper.
 - I checked other single timescale methods, especially ALSET [3] and STABLE (Chen, Tianyi, et al.). Compared with the two-timescale algorithm TTSA (Hong, Mingyi, et al.), both ALSET and STABLE seem to put much effort into a careful design of the update formulae, especially on the upper level (eqs (8)(9) in STABLE and eqs (9)&(10) in ALSET). And they claim that this is why they can achieve convergence under single timescale updates (sec 2.2 in (Chen, Tianyi, et al.)). In comparison, Algorithm 1 in the submission seems to be simple yet efficient, since it only consists of a TD(0) in the critic update and an advantage actor in the actor update. It will be great if the authors can explain more on such a comparison, which will definitely bring more intuition to the proof idea.
 - Can the authors also compare with [18] in terms of the proof technique used for single agent AC?

Minor issues:
 - The definition of the function $\omega^*$ seems first appear in the appendix (line 613) but is used before in the paper (line 145).

 - Line 206: $\varphi(s_{|\mathcal S|}, a_{|\mathcal A|})$

 - Algorithm 1 line 4: $s_k \sim \mu_{\theta_k}(\cdot)$

 - Eq (5): should be $\omega_{k+1}^i$ instead of $w_{k+1}^i$

 - The supp material seems to be a little bit different to the paper? E.g., reference [2] is different.


Chen, Tianyi, et al. "A single-timescale method for stochastic bilevel optimization." International Conference on Artificial Intelligence and Statistics. PMLR, 2022.

Hong, Mingyi, et al. "A two-timescale framework for bilevel optimization: Complexity analysis and application to actor-critic." arXiv preprint arXiv:2007.05170 (2020).


**Limitations:**

Yes. The authors can also have a more detailed comparison with the works that prove convergence to a saddle point / global optimality.

**Strengths And Weaknesses:**

Strengths:
 - Single timescale AC is widely adopted in practice but lack of theoretical understanding, even in the scenario of single agent with linear function approximation. Most theoretical works aiming to show convergence of AC end up considering two-timescale or bilevel updates, which ensure the approximate convergence of critic update before the update of actor. The submission not only consider the single-timescale AC, but also establish the convergence of such algorithm in MARL setting, which closes the gap between practice and theory for the first time, to the best of my knowledge.
 - As pointed out by the authors, previous works [8] use LSTD in critic update to study single timescale AC. The naive sample complexity of [8] is $O(\varepsilon^{-4})$. Further, they propose to use off-policy to improve the sample complexity to $O(\varepsilon^{-2})$, which matches the sample complexity in the submission. In comparison, the submission uses TD(0) in the critic update, which is more similar to the practical use.

Weaknesses:
 - The submission only shows convergence to a saddle point, instead of an optimal point. Previous work [5] seems to have a stronger result showing optimality, though adopting a bilevel scheme.

---

> ### Author Response · Authors · 2022-08-02
> **Author's Response**
>
> Thanks so much for the supportive and valuable comments. Let us address the your concerns in a point-by-point manner below. Major changes in the manuscript are highlighted in blue.
>
> **A. Convergence to a saddle point, instead of an optimal point.**
> We have added a new section, i.e., Appendix F in the revised manuscript, to design a single-timescale Natural AC (NAC) algorithm and provide its sample complexity results. Similar to the result in [Chen et al., 2022], this algorithm will converge to a global optimum rather than an arbitrary stationary point. However, we have to admit that the current sample complexity result $\mathcal{O}(\varepsilon^{-6})$ for our singe-timescale NAC is worse than that of [Chen et al., 2022] ($\mathcal{O}(\varepsilon^{-3})$). This worse bound is expected since the derivation is based on a straightforward application of the analysis of [Chen et al., 2022], which is designed for double-loop algorithm. Therefore, such a proof technique may not be the tightest one for single-timescale NAC (intuitively, this result is not tight). We leave the research on the improvement of the suboptimal results of single-timescale NAC as a future work.
>
> **B. Discussion of the communication frequency.**
> The communication frequency $K_c$ affects the convergence rate by influencing the order of consensus error $\sum_{i=1}^{N}||\bar{\omega}_{k} - \omega_k^i||^2$, which is of order $\mathcal{O}(K_c^2\beta_k^2)$. To maintain the optimal sample complexity, the order of the consensus error should be no larger than $\mathcal{O}(\beta_k)$. Therefore, $K_c$ can be any order below $\mathcal{O}(\beta_k^{-\frac{1}{2}})$ without influencing the sample complexity. Since the step size $\beta_k$ is of order $\mathcal{O}(K^{-1/2})=\mathcal{O}(\varepsilon^{-1})$, we conclude that $K_c$ can be any integer lies in $[1, \mathcal{O}(\varepsilon^{-\frac{1}{2}})]$. We have added the requirements on $K_c$ in the statements of all our theorems. In fact, we did put the above discussions on $K_c$ below Lemma 1 in Subsection 4.4.
>
> **C. Comparison of update between bi-level optimization and our algorithm.**
> We now provide more insights about the algorithmic design.
> In fact, the bi-level optimization algorithms TTSA [Hong et al., 2020] and ALSET [Chen, et al., 2021] essentially use the same update formulae with different step size rule. The AC's update direction, i.e. policy gradient direction, is the same as theirs (equation (7) of [Chen, et al., 2021], which we refer as (7) throughout the discussion). The reason why AC's update looks simpler is due to reinforcement learning (RL) problem's special structure. In particular, when calculating policy gradient in RL problem, the the second term on the RHS of (7) forms a nice recursion with the gradient, and helps in establishing a compact form of the gradient. We have appended the policy gradient's derivation in Appendix H for your reference. When we have more space for the main content, we will add this discussion below the update (6).
>
> **D. Proof technique comparison with [Qiu et al., 2019].**
> The work [Qiu et al., 2019]'s analysis is based on assumptions that are essentially different from ours. Specifically, they assume that the error of critic is uniformly bounded by a constant $C_\omega$ through all the iterations. When analyzing the actor's update, they use the constant $C_\omega$ to bound the critic's error, and thereby resulting an error term of order $\mathcal{O}(C_\omega)$ for the convergence point.
> In comparison, our analysis does not assume such a uniform bound, and derive the descent of the critic's error for each iteration. Consequently, the convergence point based on our analysis does not involve such a constant term. We remark that the analysis of the critic's error is usually the challenging part in AC's analysis.
>
> **E. Minor issues.**
> Thank you for your careful review, we have corrected these typos and added the missing definitions; see our latest revised manuscript.
>
> We hope that our response and revisions are satisfactory to the reviewer and that all concerns have been addressed appropriately.
>
> **References**
>
> [Hong et al., 2020]. “A Two-Timescale Framework for Bilevel Optimization: Complexity Analysis and Application to Actor-Critic", arXiv:2007.05170 (2020).
>
> [Chen et al., 2021]. "Tighter Analysis of Alternating Stochastic Gradient Method for
> Stochastic Nested Problems", NeurIPS 2021.
>
> [Qiu et al., 2019]. "On finite-time convergence of actor-critic algorithm", NeurIPS 2019, workshop.

---

> > ### Comment · Reviewer_hKdD · 2022-08-05
> > **Follow-up questions**
> >
> > Thank you very much for your reply.
> >
> > Could you please kindly give a pointer to the constant $C_\omega$ in [Qiu et al., 2019]? I am sorry if I miss something but I cannot find it in their IEEE version: https://ieeexplore.ieee.org/stamp/stamp.jsp?tp=&arnumber=9435807.

---

> > > ### Author Response · Authors · 2022-08-06
> > > **Response on the follow-up questions**
> > >
> > > Thanks so much for your careful reading on our previous response and for your reply. The constant $C_{\omega}$ refers to $\varepsilon_{\mu} + \varepsilon_{\omega}$ in their paper; see Theorem 3 of [Qiu et al., 2021].  Let us also interpret this constant here. As shown in their Theorem 3, the constant $C_{\omega}$ determines the convergence rate of the algorithm. Specifically, in order
> > > to obtain the $\widetilde{\mathcal{O}}\left(\frac{1}{\sqrt{T}}\right)$ iteration complexity,
> > > their result requires $C_{\omega}$ to be of order $\widetilde{\mathcal{O}}\left(\frac{1}{\sqrt{T}}\right)$ at each iteration of their Algorithm 1; see their Remark 3. Towards that end, they employ the double-loop algorithmic framework, in which they invoke TD(0) update (their Algorithm 2) with $K+L=\mathcal{O}(T)$ inner iterations in their Algorithm 1; see their Remark 3 for the number of inner iterations $K$ and $L$. Consequently, theie result requires $T(K+L)=\mathcal{O}(T^2)$ samples to achieve $\widetilde{\mathcal{O}}\left(\frac{1}{\sqrt{T}}\right)$ iteration complexity for $||\nabla J(\pi_{\theta_t})||^2$, or equivalently, they obtain the sample complexity of $\widetilde{\mathcal{O}}(\varepsilon^{-4})$.
> > >
> > > By contrast, our algorithm uses single-timescale update, where the critic only updates once at each iteration. Consequently, we cannot obtain the uniform bound $C_{\omega}$ on critic's error at each iteration, which makes our analysis very different from theirs. In particular, we have to characterize the descent property of the critic's error at each iteration. To this end, we explore the intrinsic problem structure and reveal the smoothness of the optimal critic variable, which allows us to derive the sufficient descent of the critic's error with an additional error term. We can then construct a Lyapunov function so that this additional error term can be absorbed into the ascent term of the actor's objective, yielding the desired sample complexity of $\widetilde{O}({\varepsilon^{-2}})$.
> > >
> > > Finally, we remark that the work [Qiu et al., 2021] considers the  the averaged reward setting, while we consider the discounted one. This results in different forms of objective functions.
> > > Once again, thank you so much for your reply. Please feel free to discuss with us if you have any other concern.
> > >
> > > **Reference**
> > >
> > > [Qiu et al., 2021]. “On finite-time convergence of actor-critic algorithm”. IEEE Journal on Selected Areas in Information Theory, 2021.

---

### Official Review · Reviewer_fGDk · 2022-07-10

**Rating:** 6
**Confidence:** 5
**Soundness:** 3 good
**Presentation:** 3 good
**Contribution:** 2 fair

**Summary:**

This paper develops the finite-time analysis of a single-timescale multiagent actor-critic algorithms. The sample complexity is $O(\epsilon^{-2}$, which matches with the sota. Many existing studies use approaches e.g., double loop, minibatch to simplify the analysis, which introduces additional parameters in practice, and thus are not preferred. This paper shows that a single time scale update can also achieve the same order of sample complexity for the problem of decentralized MARL.


**Questions:**

There is one paper in the literature also shows that a single-time scale update can also achieves the same sample complexity as those approaches using minibatch or double loop. Their algorithm does not need a double-loop structure not a mini-batch approach, and only uses one single sample at each time step. The analysis is novel in that they bound the tracking error, (or equivalently, the critic error in this paper) using the gradient norm of the objective so that they can obtain a tighter complexity bound. How does the approach in this paper compare to that one?
Non-Asymptotic Analysis for Two Time-scale TDC with General Smooth Function Approximation, NeurIPS 2021.



**Limitations:**

The algorithm in this paper requires the action to be globally observable, which may not be possible in practice.

**Strengths And Weaknesses:**

Strengths:
1. The analysis in this paper is of more practical interest as it focuses on the single-timescale update rule, which does not involve additional tuning parameters e.g., batch size, inner loop iterations.

2. The complexity of the single-time scale method matches with those double-loop, mini-batch methods in the literature.


Weaknesses:

1. This is not essentially a weaknesses, more like a suggestion. It shall be highlighted in the paper why those double-loop and minibatch approaches with inner loop iterations =1, minibatch size =1, fail to obtain a complexity of $O(\epsilon^{-2}$; and what term needs a tighter bounds than those approaches?

---

> ### Author Response · Authors · 2022-08-02
> **Author's Response**
>
> Thanks so much for the supportive and valuable comments. Let us address the your concerns in a point-by-point manner below. Major changes in the manuscript are highlighted in blue.
>
> **A. Suggestion of highlighting the inner loop size and batch size of double-loop scheme.**
> To analyze the errors of critic $||\omega^*(\theta_k) - \omega_{k+1}^{i}||^2$ and reward estimator $||\lambda^*(\theta_k) - \lambda_{k+1}^{i}||^2$,
> the double-loop approach runs lower-level update for $\mathcal{O}(\log(\varepsilon^{-1}))$ times with batch size $\mathcal{O}(\varepsilon^{-1})$ to drive these errors below $\varepsilon$ and hence, they cannot allow inner loop size and bath size to be $\mathcal O(1)$ simultaneously. We have added this highlight in the revised manuscript; see lines 265--267.
>
> **B. Comparison with the related work [Wang et al., 2021].**
> The work [Wang et al., 2021] studies the temporal-difference learning with gradient correction (TDC). When analyzing the convergence of the algorithm, the essential challenge of our work and [Wang et al., 2021] is the same, which is to bound the product term involving $\omega^*(\theta_{k+1}) - \omega^*(\theta_k)$ (or $\omega(\theta_{t+1}) - \omega(\theta_t)$ in their paper). However, the technique we use for bounding this term is different from theirs, mainly due to different problem structure. Specifically, [Wang et al., 2021] utilizes the problem structure nicely so that they are able to decompose it into a term of order $\mathcal{O}(||\nabla J(\theta_t)||)$ and other terms with lower orders, and thereby establish the descent property of the tracking error $||z_{t}||^2$. In comparison, our approach transform the error into $\omega^*(\theta_{k+1}) - \omega^*(\theta_k) - \nabla \omega^*(\theta_k)^T(\theta_{k+1}-\theta_k) + \nabla \omega^*(\theta_k)^T(\theta_{k+1}-\theta_k)$, where the norm of the first three terms are of order $\mathcal{O}(||\theta_{k+1}-\theta_k||^2)$ due to the smoothness of $\omega^*(\theta)$ (the key ingredient established in our paper), and the norm of the last term will be absorbed in our Lyapunov function. We thank the reviewer for pointing out this related work. We have added it in our literature review; see lines 53--55 in our revised manuscript.
>
> **C. Global action observation.**
> In fact, we did have the noisy reward version of our proposed algorithm, which does not require access to global action. Please see Appendix B for the noisy reward version algorithm and its analysis.
>
> We hope that our response and revisions are satisfactory to the reviewer and that all concerns have been addressed appropriately.
>
> **Reference**
>
> [Wang, et al., 2021]. "Non-asymptotic analysis for two time-scale TDC with general smooth function approximation." NeurIPS 2021.

---

### Official Review · Reviewer_T2LJ · 2022-07-13

**Rating:** 5
**Confidence:** 5
**Soundness:** 3 good
**Presentation:** 3 good
**Contribution:** 2 fair

**Summary:**

This paper studies a single timescale decentralized actor-critic algorithm. The setting is as follows. Each agent attain their respective action, and reward. There is also a global state which is observed by all the agents. Each of the agents attain a low low dimensional approximation of the rewards function, the value function, and the policy. At each time step the agents update these parameters using a single sample from the environment, either iid or Markovian. The authors show the sample complexity of their algorithm for both iid and Markovian sampling, and show that it attains the lower bound for such sample complexity.

**Questions:**

- You did not define what is \lambda_max in line 205.
- Can you express the communication cost of the algorithm and compare it with the related work along with the sample complexity?
- Why not characterize the convergence for natural actor critic as well? I believe establishing the result should be straightforward given the current results. Note that natural actor-critic can guarantee global convergence.

**Limitations:**

The main suggestion for improvement has already been suggested. I suggest to include the convergence of the natural actor critic as well as the communication cost of the algorithm.

**Strengths And Weaknesses:**

- My main criticism of this work is that I do not see much advantages of the results in this paper over the one in [5]. The authors argue that they study a single timescale algorithm compared to [5] which studies a two timescale algorithm. However, I have two comments about that. First of all, I believe the result in [5] is not two timescale. In particular, the step sizes for both actor and critic are constant. The only issue with [5] might be the need to have an inner loop and an outer loop. But I do not see a significant problem with this structure. The reason is that the sampling is not affected during the move from inner loop and outer loop. In [5] the sampling is through a single trajectory throughout the performance of the algorithm. Can the authors explain why their results are better than [5], or why their algorithm has an advantage over [5]? I am willing to increase my score if this is addressed.
- The main strength of the paper is a convergence bound which attains the lower bound up to logarithmic factors.

---

> ### Author Response · Authors · 2022-08-02
> **Author's Response: Part II**
>
>
> **E. NAC and global convergence.**
> Thanks so much for the nice suggestion. We have designed single-timescale NAC and provided its convergence results to global optimality in Appendix F in the revised manuscript.
>
> In terms of the algorithm design, our proposed AC can be straightforwardly extended to NAC by adopting [Chen et al., 2022]'s idea on estimating natural policy gradient in a decentralized way. However, the analysis of AC cannot be directly adjusted to NAC when analyzing the ascent of the objective, since it will result in a bias term between the true gradient and the approximated natural gradient, which is difficult to bound. To this end, we derived a matrix decomposition trick on the Fisher information matrix to recover it back to the original gradient bias term, and thus derive the ideal sample complexity; see equation (139) in the revised manuscript.
>
> We obtained the sample complexity $\widetilde{\mathcal{O}}(\varepsilon^{-3})$ for $||\nabla J(\theta_k) ||^2$, which matches the best existing sample complexity of NAC under both decentralized setting [Chen et al., 2022] and single-agent setting [Xu et al., 2020]. In terms of the global optimality gap, the algorithm requires $K=\mathcal{O}(\varepsilon^{-4})$ iterations to achieve $\varepsilon$-error, and thus has $\widetilde{\mathcal{O}}(\varepsilon^{-6})$ sample complexity. Such a sample complexity is much worse than the best existing sample complexity of $\widetilde {\mathcal{O}}(\varepsilon^{-3})$ [Chen, et al., 2022; Xu et al., 2020].  Let us explain the intuition of this gap for the sample complexity below.
>
> Mimicking  the analysis of [Chen, et al., 2022] allows to establish the following inequality
>
> $$\small \frac{1}{K}\sum_{k=0}^{K} J\left(\theta^{*}\right) - \mathbb{E}[J(\theta_k)] \leq  \mathcal{O}\left(\frac{1}{K}\sum_{k=1}^{K}\mathbb{E}[||\nabla J(\theta_k)||^2] \right) + \mathcal{O}\left(\frac{1}{K}\sum_{k=1}^{K}\mathbb{E} [|| \omega_k - \omega^{ * }(\theta_k)||]\right) + \mathcal{O}\left(\frac{1}{K\alpha_k}\right)$$
>
> While our analysis can obtain the iteration complexity of $\mathcal{O}(\frac{1}{\sqrt{K}})$ for $||\nabla J(\theta_k)||^2$, we can only achieve $\mathcal{O}(\frac{1}{K^{1/4}})$ iteration complexity for critic's error $||\omega_k - \omega^*(\theta_k)||$. This is because our algorithm uses single-timescale update, where the critic's error inevitably converges slower than that of double-loop based algorithms which have $\mathcal{O}(\frac{1}{\sqrt{K}})$ complexity for the critic's error at each iteration. Therefore, the sample complexity in terms of global optimality gap of our single-timescale NAC is dominated by this critic's error term, resulting in the final complexity of $\widetilde{\mathcal{O}}(\varepsilon^{-6})$.
>
> We remark that this sample complexity result is based on a straightforward application of the analysis of [Chen et al., 2022], which is designed for double-loop algorithm. Therefore, such a proof technique may not be the tightest one for single-timescale NAC (intuitively, this result is not tight). We leave the research on the improvement of such highly suboptimal results of single-timescale NAC as a future work.
>
> We hope that our response and revisions are satisfactory to the reviewer and that all concerns have been addressed appropriately.
>
> **References**
>
> [Chen et al., 2022]. "Sample and communication-efficient decentralized actor-critic algorithms with finite-time analysis." ICML 2022.
>
> [Xu et al., 2020]. "Improving sample complexity bounds for actor-critic algorithms." NeurIPS 2020.

---

> ### Author Response · Authors · 2022-08-02
> **Author's Response: Part I**
>
> Thanks so much for the supportive and valuable comments. Let us address the your concerns in a point-by-point manner below. Major changes in the manuscript are highlighted in blue.
>
> **A. [Chen et al., 2022] is two-timescale.**
> There might be a misunderstanding about our literature review. In fact, we did state that [Chen et al, 2022] implements a double-loop algorithmic framework instead of a two-timescale one; see lines 36--43 in the revised manuscript.
>
> **B. Advantages over [Chen et al., 2022].**
> The main difficulty for implementing the double-loop scheme in [Chen et al., 2022] lies in the hyper-parameter tuning instead of sampling. Specifically, their algorithm introduces two additional hyper-parameters that have large influence on the model's performance, which are the batch size and inner loop size. We refer to Figures 2-3 in our Appendix A to see the influence of these two hyperparameters on the performance of the double-loop algorithm in [Chen et al., 2022]. Theoretically, the batch size and inner loop size in their double-loop algorithm need to be of order $\mathcal{O}(\varepsilon^{-1})$ and  $\mathcal{O}(\log(\varepsilon^{-1}))$, respectively, in order to achieve their sample complexity results. In practice, it might be hard to pre-determine the accuracy level $\varepsilon$ and hence, these two hyperparameters usually needs to be manually tuned. By contrast, our proposed single-timescale approach admits the batch size and inner loop size to be 1. In addition, our theory can be directly extended to cover any batch size and inner loop size that are of order $\mathcal O(1)$. We have updated the corresponding parts in introduction in the revised manuscript; see lines 41--43.
>
> We have to clarify that our goal is not to show that the single-timescale algorithmic framework is superior to the double-loop one. Instead, we believe that both schemes have pros and cons. Our objective is to provide a theoretical understanding for the AC algorithms based on single-timescale update, which is commonly utilized in practice.
>
> **C. Definition of $\lambda_{\max}$.**
> We have added the definition; see line 206 in the revised manuscript.
>
> **D. Communication cost of the algorithm.**
> We have added Table 1 in Appendix G in the revised manuscript, which gives an overview about several related and representative works on communication cost. When we have more space, we will move this table to the main content.

---

> ### Author Response · Authors · 2022-08-09
> **Feedback & Additional Remarks and Questions (if any)**
>
> Dear Reviewer T2LJ,
>
> We hope that our response addresses most of your concerns. Specifically, we have made clarifications on [chen et al., 2022]'s algorithmic framework and stated our algorithm's advantage over it. We have also provided the communication complexity summary (Appendix G in the revised manuscript) and single-timescale natural Actor-Critic's convergence result (Appendix F in the revised manuscript).
>
> Since the deadline of the discussion period is approaching, we would highly appreciate to receive feedback from you. If you have any further questions or remarks, please let us know so that we will have enough time to address your concerns. We will be more than happy to provide additional clarifications and details.
>
> Best,
>
> Authors.

---

> ### Author Response · Authors · 2022-08-09
> **Looking Forward to the Feedback**
>
> Dear Reviewer T2LJ,
>
> Since the deadline of the open-discussion period is within hours, we eagerly look forward to your feedback on our response. If you think we have addressed most of your concerns, we would greatly appreciate if you could reconsider your score (as indicated in your initial review report).
>
> Best,
>
> Authors.

---

### Author Response · Authors · 2022-08-08
**Further Questions and Remarks (if any)**

Dear Reviewers,

We hope that our responses address most of your concerns. If you have any further questions or remarks, please let us know and we will be more than happy to answer them before the end of the discussion period.

Best,

Authors.

---

### Meta-Review · Area_Chair_yiCp · 2022-08-27

**Recommendation:** Reject
**Confidence:** Certain

**Metareview:**

This paper proposes and analyzes the convergence rate of a single-timescale actor-critic algorithm. The reviewers reached the consensus that it is above the borderline-acceptance bar. However, after an in-depth discussion during the reviewer-metareviewer discussion period, the reviewers pointed out some critical problems: although the superiority of the proposed algorithm over classic double-loop algorithms is validated, that over previous single-timescale single-loop is not discussed. One reviewer carefully checked both the original and the revised version of the paper but still claimed that extending the proof technique adopted in this paper would lead to a worse convergence rate compared with previous work. In general, I agree with the reviewer. Thus, I think the current version of the paper does not make a compelling case for its archive value due to the lack of a more rigorous discussion of related issues.


**Award:**

No

---

### Decision · Program_Chairs · 2022-09-14

Reject